# Poisson-Algebraic Parallel Scan: A fast symplectic framework for neural Hamiltonians

## Abstract

Learning Hamiltonian neural networks (HNNs) that respect the intrinsic symplectic structure of physical systems has emerged as a foundational framework for robust long-term predictions in scientific machine learning. Nevertheless, existing HNN methods face critical limitations: (i) inherent sequential integration prevents parallel computation, causing significant computational bottlenecks, and (ii) unconstrained neural architectures lead to instability when extrapolating dynamics beyond training regimes. To address these fundamental challenges, we introduce *Poisson-Algebraic Parallel Scan (PAPS)*, a novel framework that leverages a carefully constructed Poisson algebraic decomposition of the learned Hamiltonian. By embedding polynomial generators explicitly closed under Poisson brackets, PAPS induces an associative Lie-group structure that naturally facilitates parallel-scan (prefix-sum) computation. Our method achieves exact symplectic integration with up to three orders-of-magnitude ($1000\times$) speedup at $10^3$ integration steps, significantly outperforming existing HNN approaches. Moreover, the structured algebraic representation inherent in PAPS ensures intrinsic physical consistency, delivering stable and reliable extrapolation far beyond the training distribution. Extensive theoretical analyses and rigorous numerical experiments validate the superior computational scalability of our approach, highlighting PAPS as a powerful new direction for scalable and physically consistent neural Hamiltonian modeling.

## 1 Introduction

Learning data-driven models that respect the *symplectic* structure of Hamiltonian mechanics has emerged as a principled and rapidly evolving direction within the broader AI4Science initiative, driven by the need for robust long-horizon prediction in physics (Li et al., 2020), control, and molecular dynamics simulation (Zhang et al., 2025a). Pioneering frameworks showed that embedding canonical invariants into the hypothesis space mitigates energy drift and stabilizes learning. Despite these advances, existing symplectic-learning approaches face critical limitations. Firstly, conventional HNNs must apply sequential updates $M$ times to predict a horizon of $M$ steps, incurring an unavoidable $\mathcal{O}(M)$ computational latency that severely restricts scalability on modern parallel accelerators. Furthermore, this inherently sequential nature is *compounded* by the nonlinear black-box neural network structure of learned force fields. As a result, standard acceleration strategies like operator-splitting (Bartel et al., 2023) become inapplicable without compromising the underlying Poisson structure, further limiting computational efficiency and scalability. Secondly, prevailing HNN frameworks are trained through a *point-wise (instantaneous) force-matching objective*: an unconstrained black-box generator $g_\theta$ is optimized to regress the accelerations $(\dot{\mathbf{q}}, \dot{\mathbf{p}}) = (\nabla_p H_\theta, -\nabla_q H_\theta)$ at each observed state. This purely local criterion ignores the fact that many practical settings pose a *global physical inverse problem* only initial conditions and noisy trajectory traces are available, while the latent Hamiltonian (or its force field) must be inferred so as to reproduce the entire path. As a result, models calibrated by instantaneous force regression often extrapolate poorly, deviating from physically admissible solution manifolds once they venture beyond the training distribution.

Meanwhile, recent advances in *parallel scan (prefix-sum)* algorithms (Hillis & Steele, 1986; Blelloch, 1990) within the broader ML community offer a compelling potential solution to these critical limitations. By reorganizing computations into associative binary trees, parallel scans achieve massive GPU/TPU throughput, reducing the critical computational depth from $M$ steps to $\mathcal{O}(\log M)$, as successfully demonstrated in large-scale attention (Dao et al., 2022) and structured state-space

sequence models (Gu et al., 2021; Smith et al., 2023). However, conventional Hamiltonian generators produce nonlinear flow maps that do not exhibit clear associativity, thereby preventing direct use of scan-based acceleration techniques.

To unlock this parallelism while simultaneously addressing extrapolation instability, we explicitly embed our learned Hamiltonian within a carefully constructed *Poisson algebraic structure* (Marsden & Ratiu, 1999; Abraham & Marsden, 1978), thereby modernizing Hamiltonian dynamics for the machine-learning community while offering an algebraic basis for workflows. Concretely, we represent the Hamiltonian as a structured polynomial built from simpler blocks that are closed under the Poisson bracket. This algebraic decomposition ensures that the flow maps form a finite-dimensional Lie group, satisfying the associativity needed for parallel scans. Each flow map is thus guaranteed to be a *symplectomorphism* (McDuff & Salamon, 1998; Arnol'd, 1989), preserving the underlying physical consistency. Consequently, our algebraic formulation not only enables stable extrapolation far beyond training data distributions but also provides an exact, log–depth parallel integration scheme, overcoming both core limitations of conventional HNNs simultaneously. Overall, our contributions in this work are therefore twofold:

**(1) Acceleration of Physical Simulation with Parallel Scan.** We introduce the *Poisson–Algebraic Parallel Scan (PAPS)*, a novel parallel-scan framework based on neural Hamiltonians leveraging Poisson algebraic decomposition. PAPS achieves exact symplecticity and up to $(10^3 \times)$ **acceleration** compared to existing Hamiltonian Neural Network (HNN) variants, by enabling logarithmic-time trajectory prediction through an associative prefix-scan approach.

**(2) Poisson-Algebraic Structured Inverse Problem.** We establish that our trajectory-matching framework robustly solves the inverse Hamiltonian learning problem. By leveraging global trajectory consistency rather than local force labels, our Poisson-algebraic neural integrator provides stable and physically meaningful extrapolation beyond the training regime, significantly outperforming conventional force-matching approaches that often suffer from instability during extrapolation.

To lay the groundwork for our framework, we first briefly review the necessary mathematical preliminaries and notation. Let $(\mathbb{R}^{2n}, \omega)$ be a $2n$-dimensional symplectic space, and let $z := (\mathbf{q}, \mathbf{p}) = (q_1, \ldots, q_n, p_1, \ldots, p_n)$ denote canonical coordinates associated with the standard Darboux form $\omega = \sum_{i=1}^{n} dq_i \wedge dp_i$. For each sufficiently smooth scalar function $g \in C^\infty(\mathbb{R}^{2n})$, referred to as a *Hamiltonian generator*, we associate a unique vector field $X_g$, called the Hamiltonian vector field of $g$, implicitly defined through the relation $\iota_{X_g} \omega = dg$, explicitly yielding $X_g = \sum_{i=1}^{n}(\partial_{p_i} g)\partial_{q_i} - (\partial_{q_i} g)\partial_{p_i}$. Furthermore, given any two smooth functions $f, g \in C^\infty(\mathbb{R}^{2n})$, the *Poisson bracket* is defined as $\{f, g\} := \omega(X_f, X_g) = \sum_{i=1}^{n}(\partial_{q_i} f\, \partial_{p_i} g - \partial_{p_i} f\, \partial_{q_i} g)$, providing a fundamental algebraic structure for analyzing Hamiltonian dynamics.

**Problem Formulation : Structured Inverse Problem.** Recall that the primary goal of *Hamiltonian neural networks* (HNNs) is to approximate an unknown scalar-valued Hamiltonian function $H_{\text{true}} : \mathbb{R}^{2n} \to \mathbb{R}$ using a neural network $g_\theta : \mathbb{R}^{2n} \to \mathbb{R}$. Upon successful training, the learned model $g_\theta$ serves as a Hamiltonian *generator*, governing the dynamical evolution via the Poisson bracket $\{\cdot, \cdot\}$:

$$\dot{z} = \underbrace{\{z, g_\theta\} = J\nabla_z g_\theta(z)}_{\text{Learned Dynamics}} \approx \underbrace{J\nabla_z H_{\text{true}}(z) = \{z, H_{\text{true}}\}}_{\text{Exact Dynamics}}, \quad J = \begin{bmatrix} 0 & \mathbf{I}_n \\ -\mathbf{I}_n & 0 \end{bmatrix}, \quad (1)$$

where $J$ denotes the symplectic matrix and $z \in \mathbb{R}^{2n}$ represents the state vector in physical space. To achieve this, we consider an initial condition $\hat{z}_0 \sim \mathbb{P}_{\text{init}}$ with corresponding trajectory data $\hat{z}_t \sim \mathbb{P}_{t>0}$. This setting defines a *physical inverse problem*, aiming to identify the parametrized Hamiltonian $g_\theta(z)$ by solving the following initial value optimization problem:

$$\min_{g_\theta \in \mathcal{H}} \mathbb{E}_{\hat{z}_0 \sim \mathbb{P}_{\text{init}}, t \sim p_t} \left[ \mathbb{E}_{\hat{z}_t \sim \mathbb{P}_{t>0}} \| z_t(g_\theta) - \hat{z}_t \|^2 \big| \hat{z}_0 \right] \xrightarrow[\text{Section 2}]{\textbf{Polynomization}} \mathcal{H} := \text{Poly}_r \ni g_\theta \approx H_{\text{True}}. \quad (2)$$

In our framework, we denote by $\mathcal{H}$ the hypothesis space in which the Hamiltonian generator is learned. Rather than allowing $g_\theta$ to range over arbitrary neural function classes, the *structured inverse problem* restricts the search to a carefully chosen $\mathcal{H}$. Such structure is essential both for computational tractability and for preserving physical invariants. In the next section, we show how $\mathcal{H}$ can be instantiated explicitly through a *polynomization* procedure, where the generator is expanded in a finite polynomial basis. This construction yields an algebraic representation naturally compatible with the underlying Poisson geometry, enabling stable extrapolation and efficient parallel composition.

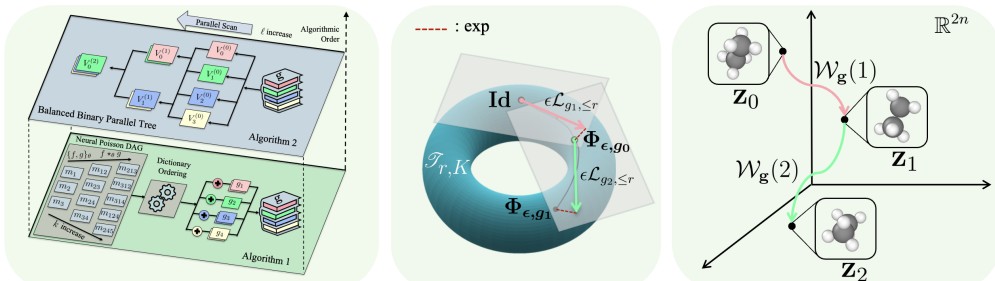

Figure 1: **Pipeline of Proposed Method.** Arrow colors represent the operations of their respective generators. *(left) Two-stage symplectic prefix–scan framework.* Algorithm 1 constructs a Neural-Poisson DAG, dictionary-orders the polynomial Hamiltonian generators $\mathbf{g}$. Algorithm 2 folds these transforms in a balanced binary tree, yielding an $O(\log M)$ parallel scan that composes them into the final structure-preserving symplectomorphism. *(mid) Conceptual illustration of the Lie–group walk on $\mathcal{T}_{r,K}$.* Successive transformation $\exp(\epsilon\mathcal{L}_{g_1,\leq r})$ and $\exp(\epsilon\mathcal{L}_{g_2,\leq r})$ carry $\mathbf{Id} \to \Phi_{\epsilon,g_0} \to \Phi_{\epsilon,g_1}$. Dashed red and solid-green arrows mark the corresponding Lie-transform paths. *(right) Phase-space trajectories from prefix scans.* Composite flows $\mathcal{W}_{\mathbf{g}}(1)$ and $\mathcal{W}_{\mathbf{g}}(2)$ carry (*e.g.*, molecular) states $z_0 \to z_1 \to z_2$.

**Poisson-algebraic Parallel Scan.** Given initial conditions $\hat{z}_0 \sim P_{\text{init}}$ and corresponding observed trajectory data $\hat{z}_t \sim P_{t>0}$, the objective is to globally infer the Hamiltonian generator $g_\theta$. Within the PAPS framework, this physical inverse problem can be equivalently reformulated as

$$\mathcal{L}(\theta) = \min_{g_\theta \in \mathcal{G}} \mathbb{E}_{\hat{z}_0,t}\left[\mathbb{E}_{\hat{z}_t}\|z_t(g_\theta) - \hat{z}_t\|^2\big|\hat{z}_0\right] = \min_\theta \mathbb{E}_{\hat{z}_0,t}\left[\mathbb{E}_{\hat{z}_t}\|\mathcal{W}_{\mathbf{g}}(\lceil T/t\rceil)\hat{z}_0 - \hat{z}_t\|^2\big|\hat{z}_0\right], \quad (3)$$

where $\mathcal{W}_{\mathbf{g}}(m)$ denotes the $m$-step composite flow induced by an ordered sequence of small symplectic updates. Its concrete Lie–group realization, which forms the core algorithmic contribution of this work, will be developed in Section 3. Rather than advancing the dynamics by applying these updates sequentially, PAPS reorganizes them into an associative *parallel scan* over Hamiltonian flow blocks. This design exposes GPU-friendly parallelism: on modern accelerators, the effective depth of an $M$-step trajectory is reduced from $\mathcal{O}(M)$ to $\mathcal{O}(\log M)$ while maintaining essentially linear throughput in the number of particles and trajectories. In practice, PAPS yields empirical speedups of three orders of magnitude ($10^3\times$) over classical symplectic integrators, while preserving energy and the symplectic form to numerical precision. Designing and analyzing this GPU-efficient parallel-scan integrator for Hamiltonian dynamics is the central computational objective of our work.

## 2 ALGEBRAIC-POLYNOMIZATION OF HAMILTONIAN GENERATORS

Before introducing the algebraic machinery, we first *delineate the hypothesis space $\mathcal{H}$* on which our model is built, ensuring that all subsequent constructions rest on a clear and consistent foundation.

**Neural Polynomial Generator.** To set the stage, our first step is to transform the conventional black-box neural network structure of the Hamiltonian generator into an explicitly structured form, termed an *algebraic decomposition* for further acceleration and analysis. We refer to this structured polynomial representation as a *Neural Polynomial Generator*, whose finite polynomial basis systematically encodes the underlying Poisson–algebraic structure.

**Definition 2.1** (Neural Polynomials). Let us consider neural network $\{\mathbf{c}_{\boldsymbol{\alpha},\boldsymbol{\beta}} := c(\alpha, \beta, z_0, t; \theta)\} \subset \mathcal{C}_\theta$, called *neural coefficients*. Given Darboux coordinates $(\mathbf{q}, \mathbf{p})$ with monomials $m_{\beta^k}^{\alpha^k}(z)$ of degree $k \leq r$, we define the associated family of parameterized polynomials as

$$g_\theta \in P_{\leq r}^\theta = \text{span}_{\mathcal{C}_\theta}\left\{m_{\boldsymbol{\beta}}^{\boldsymbol{\alpha}} = q_n^{\boldsymbol{\alpha}} p_n^{\boldsymbol{\beta}} \,\Big|\, |\boldsymbol{\alpha}| + |\boldsymbol{\beta}| \leq r\right\} \supseteq \left\{\sum_{k=1}^{|\mathcal{C}|} \mathbf{c}_{\boldsymbol{\alpha},\boldsymbol{\beta}} \cdot m_{\boldsymbol{\beta}}^{\boldsymbol{\alpha}} \,\Big|\, |\mathcal{C}| = \binom{2n+r}{r}\right\}.$$

*Intuition.* By substituting the black-box neural network with a finite, structured *polynomial basis*, we obtain an explicit algebraic framework for representing Hamiltonian functions. This construction is analogous to composing music from a predetermined set of notes, in which each melody is built from structured and interpretable elements. Crucially, the polynomial basis is closed under the Poisson bracket operation, which endows the space of Hamiltonians with a rich algebraic structure. As we will demonstrate in Section 3, this intrinsic algebraic organization is fundamental to enabling parallel prefix-scanning, thereby achieving the target computational complexity of $\mathcal{O}(\log M)$. Formally, in an arbitrary $2n$-dimensional phase space with coordinates $z = (q_1, \ldots, q_n, p_1, \ldots, p_n)$ and polynomial

multi-indices $\boldsymbol{\alpha}, \boldsymbol{\beta} \in \mathbb{N}^n$, we represent the neural polynomial Hamiltonian generator explicitly as:

$$g_\theta(q, p) = \sum_{\boldsymbol{\alpha}, \boldsymbol{\beta}, \, |\boldsymbol{\alpha}| + |\boldsymbol{\beta}| \le r} \underbrace{c(\alpha, \beta, z_0, t; \theta)}_{=\mathbf{c}_{\boldsymbol{\alpha}, \boldsymbol{\beta}}} \cdot \underbrace{q_1^{\alpha_1} \cdots q_n^{\alpha_n} p_1^{\beta_1} \cdots p_n^{\beta_n}}_{m_{\boldsymbol{\beta}}^{\boldsymbol{\alpha}} = q_n^{\boldsymbol{\alpha}} p_n^{\boldsymbol{\beta}}}. \tag{4}$$

Given the $i^{\text{th}}$ unit vector $\mathbf{e}_i$, the Hamiltonian dynamics introduced in equation 1 can be explicitly rewritten by using the learned polynomial generator in equation 4 as follows:

$$\dot{z} = J\nabla_z g_\theta(z) = \left[\dot{q}_i = \frac{\partial g_\theta}{\partial p_i} = \sum_{\boldsymbol{\alpha}, \boldsymbol{\beta}} c_{\boldsymbol{\alpha}, \boldsymbol{\beta}} \beta_i q^{\boldsymbol{\alpha}} p^{\boldsymbol{\beta} - \mathbf{e}_i}, \; \dot{p}_i = -\frac{\partial g_\theta}{\partial q_i} = -\sum_{\boldsymbol{\alpha}, \boldsymbol{\beta}} c_{\boldsymbol{\alpha}, \boldsymbol{\beta}} \alpha_i q^{\boldsymbol{\alpha} - \mathbf{e}_i} p^{\boldsymbol{\beta}}\right]^T$$

While the above expansion provides explicit dynamics, its true advantage lies in the algebraic closure under multiplication and Poisson brackets. This observation motivates an abstract viewpoint. In our setting, the *hypothesis space* $\mathcal{H}$ in Eq 2 itself is regarded as an algebraic structure. Concretely, by embedding the polynomial model space $P_{\le r}^\theta$ into an algebraic system $\mathcal{H} = \left(P_{\le r}^\theta, \star_\theta, \{\cdot, \cdot\}_\theta\right)$ that supports both product and bracket operations, one obtains a richer and more structured representation. This perspective leads naturally to the following definition.

> **Definition 2.2** (Neural Poisson algebra). For a set of pairs $f, g \in P_{\le r}^\theta$ on the neural polynomials $P_{\le r}^\theta{}^a$, we define pointwise product $f \star_\theta g := \pi_{\le r}(fg)$ and canonical Poisson bracket $\{f, g\}_\theta := \pi_{\le r}(\{f, g\})$. Then, we call the space $\left(P_{\le r}^\theta, \star_\theta, \{\cdot, \cdot\}_\theta\right)$ the *neural Poisson algebra*.
>
> ---
> $^a$Note the projection $\pi_{\le r} : C^\infty(M) \to P_{\le r}$ simply discards every monomial whose total degree exceeds $r$.

While both operations $\star_\theta : P_{\le r}^\theta \times P_{\le r}^\theta \to P_{\le r}^\theta$ and $\{\cdot, \cdot\}_\theta : P_{\le r}^\theta \times P_{\le r}^\theta \to P_{\le r}^\theta$ are closed, any finite sequence of multiplications and Poisson brackets starting from neural polynomials remains in $P_{\le r}^\theta$. Consequently, beginning with a basis of monomials whose coefficients lie in $\mathcal{C}_\theta$, one may generate an arbitrarily rich family of candidate Hamiltonians without ever leaving the truncation subspace.

**Neural-Poisson Composition DAG.** To generate all possible products and Poisson-bracket combinations within our neural Poisson algebra, we organize the procedure to create a *directed acyclic graph* (*i.e.*, DAG) of Hamiltonian generators. Starting from the base set $\mathcal{G}^{(0)} = P_{\le r}^\theta$, each depth-$k$ layer is constructed by systematically applying exactly two operations (*i.e.*, $\star_\theta, \{\cdot, \cdot\}_\theta$) to every ordered pair in the previous layer, yielding $\mathcal{G}^{(k)}$. By traversing this DAG, we can both guarantee strict closure within $P_{\le r}^\theta$ and leverage depth-wise batching of algebraic operations.

> **Proposition 2.3** (Neural–Poisson Composition DAG). *Let $P_{\le r}^\theta$ be the base generators and $\mathcal{C}_\theta$ the neural coefficient set. For each $k \ge 1$, let us consider the quotient space of Hamiltonian generators:*
>
> $$\mathcal{G}^{(k)} = \left\{ \{\pi_{\le r}(f \star_\theta g)\} \bigcup \{\pi_{\le r}(\{f, g\}_\theta)\} / \sim \; \Big| \; \forall f, g \in \mathcal{G}^{(k-1)} \right\}, \quad \mathcal{G}^{(0)} = P_{\le r}^\theta,$$
>
> *Then, the composition full graph $\mathcal{G}^{(\le K)} := \bigcup_{k=0}^K \mathcal{G}^{(k)}$ is DAG of depth $K$, where an equivalence relation $\sim$ induces quotient space $\mathcal{G}^{(k)}$ with following three rules (e.g., product, Poisson bracket, Jacobi identity):*
>
> $$f \star_\theta g \sim g \star_\theta f, \quad \{f, g\}_\theta \sim -\{g, f\}_\theta, \quad \{f, \{g, h\}_\theta\}_\theta + \{g, \{h, f\}_\theta\}_\theta + \{h, \{f, g\}_\theta\}_\theta \sim 0.$$

*Motivation for DAG Structure.* By embedding our neural polynomial space $P_{\le r}^\theta$ into a structured Poisson algebra, we systematically construct a *Directed Acyclic Graph* (DAG) of Hamiltonian generators endowed with two essential algebraic operations: the product $f \star_\theta g$, which enlarges the state space by stacking features (analogous to channel stacking in CNNs), and the Poisson bracket $\{f, g\}_\theta = \nabla_q f^\top \nabla_p g - \nabla_p f^\top \nabla_q g$, which acts as a phase-space filter explicitly capturing how each pair of position and momentum coordinates $(\mathbf{q}, \mathbf{p})$ twists the underlying flow (analogous to spatial convolution filters in CNNs, $(\mathbf{x} * \mathbf{w})[i] = \sum_j x[i-j]w[j]$ for pixels $\mathbf{x}$ and $\mathbf{w}$). To illustrate concretely, let us consider a simple initial scenario with polynomial degree cutoff $r = 2$. Starting from a neural coefficient set at depth 0, the DAG construction unfolds explicitly as follows:

> **Example: Polynomial Expansions of Hamiltonian Generators in DAG Construction ($r = 2, n = 2$)**
>
> $\mathcal{G}^{(0)}: \left\{ c_1 q_1 + c_2 q_2 + c_3 p_1 + c_4 q_1 q_2 + c_5 q_1 p_2 + c_6 q_2 p_1 + c_7 q_2 p_2 + + c_8 p_1 p_2 + c_9 p_2^2, \dots \right\} := P_{\leq r}^{\theta}$ **in Def 2.1**
>
> $\mathcal{G}^{(1)}: \left\{ c_{12} q_1 \star_\theta q_2, \underline{c_{21} q_2 \star_\theta q_1}, c_{23} \{q_2, p_1\}, c_{24} q_2^2 \star_\theta p_2, c_{34} \{q_1^2, q_1 q_2\}, \dots \right\}$, **Equivalence in Product**
>
> $\mathcal{G}^{(2)}: \left\{ \overline{c_{123} \{\{q_1, q_2\}, p_1\}}, c_{231} \{\{q_2, p_1\}, q_1\}, c_{312} \{\{p_1, q_1\}, p_2\}, \dots \right\}$, **Equivalence in Jacobi Identity**

In the toy example above ($r = 2, n = 2$), the directed acyclic graph (DAG) grows by iteratively combining elementary polynomial representations through Poisson bracket and product operations. The crossed-out terms in the diagram correspond to representations eliminated due to algebraic equivalences. Formally, given arbitrary neural vector-valued coefficients $\mathbf{c}_{\alpha,\beta}(z, t; \theta) \in \mathbb{R}^d$, higher-order coefficient tensors are recursively constructed as follows:

$$\mathbf{c}_{\boldsymbol{\alpha},\boldsymbol{\beta}}(z, t; \theta) = \mathbf{c}_{\boldsymbol{\alpha}_{1:m-1}, \boldsymbol{\beta}_{1:m-1}}(z, t; \theta) \otimes \mathbf{c}_{\alpha_m, \beta_m}(z, t; \theta) - \mathbf{c}_{\alpha_m, \beta_m}(z, t; \theta) \otimes \mathbf{c}_{\boldsymbol{\alpha}_{1:m-1}, \boldsymbol{\beta}_{1:m-1}}(z, t; \theta),$$

where we denote multi-indices $\boldsymbol{\alpha} = (\alpha_1, \dots, \alpha_m)$ and $\boldsymbol{\beta} = (\beta_1, \dots, \beta_m)$. This antisymmetric outer-product construction ensures that the resulting coefficients represent increasingly complex interactions while explicitly maintaining algebraic antisymmetry. As can be seen, neural coefficients that begin as simple first- and second-order monomials are recursively blended through successive products and Poisson brackets into progressively richer higher-order features.

## 3 POISSON-ALGEBRA MEETS PARALLEL SCAN

To systematically leverage our DAG-indexed structure $\mathcal{G}^{(\leq K)}$ as Hamiltonians generators to approximate arbitrary Hamiltonian dynamics, we first introduce the notion of a truncated Lie operator. Building on this, the next definition formalizes the resulting family of Lie–transforms as a genuine Lie group, indexed by elements of the Neural–Poisson Composition DAG.

> **Proposition 3.1** (Neural–Poisson DAG Lie–Transform Family). *Fix integers $r \geq 1$ and $K \geq 0$ and recall the Neural–Poisson Composition Graph $\mathcal{G}^{(\leq K)}$ in Definition 2.3. For any element $g \in \mathcal{G}^{(\leq K)}$ let $\mathcal{L}_{g, \leq r} : P_{\leq r} \to P_{\leq r}$ denote the truncated Lie operator $\mathcal{L}_{g, \leq r} f := \pi_{\leq r} \{f, g\}$. Then the family of DAG-indexed Lie transforms $\mathscr{T}_{r,K}$ is a **Lie group**[a]:*
>
> $$\mathscr{T}_{r,K} := \left\{ \Phi_{\epsilon, g} := \exp(\epsilon \mathcal{L}_{g, \leq r}) = \sum_{j=0}^{r} \frac{\epsilon^j}{j!} \mathcal{L}_{g, \leq r}^j \; \middle| \; g \in \mathcal{G}^{(\leq K)} \right\}. \tag{5}$$
>
> ---
> [a] *Lie group* is, by definition, a differentiable manifold endowed with a smooth binary operation, the composition $\circ$ of canonical flows together with a smooth inversion map.

**Why do we need Lie group structure?** Intuitively, our method does not parallelize over the *states* $z$ themselves, but over the *small updates* that move the system forward in time. To use a parallel prefix scan, these updates must be combined by a single binary operation that is (i) associative, (ii) stays inside the same class of updates after every composition, and (iii) can be described with finitely many parameters. A Lie group of Hamiltonian flows gives exactly this: each truncated Lie operator $\mathcal{L}_{g, \leq r}$ represents an infinitesimal Hamiltonian update, its exponential $\Phi_{\epsilon, g} = \exp(\epsilon \mathcal{L}_{g, \leq r})$ is a finite time flow, and all such flows live in the finite dimensional group $\mathscr{T}_{r,K}$, Figure 1 (mid). Inside this group, composition of flows is automatically associative and closed, so a long chain of small steps can be safely reorganized into a tree structure and evaluated by a prefix scan. If we instead tried to compose completely arbitrary smooth Hamiltonians, the space of possible updates would explode to infinite dimension and there would be no finite, well behaved group law to scan over, making the kind of parallel composition we use essentially impossible.

To make this discussion concrete, we now spell out how a truncated polynomial Hamiltonian generator $g$ gives rise to the Lie operators and flow maps that our scan composes. Let $z = (q, p) \in \mathbb{R}^{2n}$ be a phase-space point and fix a small time increment $g$ be an arbitrary Hamiltonian generators. For every index $j$ the truncated Lie operator $\mathcal{L}_{g_j, \leq r} f = \pi_{\leq r} \{f, g\}$ defines a Hamiltonian vector field $X_{g_j, \leq r} = (\partial_p g, -\partial_q g)$ that is polynomial of degree at most $r$. Exponential mapping turns this vector field into the exact time $\epsilon$ flow $\Phi_{\epsilon, g} = \exp(\epsilon L_{g, \leq r}) \in \mathscr{T}_{r,K}$, so acting on the coordinates induces the Taylor representation:

$$\hat{z} := \Phi_{\epsilon, g} z = z + \epsilon X_{g, \leq r}(z) + \frac{\epsilon^2}{2} X_{g, \leq r}^2(z) + \dots + \frac{\epsilon^r}{r!} X_{g, \leq r}^r(z), \tag{6}$$

where the series terminates after the $r$-th Lie derivative terms due to the nilpotency. Here, applying $\Phi_{\epsilon,g}$ to the phase-space point $\hat{z}$ means *"flowing"* $z$ along the Hamiltonian vector field $X_{g_j, \leq r}$ for a duration $\epsilon$. Fig 1 *(mid)* schematically depicts the incremental transition of a state $z$ to its next state $\hat{z}$.

**Neural–Poisson Lie Group Walk**. In further implementation, we define an *ordered Hamiltonian dictionary* $\mathbf{g} = [g_1, \ldots, g_m, \ldots, g_M]$ as an ordered sequence of Hamiltonian generators, *i.e.*, Fig 1 *(left)*, where each individual generator $g_m$ is explicitly constructed as a sum of $|\mathcal{G}^{(\leq K)}|/M$ elements randomly selected from the complete DAG $\mathcal{G}^{(\leq K)}$. Each of these generators induces a corresponding small-time symplectomorphism $\Phi_{\epsilon,g_m}$ representing a tiny symplectic update that incrementally captures the Hamiltonian evolution over finite time horizons. Formally, this composition introduces the *Neural–Poisson Lie group walk*, which interprets the ordered application of each small-time flow $\Phi_{\epsilon,g_m \in \mathbf{g}}$ as a discrete walk on Lie group $\mathscr{T}_{r,K}$:

**Definition 3.2** (Neural–Poisson Lie Group Walk). Fix integers $r \geq 1, K \geq 0$ and consider the Lie group $\mathscr{T}_{r,K}$ from Definition 3.1. Let $\mathbf{g}$ be an ordered dictionary of Hamiltonian generators. Then, we define the *Neural–Poisson Lie Group walk* $\mathcal{W}_{\mathbf{g}} : \{0, 1, \ldots, M\} \to \mathscr{T}_{r,K}$ recursively by

$$\mathcal{W}_{\mathbf{g}}(0) := \mathbf{Id}, \qquad \mathcal{W}_{\mathbf{g}}(m) := \Phi_{\epsilon,g_m} \circ \mathcal{W}_{\mathbf{g}}(m-1), \quad 1 \leq m \leq M.$$

For the initial state $z_0 = (q_0, p_0)$ and ordered dictionary $\mathbf{g}$, the discrete time-evolution is then realized by successively applying the exact time $\epsilon$ flows generated by each $g_j$ with Lie-group walk as follows:

$$z_0 \xrightarrow{\Phi_{\epsilon,g_1}} z_1 \xrightarrow{\Phi_{\epsilon,g_2}} z_2 \longrightarrow \cdots \longrightarrow z_{M-1} \xrightarrow{\Phi_{\epsilon,g_M}} z_M$$

$$\underbrace{\phantom{z_0 \xrightarrow{\Phi_{\epsilon,g_1}} z_1 \xrightarrow{\Phi_{\epsilon,g_2}} z_2 \longrightarrow \cdots \longrightarrow z_{M-1} \xrightarrow{\Phi_{\epsilon,g_M}} z_M}}_{\mathcal{W}_{\mathbf{g}}(M)}$$

$$\begin{aligned} z_m &= \Phi_{\epsilon,g_m} z_{m-1}, \\ z_M &= \left(\Phi_{\epsilon,g_M} \circ \cdots \circ \Phi_{\epsilon,g_1}\right) z_0, \\ z_M &= \mathcal{W}_{\mathbf{g}}(M) z_0 = (q_M, p_M). \end{aligned}$$

As each factor $\Phi_{\epsilon,g_m} = \exp(\epsilon L_{g_m, \leq r})$ represents an exact $\epsilon$-time evolution generated by a polynomial Hamiltonian $g_m$, their composition $\mathcal{W}_{\mathbf{g}}$ can be viewed intuitively as stitching together multiple small, physically consistent steps to form a complete trajectory. This approach effectively discretizes continuous-time Hamiltonian dynamics into manageable segments, producing a physically coherent sequence of states $(\mathbf{z}_1, \ldots, \mathbf{z}_M)$. Now, we introduce an important associativity binary operation on the Lie group structure in Definition 3.1 for further discussion.

$$\textbf{(Lie Group Associativity):} \quad \left(\Phi_{\epsilon,g_m} \circ \Phi_{\epsilon,g_{m-1}}\right) \circ \Phi_{\epsilon,g_{m-2}} = \Phi_{\epsilon,g_m} \circ \left(\Phi_{\epsilon,g_{m-1}} \circ \Phi_{\epsilon,g_{m-2}}\right). \quad (7)$$

> **Remark.** This associativity in our framework arises because we confine the hypothesis space $\mathcal{H}$ to *finite-degree polynomials*, forcing all Hamiltonian generators and their combinations to remain closed in a finite-dimensional algebra. By contrast, arbitrary black-box models such as conventional HNNs lack this structure, and their compositions do not guarantee associativity.

The Lie group structure defined above emerges naturally from the composition operation $\circ$ acting directly on elementary symplectic flow maps $\Phi_{\epsilon,g_m}$. This perspective recasts the original dynamical evolution into a concise geometric-algebraic framework. Within this setting, the associativity property captured by equation 7 is the critical condition enabling *physically-consistent parallel scanning*. This ensures that any rearrangement of flow-map compositions leaves the symplectic structure, energy, and all invariants strictly unchanged.

**Poisson-Algebraic Parallel Scan.** We now describe how to efficiently parallelize the computation of the trajectory via a Poisson-algebraic prefix-scan. Let $U_m := \Phi_{\epsilon,g_m}$ denote the $m$-th Lie-group element in the walk $\mathcal{W}_{\mathbf{g}}(m)$ with $m$-th member $g_m$ in the ordered dictionary $\mathbf{g}$. Then, a *parallel prefix-scan* (Blelloch, 1990) seeks the inclusive prefixes within $O(\log M)$ parallel depth with

$$\mathcal{W}_{\mathbf{g}}(m) := U_m \circ U_{m-1} \circ \cdots \circ U_1, \qquad m = 1, \ldots, M.$$

While Lie-group composition is associative by equation 7, we can evaluate all walks simultaneously by constructing a balanced binary tree of *temporary composites* as follows:

$$V_m^{(0)} := U_m, \quad V_m^{(1)} := U_{2m} \circ U_{2m-1}, \quad V_m^{(\ell)} := V_{2m}^{(\ell-1)} \circ V_{2m-1}^{(\ell-1)}, \quad \ell = 2, \ldots, \lceil \log_2 M \rceil,$$

where $\ell$ indexes the depth level of the balanced binary tree used in parallel scanning, $V_m^{(\ell)}$ represents the combined Lie flow of the contiguous block $U_{(2m-1)2^{\ell-1}+1}, \ldots, U_{2m2^{\ell-1}}$. Recursively evaluating

these composites up to $\ell = \lceil \log_2 M \rceil$ yields every prefix in parallel. In following theoretical result, we formally demonstrate that our Lie-group walk, together with its parallel prefix-scan implementation, naturally preserves fundamental physical invariants of the dynamics.

# 4 THEORETICAL FINDINGS

In this section, we rigorously establish two key theoretical results regarding the proposed *Neural–Poisson Lie–group walk*. First, we demonstrate its capacity for *universal approximation* of Hamiltonian functions (Proposition 4.2), providing explicit control over approximation errors. Second, we derive precise *generalization bounds* (Proposition 4.3), characterizing their dependence on the algebraic complexity inherent in the model.

**Theorem 4.1** (Neural–Poisson Lie–group walk is symplectic). *For any ordered dictionary* **g***, let* $\mathscr{T}_{r,K}$ *be the inclusive prefixes of the Neural–Poisson Lie–group walk, which preserves the following three distinctive symplectic properties:*

$$\underbrace{\mathcal{W}_{\mathbf{g}}^* \omega = \omega}_{structure}, \qquad \underbrace{H\big(\mathcal{W}_{\mathbf{g}} z\big) = H(z)}_{energy}, \qquad \underbrace{\det D\mathcal{W}_{\mathbf{g}} = 1}_{volume}, \quad \forall m \leq M, \quad z \in \mathcal{M}.$$

*Consequently the discrete trajectory* $z_m = \mathcal{W}_{\mathbf{g}}(m) z_0$ *preserves the symplectic form, the Liouville measure* $\mu_\omega = \omega^{\wedge n}/n!$, *and the energy exactly at every step.*

Theorem 4.1 establishes that the proposed Neural–Poisson Lie–group walk preserves multiple fundamental symplectic invariants. This invariance guarantees that the parallel prefix–scanning procedure remains fully consistent with the underlying physical laws, ensuring that long-horizon trajectory composition retains the exact geometric constraints dictated by Hamiltonian mechanics.

**Proposition 4.2** (Informal). *Fix a finite horizon* $T > 0$ *and a compact set* $K \subset \mathbb{R}^{2n}$. *For exact Hamiltonians* $H$ *lying in the bounded class* $\mathcal{H}_{R,s} = \{H \in C^s : \|H\|_{C^s} \leq R\}$ *with* $s > 2n$, *assume our Neural-Poisson DAG can realize any polynomial of total degree* $\leq r$. *Then, we have*

$$\sup_{H \in \mathcal{H}_{R,s}} \sup_{\hat{z}_0 \in K} \left\| \mathcal{W}_{\mathbf{g}}(M)(\hat{z}_0) - \hat{z}_{\lceil T/\epsilon \rceil} \right\| \lesssim \mathcal{O}\left( r^{1-s} + \epsilon^r/r! \right).$$

This theoretical result highlights that our Neural–Poisson DAG framework serves as a highly effective Hamiltonian approximator: increasing the polynomial degree $r$ systematically reduces the model approximation error $O(r^{1-s})$, while refining the step size $\epsilon$ improves the Lie–series truncation error $O(\epsilon^r/r!)$. These complementary decay rates ensure that both richer polynomial expressivity and finer temporal resolution contribute to uniformly enhanced accuracy across the entire trajectory.

**Proposition 4.3** (Informal). *Consider the depth–$K$ Neural–Poisson Lie-walk family* $\mathcal{F}_K$, *which performs* $M = \lceil T/\epsilon \rceil$ *prefix-scan steps. Then, for confidence level* $\delta \in (0,1)$, *with probability at least* $1 - \delta$ *over an i.i.d. sample size* $n$, *the worst-case difference between train and test loss satisfies*

$$\sup_{\mathcal{W}_{\mathbf{g}} \in \mathcal{F}_K} \left| \text{Train}(\mathcal{W}_{\mathbf{g}}(M)) - \text{Test}(\mathcal{W}_{\mathbf{g}}(M)) \right| \lesssim \mathcal{O}\left( \sqrt{\frac{M \log |\mathcal{G}^{(\leq K)}|}{n}} + \sqrt{\frac{\log(1/\delta)}{n}} \right).$$

Proposition 4.2 motivates the enlargement of the polynomial order $r$ and consequently the DAG, as such an increase reduces the approximation error. Proposition 4.3, in contrast, indicates that the same expansion amplifies the factor $\sqrt{\log |\mathcal{G}^{(\leq K)}|/n}$, thereby widening the generalization gap. This interplay reflects a bias–variance trade-off, wherein enhanced algebraic expressivity must be balanced against statistical reliability.

# 5 RELATED WORK

**Hamiltonian Neural Networks.** Foundational structure-preserving approaches include Hamiltonian Neural Networks (HNNs) (Greydanus et al., 2019), which regress a scalar Hamiltonian whose canonical derivatives generate the dynamics, and SympNets (Jin et al., 2020), which directly parameterize symplectic maps through composition of symplectic layers. These classical models preserve important geometric structure, but their updates remain fundamentally sequential and operate through local residual minimization. Recent physics-informed approaches tailor training objectives to Hamiltonian systems by enforcing energy conservation or symplecticity directly in the loss or architecture. Beyond

| Integrator / Method | GPU Mem | Inference Time (s) | ADE / FDE ↓ | $\Delta E$ ↓ | $\Delta \omega$ ↓ |
|---|---|---|---|---|---|
| *(A) Hamiltonian Neural Network based Trajectory Generators* with *Energy / Symplectic Preservation* | | | | | |
| HNN + Störmer–Verlet | 1.2 | $3.47\times10^1$ | $3.12\times10^{-3}/3.26\times10^{-3}$ | $\mathbf{2.06\times10^{-3}}$ | $\mathbf{5.31\times10^{-2}}$ |
| HNN + Forest–Ruth 4th | 1.5 | $4.01\times10^1$ | $7.83\times10^{-4}/8.54\times10^{-4}$ | $\mathbf{4.04\times10^{-4}}$ | $\mathbf{2.73\times10^{-1}}$ |
| HNN + Gauss–Legendre 4th | 1.8 | $5.16\times10^1$ | $5.76\times10^{-4}/7.12\times10^{-4}$ | $\mathbf{1.15\times10^{-4}}$ | $\mathbf{9.08\times10^{-1}}$ |
| SymODEN (Zhong et al., 2020) | 2.1 | $3.87\times10^1$ | $8.64\times10^{-4}/9.93\times10^{-4}$ | $7.46\times10^{-4}$ | $2.63\times10^0$ |
| SRNN (Chen et al., 2020) | 2.5 | $4.53\times10^1$ | $9.24\times10^{-4}/9.63\times10^{-4}$ | $8.13\times10^{-4}$ | $3.13\times10^0$ |
| Nonsep-SNN (Xiong et al., 2021) | 2.9 | $3.58\times10^1$ | $7.56\times10^{-4}/8.63\times10^{-4}$ | $6.33\times10^{-4}$ | $2.23\times10^0$ |
| SympNet (Jin et al., 2020) | 2.3 | $3.21\times10^1$ | $6.98\times10^{-4}/7.85\times10^{-4}$ | $5.74\times10^{-4}$ | $2.01\times10^0$ |
| *(B) Operator-based Trajectory Generators* without *Energy / Symplectic Preservation* | | | | | |
| KoVAE (Naiman et al., 2024) | 2.3 | $3.97\times10^{-1}$ | $2.75\times10^{-4}/3.67\times10^{-4}$ | $2.31\times10^{-3}$ | $2.43\times10^1$ |
| KoNODE (Bai & Ding, 2025) | 2.7 | $5.04\times10^{-1}$ | $3.22\times10^{-4}/4.67\times10^{-4}$ | $1.98\times10^{-3}$ | $1.76\times10^1$ |
| *(C) Black-box Trajectory Generators* without *Energy / Symplectic Preservation* | | | | | |
| EqMotion (Xu et al., 2023) | 24.6 | $6.88\times10^0$ | $6.32\times10^{-5}/7.07\times10^{-5}$ | $\mathbf{1.27\times10^{-1}}$ | $\mathbf{5.23\times10^1}$ |
| GeoTDM (Han et al., 2024) | 18.5 | $1.83\times10^0$ | $4.13\times10^{-5}/5.51\times10^{-5}$ | $\mathbf{1.06\times10^{-1}}$ | $\mathbf{4.88\times10^1}$ |
| ET-SEED (Tie et al., 2025) | 27.1 | $3.25\times10^0$ | $3.96\times10^{-5}/5.62\times10^{-5}$ | $\mathbf{9.97\times10^{-2}}$ | $\mathbf{8.26\times10^1}$ |
| **Ours (PAPS)** | 4.0 | $\mathbf{3.31\times10^{-2}}$ | $\mathbf{3.28\times10^{-5}}/\mathbf{4.59\times10^{-5}}$ | $5.27\times10^{-4}$ | $4.02\times10^0$ |

Table 1: **Quantitative Comparison on Atomistic Spin Dynamics (QSD).** Each column reports inference time (seconds), GPU memory, ADE/FDE, relative energy drift $\Delta E$, and symplectic violation $\Delta \omega$ of 1000 time-steps. (A) Hamiltonian-based trajectory generators with explicit energy/symplectic preservation (B) operator-based generators without explicit preservation (C) black-box generators without preservation.

residual penalties, time-dependent symplectic neural flows parameterize Hamiltonian flow maps (Cañizares et al., 2024), and symplectic-loss training for HNNs improves long-horizon stability with a provable Hamiltonian (David & Méhats, 2023). Discrete, variational-integrator flavored models on Lie groups preserve both group structure and symplectic form in learned maps (Duruisseaux et al., 2023). Despite these advances, most methods still unroll dynamics sequentially and regularize *local* residuals or per-step updates. In contrast, our framework algebraizes the hypothesis class into a finite Poisson-polynomial basis, producing exact and associative symplectomorphisms and enabling an $\mathcal{O}(\log M)$-depth prefix-scan optimized for trajectory-level identification.

**Neural Operator for Physical Dynamics.** Operator-theoretic learning lifts nonlinear dynamics to linear evolution in an observable space. The Koopman Neural Operator (KNO) learns solution operators with mesh-free, long-horizon generalization. It has been applied to fast prediction of aerodynamic transonic buffet (Meng et al., 2024a). Invertible-embedding approaches learn observable dictionaries with guaranteed reconstruction (Meng et al., 2024b). Multiplicative-structure-preserving variants refine finite-dimensional approximations of observable algebras (Boullé et al., 2024). Probabilistic and sequence-model extensions include KoVAE, which places a Koopman prior in a VAE for generative modeling of dynamical data (Naiman et al., 2024). SKOLR approximates a structured Koopman operator via a linear-RNN factorization of lagged measurements (Zhang et al., 2025b). KoNODE (Bai & Ding, 2025) introduces a Koopman-driven neural ODE where parameters evolve over time to capture nonstationary dynamics.

**Parallelizing Hamiltonian and Dynamical Systems.** Several classical approaches have investigated parallel-in-time or scan-based acceleration of *known* dynamical systems. The symmetric parareal algorithm of (Dai et al., 2013) accelerates the numerical integration of a given Hamiltonian ODE by combining coarse and fine solvers. However, these approaches are unable to apply to learning unknown Hamiltonians, nor do they yield a learnable symplectic integrator. Similarly, (Yang et al., 2017) reformulate the recursive Newton–Euler algorithm into semigroup scan operations to parallelize rigid-body inverse and forward dynamics.

## 6 Experiments

We now present a series of experiments designed to evaluate the proposed framework across representative dynamical systems. Our goal is to test not only predictive accuracy but also the extent to which physical invariants and symplectic structures are preserved in long-horizon. Full details on numerical settings, benchmark specifications, dataset preparation, and additional statistics are deferred to Section A.6. Here we first highlight experiments on quantum spin dynamics, which serve as a canonical and physically rich testbed for assessing both accuracy and structure preservation.

**Quantum Spin Dynamics (QSD).** We consider a classical spin model widely used in magnetism and spintronics, where each spin is represented as a unit vector $\mathbf{S}_i \in \mathbb{S}^2 \subset \mathbb{R}^3$. Spins in this system interact through a Hamiltonian incorporating multiple magnetic interactions, including

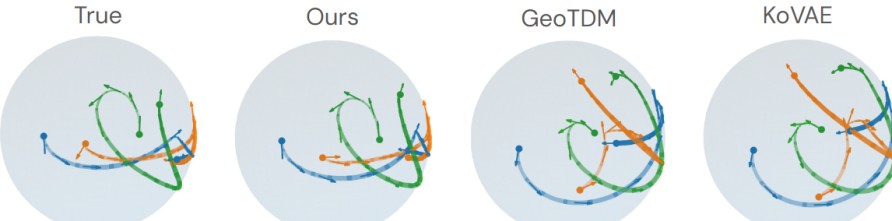

Figure 2: **Qualitative Comparison on Atomistic Spin Dynamics (QSD).** We plot three representative trajectories of 4 spins on Bloch sphere $\mathbb{S}^2$ under the QSD Hamiltonian. The proposed Poisson–Algebraic Parallel-Scan integrator (Ours) reproduces the true paths with matched curvature and phase progression, whereas GeoTDM and KoVAE accumulate trajectory-level errors and deviate from the ground-truth trajectories.

| Method | Inference (s) | ADE / FDE (Methane) ↓ | ADE / FDE (Aspirin) ↓ | $\Delta E$ ↓ / $\Delta \omega$ ↓ |
|---|---|---|---|---|
| *(A) Black-box Trajectory Generators without Energy / Symplectic Preservation* | | | | |
| GeoTDM (Han et al., 2024) | $1.91\times10^{0}$ | $4.27\times10^{-5}/5.24\times10^{-5}$ | $3.84\times10^{-5}/5.65\times10^{-5}$ | $\mathbf{1.11\times10^{-1}/5.02\times10^{1}}$ |
| ET-SEED (Tie et al., 2025) | $3.09\times10^{0}$ | $4.16\times10^{-5}/5.34\times10^{-5}$ | $4.28\times10^{-5}/5.46\times10^{-5}$ | $\mathbf{9.62\times10^{-2}/7.89\times10^{1}}$ |
| *(B) Machine Learning Force Field (MLFF) Generators based Stochastic Integrator with Energy / Symplectic Preservation* | | | | |
| Störmer-Verlet + EquiformerV2 (Liao et al., 2024) | $4.39\times10^{1}$ | $7.54\times10^{-4}/8.23\times10^{-4}$ | $8.39\times10^{-4}/9.36\times10^{-4}$ | $\mathbf{3.89\times10^{-4}/2.84\times10^{-1}}$ |
| Störmer-Verlet + EGAP (Qu et al., 2024) | $5.41\times10^{1}$ | $5.93\times10^{-4}/7.34\times10^{-4}$ | $6.21\times10^{-4}/7.64\times10^{-4}$ | $\mathbf{1.20\times10^{-4}/9.46\times10^{-1}}$ |
| **Ours (PAPS)** | $\mathbf{4.39\times10^{-2}}$ | $\mathbf{2.92\times10^{-5}/3.71\times10^{-5}}$ | $\mathbf{3.36\times10^{-5}/4.27\times10^{-5}}$ | $8.12\times10^{-6}/1.26\times10^{-4}$ |

Table 2: **Quantitative Comparison on Molecular Dynamics (MD).** We report runtime (Inference), per-molecule trajectory accuracy (ADE/FDE for aspirin and methane), and preservation of physical invariants ($\Delta E$, $\Delta \omega$). Inference time and invariant metrics were measured on aspirin dynamics. (A) Full-trajectory matching models. (B) force field based symplectic-integrator matching models

Heisenberg exchange, Dzyaloshinskii–Moriya interactions (DMI), magnetocrystalline anisotropy, Zeeman coupling, long-range dipolar interactions, and next-nearest-neighbor exchange that accounts for frustration effects (Evans et al., 2014; Abert, 2019):

$$H_{\text{true}} = -\sum_{\langle ij\rangle}[J_{ij}\mathbf{S}_i\cdot\mathbf{S}_j - \mathbf{D}_{ij}\cdot(\mathbf{S}_i\times\mathbf{S}_j)] - K\sum_i(\mathbf{S}_i\cdot\hat{\mathbf{n}})^2 - \sum_i\mathbf{h}\cdot\mathbf{S}_i + \frac{1}{2}\sum_{i\neq j}\mathbf{S}_i^\top\mathbf{T}_{ij}\mathbf{S}_j - \sum_{\langle\langle ij\rangle\rangle}J_{ij}^{(2)}\mathbf{S}_i\cdot\mathbf{S}_j.$$

Here, parameters $J_{ij}$, $\mathbf{D}_{ij}$, $K$, $\mathbf{h}$, $\mathbf{T}_{ij}$, and $J_{ij}^{(2)}$ quantify the strength and character of these respective magnetic interactions. Spin configurations evolve according to Hamiltonian dynamics derived from $H_{\text{true}}$, driven by local effective magnetic fields generated by these competing terms. All experiments were performed on systems consisting of 32 spins over trajectories spanning 1000 simulation steps. Table 1 compares our proposed method with three integrator categories: classical Hamiltonian-based integrators (A), neural operator-based approaches (B) and black-box methods (C). Methods in group (A) explicitly preserve physical invariants (highlighted in blue) but exhibit limited predictive accuracy, while methods in (B) achieve moderate accuracy with partial preservation of physical structure. Despite their superior accuracy, the approaches in (C) may violate the physical invariants marked in red. In contrast, our method strikes an optimal balance by significantly enhancing predictive accuracy while robustly preserving essential physical invariants.

**Molecular Dynamics (MD).** We conducted molecular dynamics simulations for two prototypical molecular systems, methane ($CH_4$) and aspirin ($C_9H_8O_4$), employing the OpenMM molecular dynamics simulation platform (Eastman et al., 2017). Each Interatomic interaction is parameterized using either the OpenFF and GAFF2 (Boothroyd et al., 2023; Wang et al., 2004), and the classical Hamiltonian describing the molecular systems followed the standard AMBER functional form,

$$H_{\text{true}} = \frac{1}{2}\sum_{i=1}^N\frac{\|\mathbf{p}_i\|^2}{m_i} + \sum_{(i,j)\in\mathfrak{B}}\frac{1}{2}k_{ij}(r_{ij}-r_{ij}^0)^2 + \sum_{(i,j,k)\in\mathfrak{A}}\frac{1}{2}k_{ijk}(\theta_{ijk}-\theta_{ijk}^0)^2 + \sum_\tau\sum_{n\in N_\tau}k_{\tau,n}\left[1+\cos(n\varphi_\tau-\gamma_{\tau,n})\right].$$

All parameters follow AMBER conventions: atomic momenta $\mathbf{p}_i$ with masses $m_i$; bonded terms comprising bond stretches $(i,j)\in\mathfrak{B}$ with force constants $k_{ij}$ and equilibrium lengths $r_{ij}^0$, angle bends $(i,j,k)\in\mathfrak{A}$ with $k_{ijk}$ and equilibrium angles $\theta_{ijk}^0$, and torsions $\tau$ with dihedrals $\varphi_\tau$, and nonbonded interactions given by Lennard–Jones and Coulombic terms evaluated with standard AMBER scaling. All experiments were performed on systems consisting of maximally 40 atoms over trajectories spanning 1000 simulation steps. When traditional symplectic integrators such as Störmer-Verlet are combined with existing MLFF generators in group (B) to simulate Hamiltonian dynamics, they require significantly large inference times, similar to many Hamiltonian neural network-based methods. In contrast, our PAPS achieves remarkably faster inference while maintaining superior accuracy and robust preservation of physical invariants.

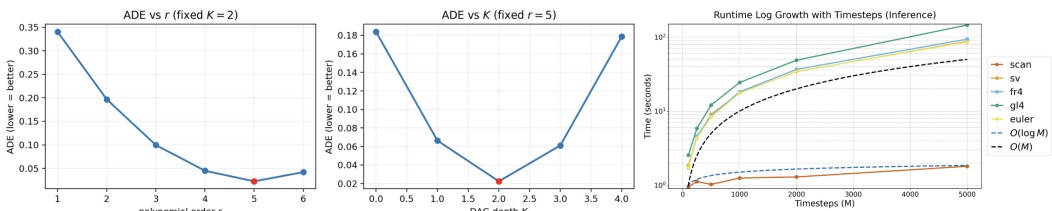

Figure 3: **Ablation study on efficiency and complexity.** Runtime scaling shows our parallel-scan integrator achieves $\mathcal{O}(\log M)$ growth compared to $\mathcal{O}(M)$ baselines, and polynomial order $r = 5$, DAG depth $K = 2$ yield the best trade-off between accuracy and generalization.

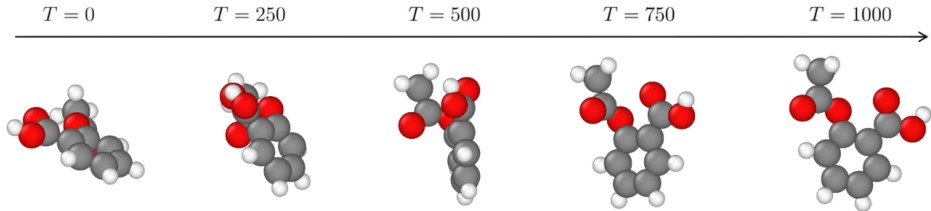

Figure 4: **Qualitative Result of Molecular dynamics trajectory of an Aspirin Molecules**. We show snapshots at selected time steps along a 1000-step NVE simulation, illustrating the rich internal motion of the molecule while its overall structure remains stable under the learned Poisson–Algebraic Parallel Scan (PAPS) integrator.

**Ablation Study.** We conducted ablation studies to assess both runtime scaling and model complexity. First, to evaluate computational efficiency, we measured the runtime of our parallel scan based integrator for a trajectory of length $M$. As shown in Figure 3 *(right)*, the runtime scales as $\mathcal{O}(\log M)$ in sharp, contrast to the $\mathcal{O}(M)$ scaling observed in baseline integrators, confirming the theoretical efficiency advantage of our Poisson-algebraic parallel scan and validating its scalability for long-horizon simulations. Second, motivated by the theoretical results in Section 4, we investigated the trade-off between polynomial order $r$, DAG depth $K$, and approximation capability. While increasing $r$ improves expressivity, the resulting growth in DAG complexity degrades generalization. Systematic experiments varying both $r$ and $K$ revealed that the best balance arises at polynomial order $r = 5$ and DAG depth $K = 2$, which achieved minimal prediction error while maintaining stable generalization performance. Together, these findings demonstrate that our method achieves both computational scalability and effective control of approximation–generalization trade-offs.

## 7 CONCLUSION

In this work, we introduced *Poisson–Algebraic Parallel Scan* (PAPS), a general symplectic-preserving integration framework that decomposes polynomial Poisson algebras into a DAG and composes coefficients via an associative prefix scan, achieving logarithmic runtime $\mathcal{O}(\log M)$. Extensive experiments across canonical and neural Hamiltonian systems demonstrate superior predictive fidelity and reduced drift in conserved quantities. Its algebra-level, operator-centric design integrates seamlessly with classical solvers, neural Hamiltonians, and generative flows, opening promising directions for future extensions to large-scale molecular simulations, quantum systems, and broader scientific modeling tasks.

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

# A APPENDIX

The appendix supplements the main text with detailed theoretical and algorithmic developments that underpin our framework. We first establish the *Symplecticity* property, demonstrating that the proposed dynamics respect the canonical two–form both in the single–exponential setting and for the full Neural–Poisson Lie–group walk. This is followed by the *Neural Poisson Algebra and Composition DAG*, where we formalize the truncated polynomial Poisson algebra, construct the associated composition DAG, and characterize the resulting Lie–transform family. In *Generalization Bound*, we provide a statistical analysis based on geometric complexity measures, leading to bounds on the generalization gap. The subsequent *Universal Approximation* subsection proves that our parameterization can approximate any target Hamiltonian flow to arbitrary accuracy within the truncated polynomial class. Finally, *Algorithmic Details* describes the implementation of the parallel–scan composition procedure and its integration into the Neural–Poisson Lie–group walk.

NOTATION TABLE

| Symbol | Meaning |
|---|---|
| $(M, \omega)$ | Symplectic manifold with symplectic 2-form $\omega$ |
| $\Pi = \omega^{-1}$ | Poisson bivector associated with $\omega$ |
| $\{f, g\}$ | Poisson bracket of $f, g \in C^\infty(M)$ |
| $\kappa : U \subset M \to \mathbb{R}^{2n}$ | Darboux chart |
| $z = (q, p)$ | Canonical coordinates $(q_1, \ldots, q_n, p_1, \ldots, p_n)$ |
| $J$ | Canonical symplectic matrix $\begin{bmatrix} 0 & I \\ -I & 0 \end{bmatrix}$ |
| $X_H$ | Hamiltonian vector field $\omega^{-1} dH = J \nabla H$ |
| $F_{X_H}^t$ | Time-$t$ Hamiltonian flow of $X_H$ |
| Poly | Polynomial ring in $(q, p)$ |
| $\text{Poly}^d$ | Homogeneous polynomials of degree $d$ |
| $F_r \text{Poly}$ | Space of polynomials of total degree $\leq r$ |
| $\pi_{\leq r}$ | Projection onto degree-$\leq r$ polynomial subspace |
| $m_\theta$ | Projected multiplication $m_\theta(f, g) = \pi_{\leq r}(fg)$ |
| $\{\cdot, \cdot\}_\theta$ | Projected Poisson bracket $\pi_{\leq r}\{f, g\}$ |
| $L_H$ | Projected Lie derivative $f \mapsto \{f, H\}_\theta$ |
| $P_{\leq r}$ | Space of degree-$\leq r$ polynomials |
| $\mathcal{G}^{(\leq K)}$ | Depth-$\leq K$ Neural–Poisson Composition DAG |
| $L_{g, \leq r}$ | Truncated Lie operator $\pi_{\leq r}\{f, g\}$ |
| $\mathfrak{t}_{r,K}$ | Lie algebra spanned by $\{L_{g, \leq r} : g \in \mathcal{G}^{(\leq K)}\}$ |
| $\mathscr{T}_{r,K}$ | Lie group of truncated flows $\exp(\epsilon L_{g, \leq r})$ |
| $\Phi_{\epsilon, g}$ | $\epsilon$-time flow generated by $g$ |
| $W_g(m)$ | Neural–Poisson Lie–group walk after $m$ steps |
| $U_m$ | $m$-th elementary Lie group update $\Phi_{\epsilon, g_m}$ |
| $V_m^{(\ell)}$ | Balanced binary-tree scan composition node |
| $C^s, H^s$ | Function spaces of order $s$ (smooth / Sobolev) |
| $\|\cdot\|_{C^s}$ | $C^s$ norm |
| $\mu_\omega$ | Liouville measure $\omega^{\wedge n}/n!$ |

Table 3: Summary of notation used throughout the paper.

## A.1 SYMPLECTICITY

In this subsection, we verify that the proposed framework preserves the fundamental symplectic structure of Hamiltonian mechanics, starting from the single–exponential case and extending to the full Neural–Poisson Lie–group walk.

**Proposition A.1.** *Let $(M, \omega)$ be a symplectic manifold with Poisson bracket $\{\cdot, \cdot\}$. Fix $H \in C^\infty(M)$ and write $X_H := \omega^{-1}dH$ for its Hamiltonian vector field with flow $F_{X_H}^t$. Assume the Lie–scan composition factorizes as a single exponential*

$$\Psi_{\Delta t} = \exp(\Delta t L_H), \quad L_H f := \{f, H\},$$

*for some $\Delta t \in I$. Then $\Psi_{\Delta t} = F_{X_H}^{\Delta t}$, hence the discrete update $z_{k+1} = \Psi_{\Delta t}(z_k)$ coincides with sampling the continuous solution of $\dot{z}_t = X_H(z_t)$ at times $t = k\Delta t$.*

*Proof.* All statements are local in time, so fix $\Delta t \in I$. Let $F_{X_H}^t$ denote the flow of $X_H$ and define, for each $t \in I$, the *Koopman pullback operator*

$$U_t : C^\infty(M) \to C^\infty(M), \qquad U_t f := f \circ F_{X_H}^t.$$

It is a strongly continuous one–parameter group of algebra homomorphisms with $U_{t+s} = U_t U_s$ and $U_0 = \mathrm{Id}$, whose infinitesimal generator is the Lie derivative along $X_H$:

$$\left.\frac{d}{dt}\right|_{t=0} U_t f = L_{X_H} f = \{f, H\} \qquad \text{for all } f \in C^\infty(M)$$

By the chain rule, for each $t \in I$ and $f \in C^\infty(M)$,

$$\frac{d}{dt}(U_t f) = \frac{d}{dt}\left(f \circ F_{X_H}^t\right) = L_{X_H} f \circ F_{X_H}^t = U_t(L_H f),$$

hence, as an operator identity,

$$\frac{d}{dt}U_t = U_t L_H, \qquad U_0 = \mathrm{Id}. \tag{8}$$

Define $V_t := \exp(t L_H) = \sum_{n \geq 0} \frac{t^n}{n!} L_H^n$. Since $L_H$ is time–independent, $V_t$ is differentiable and

$$\frac{d}{dt}V_t = \sum_{n \geq 1} \frac{t^{n-1}}{(n-1)!} L_H^n = V_t L_H, \qquad V_0 = \mathrm{Id}.$$

Thus $V_t$ solves the same operator ODE equation 8 with the same initial condition. By uniqueness for linear ODEs (verified pointwise on each $f \in C^\infty(M)$), $U_t = V_t$ for all $t \in I$. Evaluating at $t = \Delta t$ gives, for every $f \in C^\infty(M)$,

$$f \circ F_{X_H}^{\Delta t} = U_{\Delta t} f = V_{\Delta t} f = \exp(\Delta t L_H) f = f \circ \Psi_{\Delta t}.$$

The equality shows that $\Psi_{\Delta t} = F_{X_H}^{\Delta t}$. By iterating this relations, one can obtain $z_{k+1} = \Psi_{\Delta t}(z_k) = F_{X_H}^{\Delta t}(z_k)$, so the discrete trajectory is the continuous solution sampled at $t = k\Delta t$. $\square$

**Theorem A.2** (Neural–Poisson Lie–group walk is symplectic)**.** *For any ordered dictionary* **g**, *let $\mathcal{T}_{r,K}$ be the inclusive prefixes of the Neural–Poisson Lie–group walk, which preserves the following three distinctive symplectic properties:*

$$\underbrace{\mathcal{W}_{\mathbf{g}}^* \omega = \omega}_{structure}, \qquad \underbrace{H(\mathcal{W}_{\mathbf{g}} z) = H(z)}_{energy}, \qquad \underbrace{\det D\mathcal{W}_{\mathbf{g}} = 1}_{volume}, \quad \forall m \leq M, \quad z \in \mathcal{M}.$$

*Consequently the discrete trajectory $z_m = \mathcal{W}_{\mathbf{g}}(m)z_0$ preserves the symplectic form, the Liouville measure $\mu_\omega = \omega^{\wedge n}/n!$, and the energy exactly at every step.*

*Proof of Theorem 4.1.* The argument extends Proposition A.1 from a *single* exponential factor to the *ordered product* that defines the Neural–Poisson Lie–group walk. We proceed in three steps.

Let the ordered dictionary be $\mathbf{g} = (g_1, \ldots, g_M) \subset \mathrm{Poly}_\theta^{(r)}$ and let $\Delta_1, \ldots, \Delta_M > 0$ be its step sizes. For each $m$ set

$$\Phi_m := \exp(\Delta_m L_{g_m}).$$

By Proposition A.1, $\Phi_m = F_{X_{g_m}}^{\Delta_m}$, the time–$\Delta_m$ flow of the Hamiltonian vector field $X_{g_m} := \omega^{-1} dg_m$. Hence each $\Phi_m$ is an exactly symplectomorphism that satisfies

$$\Phi_m^* \omega = \omega, \qquad g_m(\Phi_m z) = g_m(z), \qquad \det D\Phi_m = 1. \tag{A.2}$$

The target Hamiltonian $H$ used for learning satisfies $\{H, g_m\} = 0$ for every generator $g_m$ by construction of the Neural–Poisson dictionary. Consequently, $H$ is constant along each individual flow:

$$\frac{d}{dt} H\left(F_{X_{g_m}}^t z\right) = \{H, g_m\}\left(F_{X_{g_m}}^t z\right) = 0 \implies H\left(\Phi_m z\right) = H(z). \tag{9}$$

Define the inclusive prefixes of the walk $\mathcal{W}_{\mathbf{g}}(m) := \Phi_m \circ \Phi_{m-1} \circ \cdots \circ \Phi_1$ for $1 \le m \le M$. As pull-backs commute with composition, one obtain $\mathcal{W}_{\mathbf{g}}(m)^* \omega = \Phi_1^*\left(\Phi_2^* \cdots (\Phi_m^* \omega)\right) = \omega$. For the estimation of volume preserving property, one can show that $\det D\mathcal{W}_{\mathbf{g}}(m) = \prod_{j=1}^m \det D\Phi_j = 1$. Energy preservation follows by iterating the relation in equation 9:

$$H\left(\mathcal{W}_{\mathbf{g}}(m)z\right) = H\left(\Phi_m \circ \cdots \circ \Phi_1 z\right) = H(z).$$

Let us define transformed coordinate $z_m := \mathcal{W}_{\mathbf{g}}(m)z_0$ for given initial state $z_0 \in \mathcal{M}$. Then for all $m \le M$, we finally have

$$\mathcal{W}_{\mathbf{g}}(m)^* \omega = \omega, \quad H(z_m) = H(z_0), \quad \det D\mathcal{W}_{\mathbf{g}}(m) = 1,$$

so the discrete path preserves the symplectic form, the Liouville measure $\mu_\omega = \omega^{\wedge n}/n!$, and the Hamiltonian energy exactly at every step. This establishes the three stated symplectic properties and completes the proof. $\square$

## A.2 Neural Poisson Algebra and Composition DAG

Before presenting the formal construction, we briefly outline the algebraic setting that underlies the proposed framework, providing the basic structures and notation used throughout the paper.

Let $(\mathcal{M}, \omega)$ be a symplectic manifold with Poisson bivector $\Pi = \omega^{-1}$ with $\{f, g\} = \Pi(df, dg)$. Given a Darboux chart $\kappa : U \subset M \to \mathbb{R}^{2n}$ for explicit coordinates, though everything below is written intrinsically in terms of $\Pi$ and pull-backs. Let us set polynomial space and their $d$-order subspace:

$$\mathrm{Poly} := \mathbb{R}[q_1, \ldots, q_n, p_1, \ldots, p_n], \qquad \mathrm{Poly}_d := \{f \in \mathrm{Poly} \mid \deg f = d\}.$$

This structure naturally admits the filtration $F^r\mathrm{Poly} := \bigoplus_{d \leq r} \mathrm{Poly}_d$ for any $r \in \mathbb{N}$. While $\deg(fg) = \deg f + \deg g$ and $\deg\{f, g\} \leq \deg f + \deg g - 2$, both the ordinary product and the Poisson bracket preserve the filtration. For the fixed truncation order $r \geq 1$ with $\pi_{\leq r} : \mathrm{Poly} \to F^r\mathrm{Poly}$ of the degree-$\leq r$ projection, note that $\pi_{\leq r}^2 = \pi_{\leq r}$. Now, we introduce the projected bilinear maps

$$m_\theta(f, g) := \pi_{\leq r}(fg), \qquad \{f, g\}_\theta := \pi_{\leq r}(\{f, g\}), \qquad f, g \in F^r\mathrm{Poly}.$$

Consequently, the following triplet $\mathrm{Poly}_\theta^{(r)} := (F^r\mathrm{Poly}, m_\theta, \{\cdot, \cdot\}_\theta)$ is called a *projected Poisson algebra*. The product $m_\theta$ is commutative, associative, and unital (with unit 1) within $F^r\mathrm{Poly}$. The projected Poisson bracket $\{\cdot, \cdot\}_\theta$ is bilinear and antisymmetric, and its Jacobiator equals $\pi_{\leq r}(\mathsf{J}(f, g, h))$, so it vanishes whenever the unprojected Jacobiator exceeds degree $r$. Moreover, the projected Leibniz rule holds: $\{f, m_\theta(g, h)\}_\theta = m_\theta(\{f, g\}_\theta, h) + m_\theta(g, \{f, h\}_\theta)$.s Note that every $H \in F^r\mathrm{Poly}$ induces a Hamiltonian derivation $L_H : f \longmapsto \{f, H\}_\theta$ and $H \mapsto L_H$ is a Lie representation: $[L_{H_1}, L_{H_2}] = L_{\{H_1, H_2\}_\theta}$. This shows that the following space forms a finite-dimensional Lie algebra.

$$\mathfrak{g}_\theta^{(r)} := \{L_H \mid H \in F^r\mathrm{Poly}\}$$

Exponentiating $L_H$ yields a canonical transformation $\exp(\Delta t L_H)$ that coincides with the time-$\Delta t$ flow of the projected Hamiltonian vector field whenever the Lie–scan collapses to a single exponential.

We now formalize this observation by characterizing the full family of such truncated Lie–generated flows within our DAG framework.

---

**Proposition A.3** (Neural–Poisson DAG Lie–Transform Family). *Fix integers $r \geq 1$ and $K \geq 0$ and recall the Neural–Poisson Composition Graph $\mathcal{G}^{(\leq K)}$ in Definition 2.3. For any element $g \in \mathcal{G}^{(\leq K)}$ let $\mathcal{L}_{g, \leq r} : P_{\leq r} \to P_{\leq r}$ denote the truncated Lie operator $\mathcal{L}_{g, \leq r} f := \pi_{\leq r}\{f, g\}$. Then the family of DAG-indexed Lie transforms $\mathscr{T}_{r,K}$ is a **Lie group**[a]:*

$$\mathscr{T}_{r,K} := \left\{ \Phi_{\epsilon, g} := \exp(\epsilon \mathcal{L}_{g, \leq r}) = \sum_{j=0}^{r} \frac{\epsilon^j}{j!} \mathcal{L}_{g, \leq r}^j \;\middle|\; g \in \mathcal{G}^{(\leq K)} \right\}. \tag{10}$$

---

[a]*Lie group* is, by definition, a differentiable manifold endowed with a smooth binary operation, the composition $\circ$ of canonical flows together with a smooth inversion map.

---

*Proof of Proposition 3.1.* Throughout, we adopt the symbols introduced just above: $P_{\leq r}$, the projected Poisson bracket $\{\cdot, \cdot\}_\theta$, the truncated vector fields $L_{g, \leq r}$, and the finite-dimensional Lie algebra $\mathfrak{t}_{r,K} \subset \mathfrak{X}(\mathcal{M})$. By construction $\mathfrak{t}_{r,K} := \mathrm{Span}\{L_{g, \leq r} \mid g \in \mathcal{G}^{(\leq K)}\}$ is a finite linear span, hence finite-dimensional. The closure relation $[L_{g_1, \leq r}, L_{g_2, \leq r}] = L_{\{g_1, g_2\}, \leq r} \in \mathfrak{t}_{r,K}$ follows from the Poisson-Jacobi identity and the fact that $\mathcal{G}^{(\leq K)}$ is closed under the truncated Poisson bracket. Thus $\mathfrak{t}_{r,K}$ is indeed a Lie subalgebra of $\mathfrak{X}(\mathcal{M})$. While $P_{\leq r}$ is finite-dimensional, one can deduce that $(P_{\leq r}) \cong \mathfrak{gl}_N(\mathbb{R})$ with $N = \dim P_{\leq r}$. Each $L_{g, \leq r}$ acts linearly on $P_{\leq r}$; hence the inclusion $\mathfrak{t}_{r,K} \hookrightarrow \mathfrak{gl}_N(\mathbb{R})$ is a Lie-algebra monomorphism. Consequently, the mapping

$$\exp : \mathfrak{t}_{r,K} \longrightarrow \mathrm{GL}_N(\mathbb{R})$$

is well defined and smooth. By the standard Lie theory (Hilgert & Neeb, 2011, Ch.II), one can show that $\exp(\mathfrak{t}_{r,K})$ is an *embedded Lie subgroup* of $\mathrm{GL}_N(\mathbb{R})$ whose Lie algebra is precisely $\mathfrak{t}_{r,K}$. Denote this subgroup by $\mathscr{T}_{r,K}$. Because $\mathfrak{t}_{r,K}$ is finite-dimensional, $\mathscr{T}_{r,K}$ is a finite-dimensional smooth manifold; the group operations are the restrictions of matrix multiplication and inversion and are

therefore smooth. Next, let us fix $g \in \mathcal{G}^{(\leq K)}$ and $\epsilon \in \mathbb{R}$. Since each $L_{g, \leq r}$ is nilpotent of order $r+1$ ($L_{g, \leq r}^{r+1} = 0$), one can obtain that the following evaluation is member of $\mathscr{T}_{r,K}$:

$$\Phi_{\epsilon, g} := \exp(\epsilon L_{g, \leq r}) = \sum_{j=0}^{r} \frac{\epsilon^j}{j!} L_{g, \leq r}^j \in \mathscr{T}_{r,K}.$$

Thus every element displayed in equation 10 belongs to the Lie subgroup $\mathscr{T}_{r,K}$. Conversely, any $X \in \mathfrak{t}_{r,K}$ is a finite linear combination $\sum_i \alpha_i L_{g_i, \leq r}$ with $g_i \in \mathcal{G}^{(\leq K)}$. The Baker–Campbell–Hausdorff formula (Hilgert & Neeb, 2011, Ch.III) expresses $\exp X$ as a finite product of factors $\exp(\alpha_i L_{g_i, \leq r})$, each of which lies in the set on the right of equation 10, showing that set coincides with $\mathscr{T}_{r,K}$. Therefore, the family of DAG-indexed Lie transforms in equation 10 forms the finite-dimensional Lie group $\mathscr{T}_{r,K}$, completing the proof. □

**Proposition A.4** (Planar-binary-DAG correspondence, quotient form). *Let $\mathcal{G}^{(0)} = \mathcal{G}_\theta = \{g_\theta^{(1)}, \ldots, g_\theta^{(m)}\}$. For $k \geq 1$ let $\widetilde{\mathcal{G}}^{(k)}$ be the set of formal binary expressions obtained from ordered pairs $(f, g) \in \widetilde{\mathcal{G}}^{(k-1)} \times \widetilde{\mathcal{G}}^{(k-1)}$ by the two constructors*

$$(f, g) \mapsto \pi_{\leq r}(f \star_\theta g), \qquad (f, g) \mapsto \pi_{\leq r}(\{f, g\}_\theta).$$

*Let $\sim$ be the smallest congruence on $\bigcup_{h \leq k} \widetilde{\mathcal{G}}^{(h)}$ generated by the following relations: commutativity of $\star_\theta$, antisymmetry of the bracket, the Jacobi identity, and the idempotence of the truncation $\pi_{\leq r}$. Define $\mathcal{G}^{(k)} := \widetilde{\mathcal{G}}^{(k)} / \sim$. Let $\widetilde{\mathcal{T}}^{(k)}$ be the set of planar binary DAGs of height $k$ with leaves labelled by $\mathcal{G}_\theta$ and internal nodes labelled by $\star_\theta$ or $\{\cdot, \cdot\}_\theta$ with ordered children. Let $\equiv$ be the smallest congruence on $\bigcup_{h \leq k} \widetilde{\mathcal{T}}^{(h)}$ generated by the corresponding local moves: swapping the two children at a $\star_\theta$-node, swapping the two children at a bracket node together with the induced global sign flip, the Jacobi rewrite on depth two bracket patterns, and erasing duplicate truncation tags. Define $\mathcal{T}^{(k)} := \widetilde{\mathcal{T}}^{(k)} / \equiv$. Then for every $k \geq 0$ there exists a bijection*

$$\Phi_k : \mathcal{G}^{(k)} \longrightarrow \mathcal{T}^{(k)}.$$

*Consequently $\mathcal{G}^{(\leq K)} \cong \bigcup_{h=0}^{K} \mathcal{T}^{(h)}$ for every $K \geq 0$.*

*Proof.* We construct mutually inverse maps by structural recursion and verify they are well defined on the quotients. For $k = 0$ set $\Phi_0(g_\theta^{(i)})$ equal to the one-node DAG labelled by $g_\theta^{(i)}$. For $k \geq 1$ take a class $[E] \in \mathcal{G}^{(k)}$ and choose any representative $E \in \widetilde{\mathcal{G}}^{(k)}$. If $E = \pi_{\leq r}(f \star_\theta g)$, let us define the following operation:

$$\widetilde{\Phi}_k(E) := \overset{\star_\theta}{\underset{\widetilde{\Phi}_{k-1}(f) \qquad \widetilde{\Phi}_{k-1}(g)}{\vert\!\!\rule[0.3em]{6em}{0.4pt}\!\!\vert}} \in \widetilde{\mathcal{T}}^{(k)}.$$

If $E = \pi_{\leq r}(\{f, g\}_\theta)$, define analogously with a $\{\cdot, \cdot\}_\theta$ root. Put $\Phi_k([E]) := [\widetilde{\Phi}_k(E)] \in \mathcal{T}^{(k)}$. If $E \sim E'$, they differ by a finite sequence of the algebraic generators listed in the statement. Each generator maps to the corresponding local move generating $\equiv$ (commutativity, antisymmetry with sign absorption, Jacobi, truncation), hence $\widetilde{\Phi}_k(E) \equiv \widetilde{\Phi}_k(E')$ and $\Phi_k$ is well defined. Next, define $\Psi_0$ as the inverse of $\Phi_0$ on leaves. For $k \geq 1$ take a class $[T] \in \mathcal{T}^{(k)}$, pick a representative $T \in \widetilde{\mathcal{T}}^{(k)}$ and define recursively

$$\widetilde{\Psi}_k \left( \overset{\star_\theta}{\underset{T_L \qquad T_R}{\vert\!\!\rule[0.3em]{6em}{0.4pt}\!\!\vert}} \right) := \pi_{\leq r}\big(\widetilde{\Psi}_{k-1}(T_L) \star_\theta \widetilde{\Psi}_{k-1}(T_R)\big),$$

$$\widetilde{\Psi}_k \left( \overset{\{\cdot, \cdot\}_\theta}{\underset{T_L \qquad T_R}{\vert\!\!\rule[0.3em]{6em}{0.4pt}\!\!\vert}} \right) := \pi_{\leq r}\big(\{\widetilde{\Psi}_{k-1}(T_L), \widetilde{\Psi}_{k-1}(T_R)\}_\theta\big).$$

Now we put $\Psi_k([T]) := [\widetilde{\Psi}_k(T)] \in \mathcal{G}^{(k)}$. If $T \equiv T'$, they differ by a finite sequence of the local moves listed above and duplicate truncation erasures. Each move is mapped by $\widetilde{\Psi}_k$ to one of the algebraic generators of $\sim$ or to idempotence of $\pi_{\leq r}$, hence $\widetilde{\Psi}_k(T) \sim \widetilde{\Psi}_k(T')$ and $\Psi_k$ is well defined.

Next, our goal is to induct on $k$. While the base case $k = 0$ is trivial, we consider the claim for $k - 1$. For $[E] \in \mathcal{G}^{(k)}$ with representative $E = \pi_{\leq r}(f \star_\theta g)$, we have

$$\Psi_k(\Phi_k([E])) = \left[\, \pi_{\leq r}\big(\widetilde{\Psi}_{k-1}(\widetilde{\Phi}_{k-1}(f)) \star_\theta \widetilde{\Psi}_{k-1}(\widetilde{\Phi}_{k-1}(g))\big) \,\right] = \left[\, \pi_{\leq r}(f \star_\theta g) \,\right] = [E].$$

By using this relation, we induct hypothesis at depth $k - 1$. The bracket case is identical, so $\Psi_k \circ \Phi_k = \mathrm{Id}$. Conversely, for $[T] \in \mathcal{T}^{(k)}$ with root operation $\odot \in \{\star_\theta, \{\cdot, \cdot\}_\theta\}$ and children $T_L, T_R$,

$$\Phi_k(\Psi_k([T])) = \left[ \begin{array}{c} \odot \\ \overline{\rule{0pt}{1ex}\hspace{3em}} \\ \widetilde{\Phi}_{k-1}(\widetilde{\Psi}_{k-1}(T_L)) \quad \widetilde{\Phi}_{k-1}(\widetilde{\Psi}_{k-1}(T_R)) \end{array} \right] = \left[ \begin{array}{c} \odot \\ \overline{\rule{0pt}{1ex}\hspace{3em}} \\ T_L \qquad\qquad T_R \end{array} \right] = [T].$$

again by the induction hypothesis. Hence $\Phi_k$ and $\Psi_k$ are inverse bijections. Now, we take disjoint unions over $h \leq K$ preserves bijectivity, so we obtain the desired result: $\mathcal{G}^{(\leq K)} \cong \bigcup_{h=0}^{K} \mathcal{T}^{(h)}$. $\qquad\square$

## A.3 GENERALIZATION

In preparation for the proof of generalization bound, we first recall the essential definitions and properties of Hamiltonian vector fields, the polynomial truncation operator $\pi_{\leq r}$, and the ambient Lie algebra $\mathfrak{X}(\mathcal{M})$ that will be used in our subsequent argument.

**Lie Subalgebra Structure of $\mathfrak{t}_{r,K}$.** Let $\mathfrak{X}(\mathcal{M})$ denote the Lie algebra of smooth vector fields on the symplectic manifold $(\mathcal{M}, \omega)$, equipped with the standard Lie bracket $[X, Y] = X \circ Y - Y \circ X$. For any Hamiltonian function $f \in C^\infty(\mathcal{M})$, the associated Hamiltonian vector field $L_f$ is defined by $\iota_{L_f}\omega = df$. Truncating to polynomial Hamiltonians of degree at most $r$, write

$$L_{f,\leq r} := \pi_{\leq r}(L_f),$$

where $\pi_{\leq r}$ projects onto the space of polynomial vector fields of degree $\leq r$. We define the set of Lie transformation as follows:

$$\mathfrak{t}_{r,K} := \mathrm{span}\big\{ L_{g,\leq r} \mid g \in G^{(\leq K)} \big\} \subset \mathfrak{X}(\mathcal{M}).$$

Note that the Poisson bracket of any two truncated polynomials in $\mathcal{G}^{(\leq K)}$ remains in $\mathcal{G}^{(\leq K)}$, and commutators of Hamiltonian vector fields satisfy the following:

$$[L_{g_1,\leq r}, L_{g_2,\leq r}] = L_{\{g_1,g_2\},\leq r}, \quad g_1, g_2 \in \mathcal{G}^{(\leq K)},$$

Thus, it follows that $\mathfrak{t}_{r,K}$ is closed under the Lie bracket, showing that $\mathfrak{t}_{r,K}$ is a finite-dimensional Lie *subalgebra* of $\mathfrak{X}(\mathcal{M})$.

**Compatible Almost–Complex Structure.** Let $(\mathcal{M}, \omega)$ be a $2n$–dimensional symplectic manifold. An *almost–complex structure* on $\mathcal{M}$ is a bundle map $J : T\mathcal{M} \longrightarrow T\mathcal{M}$ satisfying $J^2 = -\mathrm{Id}$. We say that $J$ is *compatible* with the symplectic form $\omega$ if, for all tangent vectors $u, v \in T\mathcal{M}$,

$$\omega(Ju, Jv) = \omega(u, v), \qquad \omega(u, Ju) > 0 \quad (\text{for } u \neq 0) \tag{11}$$

In particular, these conditions guarantee that the bilinear form $g(u, v) := \omega(u, Jv)$ is symmetric and positive–definite, hence defines a Riemannian metric on $M$. Together, $(\omega, J, g)$ form an *almost–Kähler* structure, which underlies the $L^2$–inner product

$$\langle X, Y \rangle = \int_M \omega\big(X(z), JY(z)\big) d\mu_\omega(z) \quad \forall \ X, Y \in \mathfrak{X}(\mathcal{M}), \tag{12}$$

and ensures that Hamiltonian flows preserve both the symplectic form and the associated phase-space volume $\mu_\omega = \omega^n / n!$ (McDuff & Salamon, 2017; Da Silva & Da Salva, 2008).

**Left-Invariant Geometry of $\mathscr{T}_{r,K}$.** We extend the proposed inner product to a Riemannian metric on the Lie group $\mathscr{T}_{r,K}$ by left translations. For any fixed Lie element $g \in \mathscr{T}_{r,K}$ and tangent vectors $X, Y \in T_g \mathscr{T}_{r,K} \subset \mathfrak{X}(\mathcal{M})$, we can rewrite the inner product by realizing the sub-algebra structure:

$$\big\langle X_g, Y_g \big\rangle_g = \big\langle \underbrace{d(L_{g^{-1}})_g[X_g]}_{=:U}, \underbrace{d(L_{g^{-1}})_g[Y_g]}_{=:V} \big\rangle \quad \text{(by definition of the left-invariant metric)} \tag{13}$$

$$= \int_M \omega\big(U(z), JV(z)\big) d\mu_\omega(z) \quad \text{(by the global } L^2\text{–inner product on } \mathfrak{X}(M)) \tag{14}$$

$$= \int_M \omega\big(d(L_{g^{-1}})_g[X_g](z), Jd(L_{g^{-1}})_g[Y_g](z)\big) d\mu_\omega(z). \tag{15}$$

where $L_{g^{-1}} : h \mapsto g^{-1}h$ is left multiplication. By construction, this metric is *left-invariant*, since left-translation by any group element preserves the inner product on each tangent space. Denote by $d_G$ the geodesic distance induced by this Riemannian metric:

$$d_G(g, h) = \inf_{\substack{\gamma(0)=g \\ \gamma(1)=h}} \int_0^1 \big\|\dot{\gamma}(t)\big\|_{\gamma(t)} dt,$$

where the infimum is over all smooth curves $\gamma : [0, 1] \to \mathscr{T}_{r,K}$ connecting $g$ to $h$. Given definition of geodesic, the *exponential map* is simply given as

$$\exp : \mathfrak{t}_{r,K} \longrightarrow \mathscr{T}_{r,K},$$

where the map is defined by sending a Lie algebra element $X$ to the time-one point $\gamma_X(1)$ of the unique geodesic $\gamma_X$ satisfying $\gamma_X(0) = e$ and $\dot{\gamma}_X(0) = X$.

**Covering Number of Neural-Poisson Lie Walk.** Exploiting the left-invariant Riemannian structure of $\mathscr{T}_{r,K}$ and the properties of the geodesic exponential map introduced above, we now proceed to derive an upper bound on the covering number of the elementary-flow set $\mathscr{S}_h$ with respect to the distance $d_G$. Let $\mathcal{W}^\theta_{r,K,M,h}$ be the Neural–Poisson Lie–group walk class. For a maximal step size $h > 0$ define the *elementary-flow set*

$$\mathscr{S}_h := \left\{ \Phi_{\varepsilon,g} = \exp\!\big(\varepsilon L_{g,\leq r}\big) \ \Big| \ g \in G^{(\leq K)}, 0 < \varepsilon \leq h \right\} \subset \mathscr{T}_{r,K}, \tag{16}$$

so every element is the exact $\varepsilon$-time Lie–exponential generated by a depth $\leq K$ polynomial Hamiltonian, truncated at degree $r$. The left-invariant Riemannian distance $d_G$ on $\mathscr{T}_{r,K}$ restricts to a metric on $\mathscr{S}_h$. We then define the *covering number* of this elementary-flow set by

$$N(\delta, \mathscr{S}_h, d_G) := \min\!\left\{ N \in \mathbb{N} \ \Big| \ \exists \, g_1, \ldots, g_N \in \mathscr{S}_h, \quad \mathscr{S}_h \subset \bigcup_{i=1}^N B_{d_G}\big(g_i, \varepsilon\big) \right\},$$

which captures the minimal number of geodesic balls of radius $\varepsilon$ needed to cover $\mathscr{S}_h$ and thus quantifies its metric complexity at scale $\delta$. Next, for each tolerance $\delta_j$, let $\mathcal{C}_{\delta_j} \subset \mathscr{S}_h$ be any $\delta_j$-cover (or $\delta_j$-net) satisfying

$$\big|\mathcal{C}_{\delta_j}\big| = N\big(\delta_j, \mathscr{S}_h, d_G\big), \qquad \mathscr{S}_h \subset \bigcup_{\Phi \in \mathcal{C}_{\delta_j}} B_{d_G}(\Phi, \delta_j),$$

where the geodesic ball $B_{d_G}$ centered at $\Phi$ in $\mathscr{S}_h$ is defined by

$$B_{d_G}(\Phi, \delta_j) := \big\{ \Psi \in \mathscr{S}_h \mid d_G(\Phi, \Psi) < \delta_j \big\}.$$

In other words, $\mathcal{C}_{\delta_j}$ is the set of centers of open balls of radius $\delta_j$ that cover $\mathscr{S}_h$, and its cardinality exactly equals the covering number.

---

**Proposition A.5** (Compositional entropy bound for Lie–group walks). *Let $M \in \mathbb{N}$. For a sequence of non–negative tolerances $\delta_1, \ldots, \delta_M$ satisfying $\sum_{j=1}^M \delta_j \leq \varepsilon$ define the Cartesian product of step covers*

$$\mathcal{C}(\delta_1, \ldots, \delta_M) := \mathcal{C}_{\delta_M} \times \cdots \times \mathcal{C}_{\delta_1}, \quad \mathcal{C}_{\delta_j} \subset \mathscr{S}_h, \quad |\mathcal{C}_{\delta_j}| = N(\delta_j, \mathscr{S}_h, d_G).$$

*For every choice $(\widehat{\Phi}_M, \ldots, \widehat{\Phi}_1) \in \mathcal{C}(\delta_1, \ldots, \delta_M)$ form the composite $\widehat{W} := \widehat{\Phi}_M \circ \cdots \circ \widehat{\Phi}_1$. Then the collection $\{\widehat{W}\}$ is an $\varepsilon$-net of the Lie–group walk class $\mathcal{W}^\theta_{r,K,M,h} = \{\Phi_{\varepsilon_M, g_M} \circ \cdots \circ \Phi_{\varepsilon_1, g_1} \mid (\varepsilon_j, g_j)$ admissible$\}$, and the covering number satisfies the purely* factorised *bound*

$$N\big(\varepsilon, \mathcal{W}^\theta_{r,K,M,h}, d_G\big) \leq \prod_{j=1}^M N\big(\delta_j, \mathscr{S}_h, d_G\big).$$

*In particular, choosing the uniform allocation $\delta_1 = \cdots = \delta_M = \varepsilon/M$ yields*

$$N\big(\varepsilon, \mathcal{W}^\theta_{r,K,M,h}, d_G\big) \leq \Big[ N\big(\varepsilon/M, \mathscr{S}_h, d_G\big) \Big]^M.$$

---

*Proof.* Fix an admissible Lie group walk $W_{\mathbf{g}}(M) := \Phi_{\varepsilon_M, g_M} \circ \cdots \circ \Phi_{\varepsilon_1, g_1} \in \mathcal{W}^\theta_{r,K,M,h}$ and, for every index $j \in \{1, \ldots, M\}$, choose $\widehat{\Phi}_j \in \mathcal{C}_{\delta_j}$ so that $d_G\big(\Phi_{\varepsilon_j, g_j}, \widehat{\Phi}_j\big) < \delta_j$. We first define the mixed partial compositions

$$W^{(j)} := \widehat{\Phi}_M \circ \cdots \circ \widehat{\Phi}_{j+1} \circ \Phi_{\varepsilon_j, g_j} \circ \cdots \circ \Phi_{\varepsilon_1, g_1}, \qquad j = 0, \ldots, M,$$

with the convention $W^{(0)} := W^\theta_{\mathbf{g},M}$ and $W^{(M)} := \widehat{\Phi}_M \circ \cdots \circ \widehat{\Phi}_1 =: \widehat{W}$. Then $d_G(W^{(0)}, W^{(M)}) \leq \sum_{j=1}^M d_G\big(W^{(j-1)}, W^{(j)}\big)$ by the ordinary triangle inequality. It therefore suffices to bound each increment $d_G\big(W^{(j-1)}, W^{(j)}\big)$. We first observe that

$$W^{(j-1)} = \widehat{\Phi}_M \circ \cdots \circ \widehat{\Phi}_{j+1} \circ \underbrace{\Phi_{\varepsilon_j, g_j}}_{\text{varying}} \circ \Phi_{\varepsilon_{j-1}, g_{j-1}} \circ \cdots \circ \Phi_{\varepsilon_1, g_1},$$

where the following equality holds for $S := \Phi_{\varepsilon_{j-1}, g_{j-1}} \circ \cdots \circ \Phi_{\varepsilon_1, g_1}$ with index $j$,

$$W^{(j)} = \widehat{\Phi}_M \circ \cdots \circ \widehat{\Phi}_{j+1} \circ \underbrace{\widehat{\Phi}_j}_{\text{proxy}} \circ \Phi_{\varepsilon_{j-1}, g_{j-1}} \circ \cdots \circ \Phi_{\varepsilon_1, g_1} = \widehat{\Phi}_M \circ \cdots \circ \widehat{\Phi}_{j+1} \circ \underbrace{\widehat{\Phi}_j}_{\text{proxy}} \circ S,$$

This shows that the distance between partial composition at $(j-1)$-th step and $j$-th can be represented as

$$d_G\left(W^{(j-1)}, W^{(j)}\right) = d_G\left(\widehat{\Phi}_M \circ \cdots \circ \widehat{\Phi}_{j+1} \circ \Phi_{\varepsilon_j, g_j} \circ S, \widehat{\Phi}_M \circ \cdots \circ \widehat{\Phi}_{j+1} \circ \widehat{\Phi}_j \circ S\right)$$

$$= d_G\left(\Phi_{\varepsilon_j, g_j} \circ S, \widehat{\Phi}_j \circ S\right) \qquad \text{(left-invariance by prefix)}$$

$$= d_G\left(\Phi_{\varepsilon_j, g_j}, \widehat{\Phi}_j\right) \qquad \text{(left-invariance by suffix } S)$$

$$< \delta_j.$$

Summing these inequalities up to $M$-th geodesic chain, we have

$$d_G\left(W^{\theta}_{\mathbf{g}, M}, \widehat{W}\right) \le \sum_{j=1}^{M} \delta_j \le \varepsilon,$$

so every walk lies in the $\varepsilon$-ball centred at a suitable $\widehat{W}$. Counting all possible choices of the proxies $\widehat{\Phi}_j \in \mathcal{C}_{\delta_j}$ produces the asserted factorised bound on the covering number, completing the proof. $\qquad \square$

We now derive a covering number bound for the elementary flow set $\mathscr{S}_h$ under the geodesic metric $d_G$. Recall that

$$\mathscr{S}_h := \{\Phi_{\varepsilon, g} = \exp(\varepsilon L_{g, \le r}) | g \in \mathcal{G}(\le K), 0 < \varepsilon \le h\} \subset \mathcal{T}_{r, K},$$

where $\mathcal{G}(\le K)$ denotes the finite set of depth-$\le K$ polynomial Hamiltonians (with at most $D$ free parameters), and $L_{g, \le r}$ is the truncated Lie operator. The set $\mathscr{S}_h$ thus forms a compact, finite-dimensional submanifold of the Lie group $\mathcal{T}_{r, K}$, parameterized by $(\varepsilon, g)$. Next, we estimate its covering number under the geodesic distance $d_G$ induced by any left-invariant Riemannian metric.

**Proposition A.6** (Volume–packing covering bound). *Let $(M, g)$ be a compact $d$-dimensional Riemannian manifold with geodesic metric $d_g$, total volume $\mathrm{Vol}(M)$, and injectivity radius $\mathrm{Inj}(M) > 0$. For every radius $0 < \delta \le \mathrm{Inj}(M)/2$ the $\delta$-covering number satisfies*

$$N(\delta, M, d_g) \le C_M \delta^{-d}, \qquad C_M = \frac{2^d \mathrm{Vol}(M)}{c_M},$$

*where the uniform lower-volume constant is defined by*

$$c_M := \inf_{x \in M} \inf_{0 < r \le \mathrm{Inj}(M)/2} \frac{\mathrm{Vol}(B_g(x, r))}{r^d} > 0.$$

*Proof.* For each $x \in M$, define the volume-ratio function within the half of injective radius

$$f(x, r) := \frac{\mathrm{Vol}(B_g(x, r))}{r^d}, \qquad 0 < r \le \frac{\mathrm{Inj}(M)}{2}.$$

Here $\mathrm{Inj}(M)$ denotes the *injectivity radius* of the Riemannian manifold $(M, g)$, namely

$$\mathrm{Inj}(M) = \inf_{x \in M} \sup\{r > 0 : \exp_x : B_{T_x M}(0, r) \to M \text{ is a diffeomorphism}\}.$$

By restricting to $0 < r \le \mathrm{Inj}(M)/2$ in the definition of $f$, we guarantee that for every $x \in M$ the geodesic ball $B_g(x, r)$ is contained entirely in the normal-coordinate neighborhood around $x$, so that the exponential map provides a smooth Euclidean chart on $B_g(x, r)$ and the usual Euclidean volume comparison applies. Note that, in geodesic normal coordinates around $x$, the metric $g$ agrees with the Euclidean metric up to second-order terms. This fact gives the following equality:

$$\lim_{r \to 0} \frac{\mathrm{Vol}(B_g(x, r))}{\mathrm{Vol}_{\mathbb{R}^d}(B(0, r))} = 1. \tag{17}$$

Since $\mathrm{Vol}_{\mathbb{R}^d}(B(0,r)) = \omega_d r^d$, it follows that $\lim_{r \to 0} f(x,r) = \omega_d$. Hence $f$ extends continuously to $\mathcal{M} \times [0, \mathrm{Inj}(M)/2]$ by setting $f(x,0) = \omega_d$, where

$$\omega_d := \mathrm{Vol}_{\mathbb{R}^d}\big(B_{\mathbb{R}^d}(0,1)\big) = \frac{\pi^{d/2}}{\Gamma\big(\frac{d}{2}+1\big)}.$$

Because $\mathcal{M} \times [0, \mathrm{Inj}(\mathcal{M})/2]$ is compact and $\omega_d > 0$, the continuous function $f$ attains a positive minimum

$$c_{\mathcal{M}} := \min_{(x,r) \in \mathcal{M} \times [0,\mathrm{Inj}(\mathcal{M})/2]} f(x,r) > 0.$$

Therefore for all $x \in \mathcal{M}$ and $0 \leq r \leq \mathrm{Inj}(\mathcal{M})/2$,

$$\mathrm{Vol}\big(B_g(x,r)\big) \geq c_M r^d. \tag{18}$$

Next, let $\{x_1, \ldots, x_N\} \subset \mathcal{M}$ be a maximal $\delta$-separated set, so $d_g(x_i, x_j) \geq \delta$ for $i \neq j$. By maximality, these points form a $\delta$-cover of manifolds $\mathcal{M}$, hence

$$\bigcup_{i=1}^{N} B_g(x_i, \delta) = \mathcal{M},$$

and by definition $N = N(\delta, \mathcal{M}, d_g)$. To show the balls $B_g(x_i, \delta/2)$ are disjoint, suppose $z \in B_g(x_i, \frac{\delta}{2}) \cap B_g(x_j, \frac{\delta}{2})$ with $i \neq j$. Then the triangle inequality gives

$$d_g(x_i, x_j) \leq d_g(x_i, z) + d_g(z, x_j) < \tfrac{\delta}{2} + \tfrac{\delta}{2} = \delta, \tag{19}$$

contradicting separation. Thus the $N$ balls $B_g(x_i, \delta/2)$ are pairwise disjoint and all lie in $M$. Combining with equation 18 yields

$$\mathrm{Vol}(\mathcal{M}) \geq \sum_{i=1}^{N} \mathrm{Vol}\big(B_g(x_i, \tfrac{\delta}{2})\big) \geq N c_{\mathcal{M}} \left(\frac{\delta}{2}\right)^d = N \frac{c_{\mathcal{M}} \delta^d}{2^d}. \tag{20}$$

Solving equation 20 for the positive integer $N$, one obtain the desired result as claimed:

$$N := N(\delta, \mathcal{M}, d_g) \leq \frac{2^d \mathrm{Vol}(\mathcal{M})}{c_{\mathcal{M}}} \delta^{-d} = \Big(\frac{2^d \mathrm{Vol}(\mathcal{M})}{c_{\mathcal{M}}}\Big) \delta^{-d} = C_{\mathcal{M}} \delta^{-d}, \quad C_{\mathcal{M}} := \frac{2^d \mathrm{Vol}(\mathcal{M})}{c_{\mathcal{M}}}.$$

$\square$

**Proposition A.7** (Covering number of the elementary–flow Lie subgroup)**.** *Equip the Lie group $\mathcal{T}_{r,K}$ with the left–invariant Riemannian metric $d_G$. For any fixed step ceiling $h > 0$ define the compact subgroup*

$$\mathscr{S}_h = \Big\{ \Phi_{\varepsilon,g} = \exp\big(\varepsilon L_{g,\leq r}\big) \mid g \in \mathcal{G}^{(\leq K)}, 0 < \varepsilon \leq h \Big\} \subset \mathcal{T}_{r,K}.$$

*Let $d_{\mathscr{S}} := \dim \mathscr{S}_h$, let $\mathrm{Vol}(\mathscr{S}_h)$ be its total Riemannian volume, and write $\mathrm{Inj}(\mathscr{S}_h) > 0$ for its injectivity radius with respect to $d_G$. Then for every radius $0 < \delta \leq \mathrm{Inj}(\mathscr{S}_h)/2$ the $\delta$–covering number of $\mathscr{S}_h$ obeys*

$$N\big(\delta, \mathscr{S}_h, d_G\big) \leq C_{\mathscr{S}} \delta^{-d_{\mathscr{S}}}, \qquad C_{\mathscr{S}} = \frac{2^{d_{\mathscr{S}}} \mathrm{Vol}(\mathscr{S}_h)}{c_{\mathscr{S}}},$$

*where the uniform lower–volume constant is*

$$c_{\mathscr{S}} := \inf_{x \in \mathscr{S}_h} \inf_{0 < r \leq \mathrm{Inj}(\mathscr{S}_h)/2} \frac{\mathrm{Vol}\big(B_G(x,r)\big)}{r^{d_{\mathscr{S}}}} > 0,$$

*and $B_G(x,r)$ denotes the geodesic ball of radius $r$ centered at $x$ in the metric $d_G$.*

*Proof.* While $\mathscr{S}_h$ is a compact submanifold of Lie group, it is compact $d_{\mathscr{S}}$–dimensional Riemannian manifold with geodesic metric $d_G$, total volume $\mathrm{Vol}(\mathscr{S}_h)$, and injectivity radius $\mathrm{Inj}(\mathscr{S}_h)$, Proposition A.6 applies verbatim. Identifying $M$ with $\mathscr{S}_h$, $d_g$ with $d_G$, and $d$ with $d_{\mathscr{S}}$ yields the stated bound together with the explicit constant $C_{\mathscr{S}}$. $\square$

**Proposition A.8.** *Let $\mathscr{S}_h^\Theta$ be elementary-flow Lie subgroup where $c = (c_1, \ldots, c_d)$ are the continuous coefficients of the polynomial Hamiltonian $g(c)$, and $\Omega$ is the domain of allowable coefficients.*

$$\mathscr{S}_h^\Theta = \left\{ \Phi_{\varepsilon, g(c)} = \exp(\varepsilon L_{g(c), \leq r}) \,\big|\, 0 < \varepsilon \leq h, c \in \Omega \subset \mathbb{R}^d \right\}$$

*Suppose the left-invariant Riemannian metric $g^{\mathscr{T}}$ on $\mathscr{T}_{r,K}$ pulls back under the mapping*

$$\phi \colon (0, h] \times \Omega \longrightarrow \mathscr{S}_h, \quad \phi(\varepsilon, c) = \exp\big(\varepsilon L_{g(c), \leq r}\big),$$

*to a block-diagonal metric of the form*

$$\phi^* g^{\mathscr{T}} = d\varepsilon^2 \oplus g_{\text{fiber}}(c), \quad g_{\text{fiber}}(c) \equiv \Sigma_\theta \in \mathbb{R}^{d \times d} \quad \text{(constant in } c\text{)}.$$

*For some $\widehat{V}\widehat{\mathcal{J}} > 0$, the Riemannian volume of $\mathscr{S}_h^\Theta$ is proportional to the cardinality of Poisson DAG:*

$$\mathrm{Vol}\big(\mathscr{S}_h^\Theta\big) \leq h|G^{(\leq K)}|\big(\widehat{V}\widehat{\mathcal{J}}\big)$$

*Proof.* In order to compute the pull-back of the left-invariant metric onto the parameter domain $(0, h] \times \Omega$ and derive the resulting volume form on $\mathscr{S}_h$, we begin by expressing how an infinitesimal change in the coefficients $c$ affects the exponential map. Here, $\partial_v \phi$ denotes the derivative of the map $\phi(\varepsilon, c) = \exp(\varepsilon L_{g(c), \leq r})$ in the direction $v \in T_c\Omega$, capturing how a small perturbation in the Hamiltonian coefficients $c$ modifies the time-$\varepsilon$ flow. The map $\phi(\varepsilon, c) = \exp\big(\varepsilon L_{g(c), \leq r}\big)$ is a smooth bijection from $(0, h] \times \Omega$ onto $\mathscr{S}_h^\Theta$, giving a global coordinate chart $(\varepsilon, c)$. Thus, for each fixed $(\varepsilon, c)$, the differential in the $c$–direction is obtained by left-invariance:

$$\partial_v \phi = d\exp_{\varepsilon L_{g(c), \leq r}} \big[\varepsilon L_{\partial_v g(c), \leq r}\big], \quad v \in T_c\Omega.$$

Left-translate this variation back to the identity in the Lie group:

$$d\big(L_{\phi(\varepsilon, c)^{-1}}\big)\partial_v \phi = d\exp_0\big(\varepsilon L_{\partial_v g(c), \leq r}\big) = \varepsilon L_{\partial_v g(c), \leq r},$$

since $d\exp_0 = \mathrm{id}$. Note that, at the identity, the Lie-algebra inner product is simply reduced to $\langle X, Y \rangle = \int_M \omega(X, JY) d\mu_\omega$. Hence, for two tangent vectors $v, w \in T_c\Omega$ one finds

$$\langle \partial_v \phi, \partial_w \phi \rangle = \varepsilon^2 \langle L_{\partial_v g(c), \leq r}, L_{\partial_w g(c), \leq r} \rangle$$

from which the pull-back metric and volume form follow. Specifically, one obtains

$$g_{\text{fiber}}(c)[v, w] := \big\langle \varepsilon L_{\partial_v g, \leq r}, \varepsilon L_{\partial_w g, \leq r} \big\rangle_{L^2(M, \mu_\omega)} \tag{21}$$

$$\overset{(a)}{=} \varepsilon^2 \int_M \omega\big(L_{\partial_v g, \leq r}(z), JL_{\partial_w g, \leq r}(z)\big) d\mu_\omega(z) \tag{22}$$

$$\overset{(b)}{=} \varepsilon^2 \int_M \big\langle J^{-1} L_{\partial_v g, \leq r}(z), J^{-1} L_{\partial_w g, \leq r}(z) \big\rangle_{T_z M} d\mu_\omega(z) \tag{23}$$

$$\overset{(c)}{=} \varepsilon^2 \int_M \big\langle \nabla(\partial_v g)(z), \nabla(\partial_w g)(z) \big\rangle_{T_z M} d\mu_\omega(z) \tag{24}$$

$$\overset{(d)}{=} \varepsilon^2 \int_M \partial_v g(z) \partial_w g(z) \underbrace{\mathrm{div}_{\mu_\omega}\big(\nabla g(z)\big)}_{= -\partial_g \log \rho_c(z)} d\mu_\omega(z) \tag{25}$$

$$\overset{(e)}{=} \varepsilon^2 \int_M \big(\partial_v \log \rho_c\big)(z)\big(\partial_w \log \rho_c\big)(z)\rho_c(z) d\mu_\omega(z) \tag{26}$$

$$= \big(\Sigma_\theta\big)_{vw}. \tag{27}$$

In line $(a)$, we express the $L^2$ inner product of the two Hamiltonian vector fields with respect to the Liouville measure $d\mu_\omega$ via the standard compatibility relation $\langle X, Y \rangle_{L^2(M, \mu_\omega)} = \int_M \omega\big(X, JY\big) d\mu_\omega$, which inserts the almost–complex structure $J$ and contributes the overall factor $\varepsilon^2$. In the second line $(b)$, we use the identity $g(\cdot, \cdot) = \omega(\cdot, J\cdot)$, so $\omega\big(X, JY\big)$ can be replaced by $g\big(J^{-1}X, J^{-1}Y\big)$, thereby conjugating each Hamiltonian vector field with $J^{-1}$. In $(c)$, we recall that for every Hamiltonian function $f$ one has $L_f = X_f = J\nabla f$, hence $J^{-1}L_{\partial_v g, \leq r} = \nabla(\partial_v g)$ and similarly for the $w$-direction, turning the integrand into the pointwise inner product of the gradients $\nabla(\partial_v g)$ and

$\nabla(\partial_w g)$. Step $(d)$ performs an integration by parts under the Liouville measure: Liouville's theorem gives $\mathrm{div}_{\mu_\omega} X_f = 0$, from which it follows that $\mathrm{div}_{\mu_\omega} \nabla g = -\partial_g \log \rho_c$, once the Gibbs density $\rho_c := d\mu_c/d\mu_\omega = e^{-g_c}/Z(\theta)$ represented by Radon–Nikodym derivative is introduced.

By using the property of this divergence, one can convert the integrand into the product $\partial_v g \partial_w g \rho_c$. Finally, in $(e)$ we notice that $\partial_v \log \rho_c = \partial_v g$ and likewise for $w$, so the integral coincides with the $(v, w)$ entry of the Fisher information matrix $\Sigma_\theta = \int_M (\nabla_\theta \log \rho_c)(\nabla_\theta \log \rho_c)^\top \rho_c d\mu_\omega$, thereby completing the identification $g_{\mathrm{fiber}}(c)[v, w] = (\Sigma_\theta)_{vw}$.

Following the computation above, the pull-back metric takes the general block–diagonal form

$$
\phi^* g^{\mathscr{T}} = d\varepsilon^2 \oplus g_{\mathrm{fiber}}(c) = \begin{pmatrix} g_{\varepsilon\varepsilon}(c) & 0 \\ 0 & \Sigma_\theta(c) \end{pmatrix} = \begin{pmatrix} A(c) & 0 \\ 0 & \Sigma_\theta(c) \end{pmatrix},
$$

where $A : \Omega \to (0, \infty)$ encodes the longitudinal scale $A(c) = g^{\mathscr{T}}\big(L_{g(c), \leq r}, L_{g(c), \leq r}\big)$ and $\Sigma_\theta(c)$ is the fiber block. Then, the Riemannian volume form in coordinates $(\varepsilon, c)$ is

$$
d\mu_{\mathscr{S}_h} = \sqrt{\det\big(\phi^* g^{\mathscr{T}}\big)} d\varepsilon dc = \sqrt{A(c) \det \Sigma_\theta(c)} d\varepsilon dc, \quad \det\big(\phi^* g^{\mathscr{T}}\big) = A(c) \det \Sigma_\theta(c).
$$

Integrating first over $\varepsilon \in (0, h]$ and then over the coefficient domain $c \in \Omega$ gives

$$
\mathrm{Vol}\big(\mathscr{S}_h^\Theta\big) = h \int_{c \in \Omega} \sqrt{A(c) \det \Sigma_\theta(c)} dc.
$$

Let $\Psi : C \to G^{(\leq K)}, c \mapsto g(c)$ be surjective, where $C \subset \mathbb{R}^d$ is the coefficient domain. For each $g \in G^{(\leq K)}$, define the *coefficient tile*

$$
T_g = \Psi^{-1}(g) = \{c \in C \mid \Psi(c) = g\}.
$$

Let us equip $\mathbb{R}^d$ with its standard $d$-dimensional Lebesgue measure $\lambda$. Then, we simply let each tile $T_g$ carry its own volume and Jacobian factor. For each $g \in G^{(\leq K)}$ define

$$
\mathrm{Vol}_g = \lambda\big(T_g\big), \quad A_g = \det\big(d\varepsilon^2 \oplus \Sigma_\theta(c)\big), \quad \Sigma_g = \Sigma_\theta(c) \quad \forall c \in T_g,
$$

and set $\widehat{\mathcal{J}}_g = \sqrt{A_g \det \Sigma_g}$. Then, the induced volume of Lie-sub group can be induced

$$
\mathrm{Vol}\big(\mathscr{S}_h^\Theta\big) = h \sum_{g \in G^{(\leq K)}} \int_{T_g} \sqrt{A(c) \det \Sigma_\theta(c)} dc = h \sum_{g \in G^{(\leq K)}} \mathrm{Vol}_g \widehat{\mathcal{J}}_g.
$$

In particular, letting $\widehat{V} = \max_g \mathrm{Vol}_g$ and $\widehat{\mathcal{J}} = \max_g \widehat{\mathcal{J}}_g$, we obtain the bound

$$
\mathrm{Vol}\big(\mathscr{S}_h^\Theta\big) \leq h |G^{(\leq K)}| \big(\widehat{V}\widehat{\mathcal{J}}\big),
$$

so the Riemannian volume and any covering number pre-factor built from it still grows linearly in the cardinality of Poisson DAG, $|G^{(\leq K)}|$. $\qquad\square$

**Corollary A.9** (DAG–sized metric entropy of the elementary-flow set). *For every radius $0 < \delta \leq \frac{1}{2}\mathrm{Inj}(\mathscr{S}_h^\Theta)$, the $\delta$-covering number of $\mathscr{S}_h^\Theta$ in the left-invariant geodesic metric $d_G$ satisfies*

$$
N\big(\delta, \mathscr{S}_h^\Theta, d_G\big) \leq \frac{2^{d_{\mathscr{S}}} h \mathrm{Vol}_g \widehat{\mathcal{J}}}{c_{\mathscr{S}}} |G^{(\leq K)}| \delta^{-d_{\mathscr{S}}}, \quad d_{\mathscr{S}} = d + 1,
$$

*where $\mathrm{Vol}_g$ is the common Lebesgue volume of each tile $T_g$, and $\widehat{\mathcal{J}} = \sqrt{A_0 \det \Sigma_0}$ as defined above.*

*Proof.* Applying Proposition A.8 with $\mathrm{Vol}(\mathscr{S}_h^\Theta) = h |G^{(\leq K)}| \mathrm{Vol}_g \widehat{\mathcal{J}}$ and $d_{\mathscr{S}} = d + 1$ gives

$$
N(\delta, \mathscr{S}_h^\Theta, d_G) \leq \frac{2^{d_{\mathscr{S}}} h \mathrm{Vol}_g \widehat{\mathcal{J}}}{c_{\mathscr{S}}} |G^{(\leq K)}| \delta^{-d_{\mathscr{S}}}.
$$

Next, we insert this result into the compositional inequality of Proposition A.6 with uniform allocation $\delta_j = \varepsilon/M$ to have

$$N\big(\varepsilon, \mathcal{W}^{\theta}_{r,K,M,h}, d_G\big) \leq \left[\frac{2^{d_{\mathscr{S}}} h V_0 \widehat{\mathcal{J}}}{c_{\mathscr{S}}}|G^{(\leq K)}|\left(\frac{M}{\varepsilon}\right)^{d_{\mathscr{S}}}\right]^M,$$

which matches the informal scaling discussed above. $\qquad\square$

---

**Proposition A.10** (Uniform trajectory-risk deviation). *Let $\mathcal{M} \subset \mathbb{R}^{2n}$ be compact with Euclidean diameter $D_2 := \sup_{x,y\in\mathcal{M}} \|x-y\|_2$. Fix integers $K \geq 0$ and $M = \lceil T/\epsilon \rceil$ for some $\epsilon > 0$, and let $\mathcal{F}_K = \mathcal{W}^{\theta}_{r(K),K,M,h_{\max}} \subset \mathscr{T}_{r(K),K}$. For $W \in \mathcal{F}_K$ define the squared Euclidean trajectory loss*

$$\ell(W; z_0, z) := \|W(z_0) - z\|_2^2, \quad L(W) := \mathbb{E}[\ell(W; Z_0, Z)], \quad \widehat{L}_n(W) := \frac{1}{n}\sum_{i=1}^n \ell\big(W; Z_{0,i}, Z_i\big).$$

*Let $C_1 := \frac{2^{d_{\mathscr{S}}} h \mathrm{Vol}_g \widehat{\mathcal{J}}}{c_{\mathscr{S}}} > 0$ be the universal covering constant from Corollary A.9 for the elementary flow manifold $\mathscr{S}_{h_{\max}}$. Then, for every $\delta \in (0,1)$, with probability at least $1 - \delta$ over the i.i.d. sample $\{(Z_{0,i}, Z_i)\}_{i=1}^n$,*

$$\sup_{W\in\mathcal{F}_K} \big|L(W) - \widehat{L}_n(W)\big| \leq \frac{8D_2^2}{\sqrt{n}}\sqrt{M\Big[\log\big(C_1|\mathcal{G}^{(\leq K)}|\big) + d_{\mathscr{S}} \log\Big(\frac{2D_2 Mn}{D_2^2}\Big)\Big]} + D_2^2\sqrt{\frac{2\log(2/\delta)}{n}}.$$

---

*Proof of Theorem 4.3.* We first relate the Euclidean trajectory–matching loss to the sup–metric on the walk class. Let $D_2 := \mathrm{diam}_2(\mathcal{M})$ and define $d_\infty^{(2)}(W, W') = \sup_{x\in\mathcal{M}} \|W(x) - W'(x)\|_2$. For $\ell(W; z_0, z) := \|W(z_0) - z\|_2^2$ we have, by the polarization identity and the triangle inequality,

$$0 \leq \ell(W; z_0, z) \leq D_2^2, \qquad |\ell(W; z_0, z) - \ell(W'; z_0, z)| \leq 2D_2\|W(z_0) - W'(z_0)\|_2,$$

so $\ell(\cdot; z_0, z)$ is $L = 2D_2$–Lipschitz w.r.t. $d_\infty^{(2)}$. Consequently (standard Lipschitz pullback of covers),

$$\log N\big(\varepsilon, \mathcal{L}_K, \|\cdot\|_\infty\big) \leq \log N\left(\tfrac{\varepsilon}{L}, \mathcal{F}_K, d_\infty^{(2)}\right), \tag{28}$$

where $\mathcal{F}_K = \mathcal{W}^{\theta}_{r(K),K,M,h_{\max}}$ and $\mathcal{L}_K = \{\ell(W; \cdot, \cdot) : W \in \mathcal{F}_K\}$. Next, we compare the output sup–metric to the left–invariant geodesic metric on the Lie group of transforms. Equip $\mathscr{T}_{r,K}$ with the left–invariant Riemannian structure induced from the $L^2$ inner product on vector fields in equation 12). For any smooth curve $\gamma : [0, 1] \to \mathscr{T}_{r,K}$ joining $W$ to $W'$ with velocity $X_\gamma(t) \in \mathfrak{t}_{r,K}$, the image curve $t \mapsto \gamma(t)(z)$ in the Euclidean output space has speed $\|X_\gamma(t)(z)\|_2 \leq \|X_\gamma(t)\|_{L^2(M,\mu_\omega)}$ by Cauchy–Schwarz. Hence $\|W(z) - W'(z)\|_2 \leq L_G(\gamma)$ for each $z$, and taking $\inf_\gamma$ then $\sup_z$ yields

$$d_\infty^{(2)}(W, W') \leq d_G(W, W'). \tag{29}$$

Combining equation 28 with equation 29 reduces the loss–class entropy to the group–metric entropy of the walk class:

$$\log N\big(\varepsilon, \mathcal{L}_K, \|\cdot\|_\infty\big) \leq \log N\left(\tfrac{\varepsilon}{L}, \mathcal{F}_K, d_G\right).$$

We now invoke the compositional covering principle for Lie–group walks. By Proposition A.5 (balanced binary prefix–scan argument), a uniform allocation $\delta_j = \varepsilon/(LM)$ gives

$$N\left(\frac{\varepsilon}{L}, \mathcal{F}_K, d_G\right) \leq \left[N\left(\frac{\varepsilon}{LM}, \mathscr{S}_h, d_G\right)\right]^M. \tag{30}$$

Thus it remains to bound the elementary–flow covering number. From the Riemannian volume–packing bound specialized to $\mathscr{S}_h$ (Proposition A.7) together with the DAG–size volume control (Corollary A.9), there exists $C_1 > 0$ (absorbing the fixed step ceiling and geometric constants) such that for every $\delta > 0$,

$$N\left(\delta, \mathscr{S}_h, d_G\right) \leq C_1|\mathcal{G}^{(\leq K)}|\left(\frac{1}{\delta}\right)^{d_{\mathscr{S}}}, \quad |\mathcal{G}^{(\leq K)}| = |G^{(\leq K)}|, \quad d_{\mathscr{S}} = d + 1. \tag{31}$$

Putting equation 30 and equation 31 together and taking logarithms yields the explicit entropy control used below:

$$\log N\big(\varepsilon, \mathcal{L}_K, \|\cdot\|_\infty\big) \leq M\left[\log\big(C_1|\mathcal{G}^{(\leq K)}|\big) + d_{\mathscr{S}}\log\Big(\tfrac{LM}{\varepsilon}\Big)\right], \qquad L = 2D_2. \tag{32}$$

To convert equation 32 into a uniform deviation bound we apply the truncated Dudley inequality for empirical Rademacher complexity:

$$\mathfrak{R}_n(\mathcal{L}_K) \leq \inf_{\alpha>0}\left\{4\alpha + \frac{12}{\sqrt{n}}\int_\alpha^{D_2^2}\sqrt{\log N\big(\varepsilon, \mathcal{L}_K, \|\cdot\|_\infty\big)}d\varepsilon\right\}. \tag{33}$$

Since the right–hand side of equation 32 is decreasing in $\varepsilon$, write

$$\phi(\varepsilon) := \sqrt{M\left[\log\big(C_1|\mathcal{G}^{(\leq K)}|\big) + d_{\mathscr{S}}\log\Big(\tfrac{LM}{\varepsilon}\Big)\right]},$$

and observe $\phi'(\varepsilon) = -(\sqrt{M}d_{\mathscr{S}})/(2\varepsilon\phi(\varepsilon)) < 0$. Therefore, for any $\alpha \in (0, D_2^2)$,

$$\int_\alpha^{D_2^2}\sqrt{\log N(\varepsilon, \mathcal{L}_K)}d\varepsilon \leq \int_\alpha^{D_2^2}\phi(\varepsilon)d\varepsilon \leq (D_2^2 - \alpha)\phi(\alpha). \tag{34}$$

Substituting equation 34 into equation 33 we arrive at the one–variable optimization

$$\mathfrak{R}_n(\mathcal{L}_K) \leq \inf_{\alpha\in(0,D_2^2)}\left\{4\alpha + \frac{12}{\sqrt{n}}(D_2^2 - \alpha)\sqrt{M\left[\log\big(C_1|\mathcal{G}^{(\leq K)}|\big) + d_{\mathscr{S}}\log\Big(\tfrac{LM}{\alpha}\Big)\right]}\right\}.$$

Set $A := \log(C_1|\mathcal{G}^{(\leq K)}|)$ and $S(\alpha) := \sqrt{M[A + d_{\mathscr{S}}\log(LM/\alpha)]}$. A direct computation gives $S'(\alpha) = -\frac{Md_{\mathscr{S}}}{2\alpha S(\alpha)} < 0$ and $S''(\alpha) > 0$, hence the objective $f(\alpha) := 4\alpha + \frac{12}{\sqrt{n}}(D_2^2 - \alpha)S(\alpha)$ is strictly convex on $(0, D_2^2)$ and admits a unique minimizer $\alpha^\star$ characterized by

$$0 = f'(\alpha^\star) = 4 + \frac{12}{\sqrt{n}}\Big[-S(\alpha^\star) + (D_2^2 - \alpha^\star)S'(\alpha^\star)\Big].$$

By multiplying the above condition by $\frac{\sqrt{n}}{12}$ and Substituting $S'(\alpha^\star) = -\frac{Md_\varphi}{2\alpha^\star S(\alpha^\star)}$, one can obtain

$$S(\alpha^\star) + \frac{Md_\varphi}{2S(\alpha^\star)} \cdot \frac{D_2^2 - \alpha^\star}{\alpha^\star} = \frac{\sqrt{n}}{3}, \tag{35}$$

which is exactly the desired equality, which has a unique solution because the right–hand side is strictly decreasing in $\alpha$. Eliminating $(D_2^2 - \alpha^\star)$ from $f(\alpha^\star)$ via equation 35 yields the balanced form

$$\mathfrak{R}_n(\mathcal{L}_K) \leq f(\alpha^\star) = \frac{8D_2^2}{\sqrt{n}}S(\alpha^\star) = \frac{8D_2^2}{\sqrt{n}}\sqrt{M\left[A + d_{\mathscr{S}}\log\Big(\tfrac{LM}{\alpha^\star}\Big)\right]}.$$

For a closed–form bound of aforementioned estimations, we choose $\widehat{\alpha} = D_2^2/n$ and evaluate $S(\widehat{\alpha}) = \sqrt{M[A + d_{\mathscr{S}}\log(LMn/D_2^2)]}$ and hence

$$\mathfrak{R}_n(\mathcal{L}_K) \leq \frac{8D_2^2}{\sqrt{n}}\sqrt{M\left[\log(C_1|\mathcal{G}^{(\leq K)}|) + d_{\mathscr{S}}\log\Big(\tfrac{LMn}{D_2^2}\Big)\right]}.$$

Finally, the standard symmetrization with Massart's inequality (Mohri et al., 2018) for bounded losses implies that, with probability at least $1 - \delta$,

$$\sup_{W\in\mathcal{F}_K}\big|L(W) - \widehat{L}_n(W)\big| \leq \frac{8D_2^2}{\sqrt{n}}\sqrt{M\left[\log(C_1|\mathcal{G}^{(\leq K)}|) + d_{\mathscr{S}}\log\Big(\tfrac{LMn}{D_2^2}\Big)\right]} + D_2^2\sqrt{\frac{2\log(2/\delta)}{n}},$$

which is the claimed uniform deviation bound in the Euclidean-loss formulation. $\square$

## A.4 UNIVERSAL APPROXIMATION

In order to rigorously establish the universal approximation capability of the proposed Neural–Poisson DAG framework, we first present a sequence of auxiliary lemmas that characterize its algebraic and approximation properties. These lemmas proved in the subsequent paragraphs provide the necessary building blocks for the final result. The main universal approximation theorem will then follow as a direct consequence of these preparatory results. For simplicity of exposition, we shall assume throughout this section that the underlying manifold is Euclidean, *i.e.*, $\mathcal{M} = \mathbb{R}^{2n}$. Nevertheless, the resulting approximation guarantees extend naturally, with only minor technical modifications, to the setting of general symplectic manifolds.

**Lemma A.11** (Every DAG node is an $r$-th Taylor polynomial). *Fix a truncation order $r \geq 1$ and let $G^{(0)} \subset \mathcal{P}_{\leq r} = \{p : \mathbb{R}^d \to \mathbb{R} \mid \deg p \leq r\}$ denote the initial generators, where $\mathcal{P}_{\leq r}$ is the space of polynomials of total degree $\leq r$. Define the depth–$K$ Neural–Poisson DAG. Then every node $g \in G^{(\leq K)}$ can be written as the (exact) $r$-th Taylor polynomial of some $C^s$ function $H$ around a base point $x_0$:*

$$g(x) = \pi_{\leq r}\left[\sum_{|\alpha| \leq r} \frac{\partial^\alpha H(x_0)}{\alpha!}(x - x_0)^\alpha\right].$$

*Proof.* We proceed by induction on the depth $k \geq 0$. For the base case, by construction $G^{(0)} \subset \mathcal{P}_{\leq r}$, so every depth-0 node has degree $\leq r$. Assume the claim holds for depth $k-1$, i.e., $G^{(k-1)} \subset \mathcal{P}_{\leq r}$. Let $f, g \in G^{(k-1)}$. Then

$$f, g \in \mathcal{P}_{\leq r} \implies fg \in \mathcal{P}_{\leq 2r}, \qquad \{f, g\} = \nabla f^\top J \nabla g \in \mathcal{P}_{\leq 2r-2},$$

since multiplication adds degrees and each gradient lowers degree by $1$ in every term. Applying the truncation $\pi_{\leq r}$ shows that the new nodes created at depth $k$ again lie in $\mathcal{P}_{\leq r}$. This completes the induction that $G^{(\leq K)} \subset \mathcal{P}_{\leq r}$. Now fix any $g \in G^{(\leq K)}$. By the above, $g$ is a polynomial with $\deg g \leq r$. Choose any base point $x_0$ and set $H := g$ (note $H \in C^\infty$). Because $g$ is a polynomial of degree $\leq r$, its order-$r$ Taylor expansion at $x_0$ recovers $g$ exactly (there is no remainder term):

$$g(x) = \sum_{|\alpha| \leq r} \frac{\partial^\alpha g(x_0)}{\alpha!}(x - x_0)^\alpha.$$

Equivalently, $g(x) = \pi_{\leq r}\left[T^r_{x_0} H(x)\right]$ with $H = g$. Finally, recall that the monomials $\{(x - x_0)^\alpha\}_{|\alpha| \leq r}$ form a basis of $\mathcal{P}_{\leq r}$, so this representation is unique in $\mathcal{P}_{\leq r}$. The lemma follows. $\square$

**Lemma A.12** (Local control of the Taylor remainder in $H^s$). *Fix a cube $Q_\ell$ of side $h$ centered at $x_\ell$. Let $H \in C^s(Q_\ell)$ and let $g_\ell$ be the $(s-1)$-st order Taylor polynomial of $H$ at $x_\ell$. Then*

$$\|H - g_\ell\|_{H^s(Q_\ell)} \leq C(d, s)\|H\|_{C^s(Q_\ell)}.$$

*Proof.* We bound the $H^s$-norm of the local Taylor remainder $H - g_\ell$ by controlling each derivative up to order $s$ on the cube $Q_\ell$. The key input is the cancellation of all derivatives up to order $s-1$ at the expansion center $x_\ell$, and the fact that the first non-vanishing term is of order $s$. This produces an $(\text{distance})^{s-|\alpha|}$ factor for $|\alpha| \leq s-1$, and the geometry of the cube then converts pointwise bounds into $L^2$ bounds with the volume factor $|Q_\ell|^{1/2} = h^{d/2}$.

For $|\alpha| \leq s - 1$, by the multivariate Taylor formula with integral remainder and the geometric bound $|x - x_\ell| \leq Ch$ (since $x$ stays inside $Q_\ell$), we have the following pointwise and $L^2$ consequences:

$$|D^\alpha(H - g_\ell)(x)| \leq C(d, s)h^{s-|\alpha|} \max_{|\gamma|=s} \|D^\gamma H\|_{L^\infty(Q_\ell)},$$

$$\|D^\alpha(H - g_\ell)\|_{L^2(Q_\ell)} \leq C(d, s)h^{s-|\alpha|+\frac{d}{2}} \|H\|_{C^s(Q_\ell)}.$$

The first line encodes the precise order-$s$ vanishing at $x_\ell$ and the uniform control of $s$-th derivatives of $H$ on $Q_\ell$; the second line is obtained by integrating the pointwise estimate and multiplying by the

square root of the volume of $Q_\ell$. For $|\alpha| = s$, the $(s-1)$-degree Taylor polynomial has zero $s$-th derivatives, so the remainder's $s$-th derivative is just the $s$-th derivative of $H$ itself. This gives the direct estimate

$$\|D^\alpha(H - g_\ell)\|_{L^2(Q_\ell)} = \|D^\alpha H\|_{L^2(Q_\ell)} \leq h^{\frac{d}{2}}\|D^\alpha H\|_{L^\infty(Q_\ell)} \leq h^{\frac{d}{2}}\|H\|_{C^s(Q_\ell)}.$$

Here we again only use the $L^\infty$-control of $s$-th derivatives on $Q_\ell$ and the volume factor $h^{d/2}$. Finally, summing the squares over all multi-indices $|\alpha| \leq s$ yields the $H^s$-bound. All constants that arise from the finite number of multi-indices and from the cube geometry depend only on $(d, s)$ and not on the particular cube index $\ell$ or on $h$ (beyond the explicit powers already shown). Using $h \leq 1$ to drop harmless $h^{d/2} \leq 1$, the proof is complete by showing

$$\|H - g_\ell\|_{H^s(Q_\ell)}^2 = \sum_{|\alpha| \leq s} \|D^\alpha(H - g_\ell)\|_{L^2(Q_\ell)}^2 \leq C(d, s)h^d\|H\|_{C^s(Q_\ell)}^2,$$

$$\Rightarrow \quad \|H - g_\ell\|_{H^s(Q_\ell)} \leq C(d, s)h^{\frac{d}{2}}\|H\|_{C^s(Q_\ell)} \leq C(d, s)\|H\|_{C^s(Q_\ell)}.$$

$\square$

**Proposition A.13.** *Let $d \geq 1$ and $s > d$ be integers, and set $\Omega = [-1, 1]^d \subset \mathbb{R}^d$. For every $H \in C^s(\Omega)$ define the $C^s$–seminorm $\|H\|_{C^s} := \sum_{|\alpha| \leq s} \|\partial^\alpha H\|_{L^\infty(\Omega)}$. Then for each integer $r \geq 1$ there exists a polynomial $\mathrm{Poly}_r \in \mathcal{P}_{\leq r}(\mathbb{R}^d)$ and a constant $C(d, s) > 0$ depending only on $d$ and $s$ such that*

$$\|\nabla H - \nabla \mathrm{Poly}_r\|_{L^\infty(\Omega)} \leq C(d, s)\|H\|_{C^s}r^{1-s}. \tag{36}$$

*Proof.* Fix $r$ and partition $\Omega$ into disjoint cubes $\{Q_\ell\}$ of side length $h := 1/r$ (so $\mathrm{diam}(Q_\ell) \asymp h$). For each $Q_\ell$ choose its center $x_\ell$ and let $g_\ell$ be the $(s-1)$-st Taylor polynomial of $H$ at $x_\ell$ (as in Lemma A.11), so that $\partial^\alpha(H - g_\ell)(x_\ell) = 0$ for all $|\alpha| \leq s - 1$. Then, we define the linear functional $F_j^{(\ell)}(u) := \partial_{x_j} u(x_\ell)$ $(1 \leq j \leq d)$. Because $s - 1 > d/2$, pointwise first–derivative evaluation is a bounded linear functional on $H^s(Q_\ell)$ via Sobolev embedding, with a bound uniform in $h$ after the standard scaling to a reference cube. Moreover, $F_j^{(\ell)}$ annihilates every polynomial of degree $\leq s - 1$ by the vanishing-derivative property above. Hence the Bramble–Hilbert theorem (Bramble & Hilbert, 1970, Theorem 2) applies with $k = s$ and yields

$$|F_j^{(\ell)}(H - g_\ell)| \leq C_0(\mathrm{diam}\, Q_\ell)^s\|H - g_\ell\|_{H^s(Q_\ell)} \leq C_0 h^s\|H - g_\ell\|_{H^s(Q_\ell)}, \tag{37}$$

with a constant independent of $h$ and $\ell$. Next, since $g_\ell$ is the $(s-1)$-st Taylor polynomial at $x_\ell$, by the remainder estimate in Taylor's theorem and Sobolev embedding on a cube of size $h$ in Lemma A.12, one obtains

$$\|H - g_\ell\|_{H^s(Q_\ell)} \leq C_1(d, s)\|H\|_{C^s}. \tag{38}$$

Combining these results, we obtain the following estimation:

$$\max_{1 \leq j \leq d} |\partial_{x_j}(H - g_\ell)(x_\ell)| \leq C_2(d, s)h^s\|H\|_{C^s}.$$

Let $\{\varphi_\ell\}$ be a $C^\infty$ partition of unity subordinate to $\{Q_\ell\}$ with the standard derivative bounds $\|\partial^\beta \varphi_\ell\|_{L^\infty} \leq Ch^{-|\beta|}$. Define the *patched* approximation $\widetilde{P}_r(x) := \sum_\ell \varphi_\ell(x)g_\ell(x)$. By using Leibniz' rule and that only $O(1)$ many $\varphi_k$ overlap at a given $x$, together with the zeroth- and first-order Taylor remainders on a cube of size $h$, we get for $x \in Q_\ell$:

$$|\nabla H(x) - \nabla \widetilde{P}_r(x)| \leq \sum_k |\varphi_k(x)| \cdot |\nabla H(x) - \nabla g_k(x)| + \sum_k |\nabla \varphi_k(x)| \cdot |H(x) - g_k(x)|$$

$$\leq Ch^{s-1}\|H\|_{C^s},$$

where we used $|\nabla \varphi_k| \lesssim h^{-1}$ and the local Taylor remainders $|\nabla H - \nabla g_k| \lesssim h^{s-1}\|H\|_{C^s}$, $|H - g_k| \lesssim h^s\|H\|_{C^s}$. Taking the supremum over $\Omega$,

$$\|\nabla H - \nabla \widetilde{P}_r\|_{L^\infty(\Omega)} \leq C_3(d, s)h^{s-1}\|H\|_{C^s}. \tag{39}$$

Finally, let $\Pi_r$ be a standard Jackson/de la Vallée–Poussin projector onto $\mathcal{P}_{\leq r}$ (total degree $\leq r$), which satisfies the derivative Jackson estimate

$$\|\nabla(f - \Pi_r f)\|_{L^\infty(\Omega)} \leq C_4(d,s) r^{1-s} \|f\|_{C^s(\Omega)} \qquad \text{for all } f \in C^s(\Omega).$$

Define the actual polynomial $\text{Poly}_r := \Pi_r(\widetilde{P}_r) \in \mathcal{P}_{\leq r}$. Then

$$\|\nabla H - \nabla\text{Poly}_r\|_{L^\infty(\Omega)} \leq \|\nabla H - \nabla\widetilde{P}_r\|_{L^\infty(\Omega)} + \|\nabla(\widetilde{P}_r - \Pi_r\widetilde{P}_r)\|_{L^\infty(\Omega)}$$
$$\leq C_3 h^{s-1}\|H\|_{C^s} + C_4 r^{1-s}\|\widetilde{P}_r\|_{C^s}.$$

Since $\|\widetilde{P}_r\|_{C^s} \leq C_5(d,s)\|H\|_{C^s}$ by the local construction and the uniform overlap of the partition, and $h = 1/r$, we conclude

$$\|\nabla H - \nabla\text{Poly}_r\|_{L^\infty(\Omega)} \leq C(d,s) r^{1-s}\|H\|_{C^s},$$

which is exactly equation 36. $\qquad\square$

**Lemma A.14** (Supremum-Norm Bound via Poisson–Bracket Coefficients). *Let $M \subset \mathbb{R}^{2n}$ be a compact phase–space domain equipped with canonical coordinates $(q_1, \ldots, q_n, p_1, \ldots, p_n)$. For a Hamiltonian $g \in C^{r+1}(M)$ with $r \geq 1$, denote the associated Hamiltonian vector field by*

$$X_g = \sum_{i=1}^n (\partial_{p_i} g \partial_{q_i} - \partial_{q_i} g \partial_{p_i}), \qquad \text{ad}_g^k(\cdot) := \underbrace{\{\ldots\{\{\cdot, g\}, g\}, \ldots, g\}}_{k \text{ times}}.$$

*Given a fixed step size $\epsilon > 0$, define the local Lie–series remainder by*

$$R_\epsilon^{r+1}(z) = \frac{\epsilon^{r+1}}{r!} \int_0^1 (1-\tau)^r X_g^{r+1}(e^{\tau\epsilon X_g}(z)) d\tau, \qquad z \in M.$$

*Then there exists a constant $C(n,r) > 0$ depending only on $n$ and $r$ such that*

$$\left\|R_\epsilon^{r+1}\right\|_{L^\infty(M)} \leq \frac{\epsilon^{r+1}}{(r+1)!} C(n,r)\|g\|_{C^{r+1}(M)}.$$

**Remark.** Up to this point some auxiliary bounds were stated on $\Omega = [-1,1]^d$. The present estimate is domain-agnostic: the coefficient counting for $X_g^k z^\beta = \text{ad}_g^k(z^\beta)$ only produces a finite linear combination of partial derivatives $\{\partial^\alpha g : |\alpha| \leq k\}$ with universal combinatorial coefficients. Hence replacing $\|\cdot\|_{L^\infty(\Omega)}$ by $\|\cdot\|_{L^\infty(M)}$ leaves the bound unchanged. If one prefers a reduction: cover the compact $M$ by finitely many axis-aligned cubes $Q_\ell$ affine to $\Omega$; since our inequality uses only suprema of derivatives of $g$ (no Jacobian weights), taking $\max_\ell \|\partial^\alpha g\|_{L^\infty(Q_\ell)} = \|\partial^\alpha g\|_{L^\infty(M)}$ yields the same constants.

*Proof.* To establish the bound, we begin by examining the action of repeated Lie derivatives generated by $X_g$ on coordinate functions. Consider a single canonical coordinate $z^\beta$ ($\beta = 1, \ldots, 2n$). Applying $X_g$ once yields a linear combination of first derivatives of $g$, with coefficients $\pm 1$ determined by the canonical Poisson structure. When we apply $X_g$ repeatedly $k$ times, we generate nested Poisson brackets $X_g^k z^\beta = \text{ad}_g^k(z^\beta)$ which expand into a finite sum of mixed partial derivatives of $g$ up to order $k$, with each term accompanied by a coefficient $C_{\alpha,\beta,k} \in \{\pm 1, 0\}$ depending on the pattern of differentiations in $q_i$ and $p_i$. Formally, we may write

$$X_g^k z^\beta = \sum_{1 \leq |\alpha| \leq k} C_{\alpha,\beta,k} \partial_z^\alpha g,$$

where $\alpha = (\alpha_1, \ldots, \alpha_{2n})$ is a multi–index and $|\alpha|$ is its order. Because each application of $X_g$ differentiates once with respect to some coordinate, all resulting derivatives have order at most $k$, and the number of such terms depends only on $n$ and $k$. Therefore, for $k = r + 1$ we obtain the uniform bound

$$\left\|X_g^{r+1} z^\beta\right\|_{L^\infty(M)} \leq C(n,r) \max_{|\alpha| \leq r+1} \|\partial_z^\alpha g\|_{L^\infty(M)} \leq C(n,r)\|g\|_{C^{r+1}(M)},$$

where $C(n, r)$ absorbs the combinatorial factor counting the possible derivative patterns. Next, recall that the Hamiltonian flow $\phi_t = e^{tX_g}$ admits a Lie-series expansion for each coordinate function, and that the remainder after truncating at order $r$ is exactly given by the integral form

$$R_{\epsilon,\beta}^{r+1}(z) = \frac{\epsilon^{r+1}}{r!} \int_0^1 (1-\tau)^r \left(X_g^{r+1}z^\beta\right)\left(\phi_{\tau\epsilon}(z)\right)d\tau.$$

Since the flow $\phi_{\tau\epsilon}$ is volume–preserving and smooth on the compact set $M$, taking the $L^\infty$ norm over $z \in M$ can be done directly inside the integrand. The factor $\int_0^1 (1-\tau)^r d\tau = \frac{1}{r+1}$ follows from direct integration of the weight. Combining these facts, we find

$$\|R_{\epsilon,\beta}^{r+1}\|_{L^\infty(M)} \leq \frac{\epsilon^{r+1}}{(r+1)!}\|X_g^{r+1}z^\beta\|_{L^\infty(M)} \leq \frac{\epsilon^{r+1}}{(r+1)!}C(n,r)\|g\|_{C^{r+1}(M)}.$$

Finally, since the bound holds uniformly for each coordinate index $\beta = 1, \ldots, 2n$, the same constant $C(n, r)$ applies to the vector-valued remainder $R_\epsilon^{r+1}$ as a whole. This completes the proof of the claimed supremum-norm bound. $\qquad\square$

**Theorem A.15** (Symplectic Lie Transform Error Expansion). *Let $H \in C^s(\mathbb{R}^{2n})$ with $s > 2n$, let $X_H = J\nabla H$ be the exact Hamiltonian vector field, and let $P_r$ be an $r$-th degree polynomial approximation of $H$ with associated vector field $X_\epsilon^H := J\nabla P_r$. Define the exact flows*

$$\hat{\Phi}_{\epsilon,H}(z) := \exp(\epsilon X_H)(z), \qquad F_\epsilon^{X_\epsilon^H}(z) := \exp(\epsilon X_\epsilon^H)(z).$$

*Assume $X_H$ and $X_\epsilon^H$ are (globally) Lipschitz on $M$ with common constant $L$. Then, for all sufficiently small $\epsilon > 0$,*

$$\left\|\hat{\Phi}_{\epsilon,H} - F_\epsilon^{X_\epsilon^H}\right\|_{L^\infty} \leq C(s,n,r)\left[r^{1-s}\|H\|_{C^s}\epsilon e^{\epsilon L} + \frac{\epsilon^{r+1}}{(r+1)!}\|H\|_{C^{r+1}}\right]. \tag{40}$$

*Proof.* Let us write the Lie–Taylor expansions of both exact and approximated flows up to order $r$ with integral remainders:

$$\hat{\Phi}_{\epsilon,H}(z) = z + \sum_{k=1}^r \frac{\epsilon^k}{k!}X_H^k(z) + \hat{R}_\epsilon^{r+1}(z)$$

$$F_\epsilon^{X_\epsilon^H}(z) = z + \sum_{k=1}^r \frac{\epsilon^k}{k!}X_{P_r}^k(z) + R_\epsilon^{r+1}(z)$$

$$\hat{R}_\epsilon^{r+1}(z) = \frac{\epsilon^{r+1}}{r!} \int_0^1 (1-\tau)^r X_H^{r+1}\left(e^{\tau\epsilon X_H}(z)\right)d\tau$$

$$R_\epsilon^{r+1}(z) = \frac{\epsilon^{r+1}}{r!} \int_0^1 (1-\tau)^r X_{P_r}^{r+1}\left(e^{\tau\epsilon X_{P_r}}(z)\right)d\tau = 0$$

Taking differences and expectation with respect to $L^\infty(M)$-norm leads the following inequality:

$$\left\|\hat{\Phi}_{\epsilon,H} - F_\epsilon^{X_\epsilon^H}\right\|_{L^\infty} \leq \sum_{k=1}^r \frac{\epsilon^k}{k!}\|X_H^k - X_{P_r}^k\|_{L^\infty} + \|\hat{R}_\epsilon^{r+1}\|_{L^\infty} + \underbrace{\|R_\epsilon^{r+1}\|_{L^\infty}}_{=0}.$$

Define linear operators on vector–valued functions $F : M \to \mathbb{R}^{2n}$ by $(\mathcal{X}_V F)(z) := DF(z)V(z)$ where $V \in \{X_H, X_{P_r}\}$ and set $A := \mathcal{X}_{X_H}$, $B := \mathcal{X}_{X_{P_r}}$. If $L := \max\{\mathrm{Lip}(X_H), \mathrm{Lip}(X_{P_r})\}$, and equip operators with the norm

$$\|T\|_{\mathrm{Lip}\to\infty} := \sup_{\mathrm{Lip}(F)\leq 1} \|TF\|_{L^\infty}.$$

Then, one can easily show that the followings hold

$$\|A\|_{\mathrm{Lip}\to\infty} \leq L, \quad \|B\|_{\mathrm{Lip}\to\infty} \leq L, \quad \|A - B\|_{\mathrm{Lip}\to\infty} \leq \|X_H - X_{P_r}\|_{L^\infty},$$

since $(A - B)F = DF(X_H - X_{P_r})$ and $\|DF\|_\infty \le \mathrm{Lip}(F)$. For some constant $k \ge 1$, the telescoping identity gives $A^k - B^k = \sum_{j=0}^{k-1} A^{k-1-j}(A - B)B^j$, showing that

$$\|A^k - B^k\|_{\mathrm{Lip}\to\infty} \le \sum_{j=0}^{k-1} \|A\|_{\mathrm{Lip}\to\infty}^{k-1-j} \|A - B\|_{\mathrm{Lip}\to\infty} \|B\|_{\mathrm{Lip}\to\infty}^{j} \le kL^{k-1}\|X_H - X_{P_r}\|_{L^\infty}.$$

Finally, since $X_H^k = (A^k \mathrm{Id})$ and $X_{P_r}^k = (B^k \mathrm{Id})$ with $\mathrm{Lip}(\mathrm{Id}) = 1$, for the $k$-fold iterates, a standard telescoping/induction bound under a common Lipschitz constant $L$ gives

$$\|X_H^k - X_{P_r}^k\|_{L^\infty} = \|(A^k - B^k)\mathrm{Id}\|_{L^\infty} \le \|A^k - B^k\|_{\mathrm{Lip}\to\infty} \le kL^{k-1}\|X_H - X_{P_r}\|_{L^\infty}.$$
$$= kL^{k-1}\big\|J(\nabla H - \nabla P_r)\big\|_{L^\infty} \le C(s,n)kL^{k-1}r^{1-s}\|H\|_{C^s(M)}$$

by the gradient–Jackson inequality (Proposition A.13). Therefore,

$$\sum_{k=1}^{r} \frac{\epsilon^k}{k!} \big\|X_H^k - X_{P_r}^k\big\|_{L^\infty} \le \sum_{k=1}^{r} \frac{\epsilon^k}{k!} C(s,n)kL^{k-1}r^{1-s}\|H\|_{C^s(M)}$$

$$= C(s,n)r^{1-s}\|H\|_{C^s(M)}\epsilon \sum_{k=1}^{r} \frac{(\epsilon L)^{k-1}}{(k-1)!}.$$

$$\le C(s,n)r^{1-s}\|H\|_{C^s(M)}\epsilon e^{\epsilon L}$$

The last inequality follows by using $\sum_{j=0}^{r-1} \frac{(\epsilon L)^j}{j!} \le e^{\epsilon L}$, For the remainders, we apply Lemme A.14 to yield

$$\big\|\hat{R}_\epsilon^{r+1}\big\|_{L^\infty} \le \frac{\epsilon^{r+1}}{(r+1)!}C(n,r)\|H\|_{C^{r+1}(M)},$$

Combining results above and setting $C(s,n,r) = \max\{C(s,n), C(n,r)\}$ establishes the desired result in equation 40. $\qquad\square$

---

**Proposition A.16** (Global error for an $M$–step Lie–group walk)**.** *Fix a step size $\epsilon > 0$ and set $T := M\epsilon$. Let $H \in C^s(\mathbb{R}^{2n})$ with $s > 2n$, and write $X_H = J\nabla H$. Let $P_r$ be any degree–$r$ polynomial approximant of $H$ and set $X_{P_r} = J\nabla P_r$. Assume $X_H$ and $X_{P_r}$ are globally Lipschitz on the relevant compact set with a common bound $L_* > 0$. Let the $M$–step compositions be $\mathcal{W}_M := G_\epsilon^M$ and $F_T := \exp(TX_H)$. If the one–step error satisfies the bound from Theorem A.15,*

$$\|G_\epsilon - F_\epsilon\|_{L^\infty} \le \underbrace{C_1(s,n,r)r^{1-s}\|H\|_{C^s}\,\epsilon e^{\epsilon L_*}}_{:=\mathfrak{A}} + \underbrace{\frac{C_2(n,r)}{(r+1)!}\|H\|_{C^{r+1}}\,\epsilon^{r+1}}_{:=\mathfrak{B}} =: \delta_\epsilon,$$

*then the final–time (global) error obeys*

$$\|\mathcal{W}_{\mathbf{g}}(M) - F_T\|_{L^\infty} \le \frac{e^{L_*T} - 1}{L_*}\left[C_1(s,n,r)r^{1-s}\|H\|_{C^s}e^{\epsilon L_*} + \frac{C_2(n,r)}{(r+1)!}\|H\|_{C^{r+1}}\epsilon^r\right]. \quad (41)$$

*Proof.* We first introduce the notation

$$\mathrm{error}_k := \|G_\epsilon^k - F_{k\epsilon}\|_{L^\infty}, \quad k = 0, 1, \dots, M,$$

so that $\mathrm{error}_k$ measures the accumulated global error after $k$ discrete steps of size $\epsilon$ compared to the exact Hamiltonian flow over time $t = k\epsilon$. Initially $\mathrm{error}_0 = 0$, since both methods start from the same initial condition and $G_\epsilon^0 = F_0 = \mathrm{Id}$. Consider the error after $(k+1)$ steps. By adding and subtracting the mixed term $G_\epsilon^k \circ F_\epsilon$ and applying the triangle inequality, we have

$$\mathrm{error}_{k+1} = \big\|G_\epsilon^{k+1} - F_{(k+1)\epsilon}\big\|_\infty = \big\|G_\epsilon^k \circ G_\epsilon - F_{k\epsilon} \circ F_\epsilon\big\|_\infty$$
$$\le \big\|G_\epsilon^k \circ G_\epsilon - G_\epsilon^k \circ F_\epsilon\big\|_\infty + \big\|G_\epsilon^k \circ F_\epsilon - F_{k\epsilon} \circ F_\epsilon\big\|_\infty.$$

The first term measures the effect of replacing $F_\epsilon$ by $G_\epsilon$ in the last step while keeping the first $k$ steps fixed; the second term propagates the error already accumulated in the first $k$ steps forward by one

exact step of $F_\epsilon$. For the second term, note that postcomposition with $F_\epsilon$ does not increase the $L^\infty$ norm of the difference, because $F_\epsilon$ is a diffeomorphism on the compact set $M$. More precisely,

$$\left\|G_\epsilon^k \circ F_\epsilon - F_{k\epsilon} \circ F_\epsilon\right\|_\infty = \sup_z\left\|(G_\epsilon^k - F_{k\epsilon})(F_\epsilon(z))\right\| \leq \sup_y\left\|G_\epsilon^k(y) - F_{k\epsilon}(y)\right\| = \text{error}_k.$$

For the first term, we use the Lipschitz stability of the $k$–fold composition. Since both $X_{P_r}$ and $X_H$ are $L_*$–Lipschitz, their flows over time $t$ have Lipschitz constant bounded by $e^{L_* t}$. In particular, $\|G_\epsilon\|_{\text{Lip}}, \|F_\epsilon\|_{\text{Lip}} \leq e^{L_* \epsilon}$, and consequently $\|G_\epsilon^k\|_{\text{Lip}} \leq e^{L_* k\epsilon}$. Thus

$$\left\|G_\epsilon^k \circ G_\epsilon - G_\epsilon^k \circ F_\epsilon\right\|_\infty \leq \|G_\epsilon^k\|_{\text{Lip}}\|G_\epsilon - F_\epsilon\|_\infty \leq e^{L_* k\epsilon}\delta_\epsilon.$$

Combining these two estimates, we arrive at the recurrence relation

$$\text{error}_{k+1} \leq e^{L_* k\epsilon}\delta_\epsilon + \text{error}_k, \qquad k = 0, 1, \ldots, M - 1.$$

This inequality reflects the fact that the error at step $k + 1$ consists of the propagated previous error plus the new error introduced by the most recent step, amplified by the Lipschitz growth factor. We can solve this recurrence exactly by unrolling it. Starting from $\text{error}_0 = 0$,

$$\text{error}_1 \leq e^{L_* \cdot 0}\delta_\epsilon = \delta_\epsilon,$$

$$\text{error}_2 \leq e^{L_* \epsilon}\delta_\epsilon + e_1 \leq \delta_\epsilon\left(1 + e^{L_* \epsilon}\right),$$

$$\text{error}_3 \leq e^{L_* 2\epsilon}\delta_\epsilon + e_2 \leq \delta_\epsilon\left(1 + e^{L_* \epsilon} + e^{L_* 2\epsilon}\right),$$

and so on. Continuing in this fashion for $M$-th sequence, we find the upper bound $e_M \leq \delta_\epsilon \sum_{j=0}^{M-1} e^{L_* j\epsilon}$. This is the discrete analogue of the Duhamel formula for the inhomogeneous linear growth equation and can be viewed as a direct consequence of the discrete Grönwall's inequality. The geometric sum on the right-hand side can be summed in closed form:

$$\sum_{j=0}^{M-1} e^{L_* j\epsilon} = \frac{e^{L_* M\epsilon} - 1}{e^{L_* \epsilon} - 1}.$$

Moreover, using the inequality $e^x - 1 \geq x$ for $x > 0$ with $x = L_*\epsilon$, we can bound the denominator from below and obtain

$$\text{error}_M \leq \delta_\epsilon \frac{e^{L_* T} - 1}{L_* \epsilon}.$$

Finally, substituting the explicit form $\delta_\epsilon = a\epsilon e^{\epsilon L_*} + b\epsilon^{r+1}$ gives

$$\text{error}_M \leq \frac{e^{L_* T} - 1}{L_* \epsilon}\left[\mathfrak{A}\epsilon e^{\epsilon L_*} + \mathfrak{B}\epsilon^{r+1}\right] = \frac{e^{L_* T} - 1}{L_*}\left[\mathfrak{A}e^{\epsilon L_*} + \mathfrak{B}\epsilon^r\right].$$

Recalling the definitions of $\mathfrak{A}$ and $\mathfrak{B}$, this is precisely the bound stated in equation 41, and the proposition is proved. $\qquad\square$

**Proposition A.17** (Relative universal approximation on compact sets). *Fix $T > 0$ and a compact set $K \subset \mathbb{R}^{2n}$. Let $s > 2n$ and $\mathcal{H}_{R,s} := \{H \in C^s(\mathbb{R}^{2n}) : \|H\|_{C^s} \leq R\}$. Assume the DAG parameterization realizes every total–degree-$\leq r$ polynomial $P_r$, hence $X_{P_r} = J\nabla P_r$ for each $r$. Let $L_* > 0$ be a Lipschitz bound on the relevant compact forward–invariant set for $\{X_H : H \in \mathcal{H}_{R,s}\} \cup \{X_{P_r} : r \in \mathbb{N}\}$. Then, for every $\delta > 0$ there exist an integer $r \in \{1, \ldots, s-1\}$ and a step size $\epsilon \in (0, \epsilon_0]$ with $M := \lceil T/\epsilon \rceil$ such that the $M$–step Lie–group walk generated by the DAG polynomial flow satisfies*

$$\sup_{H \in \mathcal{H}_{R,s}} \sup_{z \in K} \left\|\mathcal{W}_{\mathbf{g}}(M)(z) - F_T^{X_H}(z)\right\| < \delta = \mathcal{O}\left(r^{1-s} + \frac{\epsilon^r}{(r+1)!}\right).$$

*Proof of Theorem 4.2.* Fix $H \in \mathcal{H}_{R,s}$. For an integer $r \in \{1, \ldots, s-1\}$, let $P_r$ be a degree–$r$ polynomial chosen (and realized by the DAG) as in Theorem A.15. Apply Proposition A.16 to the pair $(H, P_r)$ on $K$; with $T = M\epsilon$ and the common Lipschitz constant $L_*$ it yields

$$\left\|\mathcal{W}_{\mathbf{g}}(M) - F_T^{X_H}\right\|_{L^\infty(K)} \leq \frac{e^{L_* T} - 1}{L_*}\left[C_1(s, n, r)r^{1-s}\|H\|_{C^s}e^{\epsilon L_*} + \frac{C_2(n, r)}{(r+1)!}\|H\|_{C^{r+1}}\epsilon^r\right].$$

(42)

Since $r \le s - 1$, we have $\|H\|_{C^{r+1}} \le \|H\|_{C^s} \le R$. Also $\|H\|_{C^s} \le R$. Thus the right–hand side of equation 42 is bounded by

$$\frac{e^{L_* T} - 1}{L_*} \left[ C_1(s, n, r) R r^{1-s} e^{\epsilon L_*} + \frac{C_2(n, r)}{(r+1)!} R \epsilon^r \right].$$

Pick $\epsilon \le \min\{\epsilon_0, \ln 2 / L_*\}$ so that $e^{\epsilon L_*} \le 2$. Choose $r \in \{1, \ldots, s-1\}$ large enough that

$$\frac{e^{L_* T} - 1}{L_*} 2 C_1(s, n, r) R r^{1-s} = \frac{e^{L_* T} - 1}{L_*} \frac{C_2(n, r)}{(r+1)!} R \epsilon^r \le \frac{\delta}{2}.$$

In the equality, with this $r$ fixed, we choose $\epsilon > 0$ small enough to satisfy the inequality. These choices make the two bracketed terms in equation 42 each at most $\delta/2$, hence we obtain

$$\left\| \mathcal{W}_{\mathbf{g}}(M) - F_T^{X_H} \right\|_{L^\infty(K)} < \delta(R, s, n, L_*, T)$$
$$= \frac{e^{L_* T} - 1}{L_*} \left[ C_1(s, n, r) R r^{1-s} e^{\epsilon L_*} + \frac{C_2(n, r)}{(r+1)!} R \epsilon^r \right].$$

$$(43)$$

The bounds depend only on $(R, s, n, L_*, T)$ and the fixed constants in Proposition A.16, so the same $(r, \epsilon)$ works uniformly for all $H \in \mathcal{H}_{R,s}$. Taking the supremum over $z \in K$ and $H \in \mathcal{H}_{R,s}$ proves the claim. □

---

**Algorithm 1** COEFNET$_\theta$: Neural–Polynomial Coefficient Network

---

**Require:** Data dimensionality $n$, Batch size $B$, horizon $T$, degree size $d_r$; normalized time grid $\mathbf{t} \in [0, 1]^{B \times T \times 1}$; flattened initial state $x_0 \in \mathbb{R}^{B \times d_s}$.

**Ensure:** Coefficient tensor $C_\theta \in \mathbb{R}^{B \times T \times d_r \times 2n}$.

1: **Time embedding**: for each $(b, t)$ compute $\tau_{b,t} \leftarrow \phi_t(t_{b,t}) \in \mathbb{R}^{d_\text{model}}$, where $\phi_t$ is an MLP.

2: **State embedding**: for each $b$ compute $s_b \leftarrow \phi_s(x_{0,b}) \in \mathbb{R}^{d_\text{model}}$, where $\phi_s$ is an MLP.

3: **Token sequence**: form $T_b \leftarrow \left[ s_b,\ \tau_{b,1}, \ldots, \tau_{b,T} \right] \in \mathbb{R}^{(T+1) \times d_\text{model}}$ for each $b$; stack to $T \in \mathbb{R}^{(T+1) \times B \times d_\text{model}}$.

4: **Transformer encoding**: $E \leftarrow \text{Encoder}_\theta(T) \in \mathbb{R}^{(T+1) \times B \times d_\text{model}}$  ▷ stack of $K$ encoder layers with width $d_\text{model}$

5: **Discard state token**: $E_{1:T} \leftarrow E[2:] \in \mathbb{R}^{T \times B \times d_\text{model}}$.

6: **Projection to coefficients**: for each $(t, b)$ compute

$$a_{t,b} \leftarrow \text{Proj}_\theta\big(E_{t,b}\big) \in \mathbb{R}^{d_r \cdot 6}, \qquad \text{Proj}_\theta = \begin{cases} \text{Linear}(d_\text{model} \to d_r \cdot 6), & \text{single layer,} \\ \text{Linear} \to \text{GELU} \to \text{Linear}, & \text{low rank option.} \end{cases}$$

7: **Reshape**: $C_\theta[b, t, :, :] \leftarrow \text{reshape}\big(a_{t,b}, d_r, 2n\big)$ for all $(b, t)$.

8: **return** $C_\theta$.

---

## A.5 ALGORITHM

In this subsection, we present the algorithmic components that realize the proposed Neural–Poisson framework in a form suitable for direct implementation. Each algorithm corresponds to a distinct stage of the overall pipeline: the construction of the truncated polynomial generator set and its Lie–algebraic structure, the generation of time–conditioned Hamiltonian coefficients by a neural network, the parallel composition of elementary flows via an associative scan, and the high–order symplectic integration of the resulting vector fields. Together, these procedures provide a complete, self–contained specification of the model from algebraic preprocessing to numerical time stepping.

**Neural–Polynomial Coefficient Network.** Algorithm 1 specifies the coefficient–generation module COEFNET$_\theta$, which defines the parametrization map from the conditioning variables to the coefficients of a truncated polynomial Hamiltonian. Given the normalized time grid and the flattened initial state, the network produces the coefficient tensor $C_\theta \in \mathbb{R}^{B \times T \times d_r \times 6}$ whose entries determine the time–dependent polynomial force field employed in the subsequent symplectic integration.

The procedure begins by embedding each time token $t_{b,t}$ into a latent vector $\tau_{b,t} = \phi_t(t_{b,t}) \in \mathbb{R}^{d_\text{model}}$ using a dedicated multi–layer perceptron, and mapping the flattened initial state $x_{0,b}$ into $s_b = \phi_s(x_{0,b}) \in \mathbb{R}^{d_\text{model}}$ through a separate multi–layer perceptron. The initial–state embedding is prepended as a special token to the sequence of time embeddings, yielding a tensor in $\mathbb{R}^{(T+1) \times B \times d_\text{model}}$. This sequence is processed by a Transformer encoder Encoder$_\theta$, whose self–attention layers allow each time step to condition on the initial state and to exchange information with other time steps, thereby capturing global temporal dependencies in the coefficient representation. After encoding, the state token is discarded, and each remaining time–step embedding is mapped by a projection head Proj$_\theta$ to a vector in $\mathbb{R}^{d_r \cdot 6}$ corresponding to the coefficients of all $d_r$ basis monomials for the six coordinate components. These vectors are then reshaped to form $C_\theta$, which is returned as the network output. This architecture realises a sequence–to–tensor regression map whose inductive biases, consisting of distinct embeddings for time and state, a shared latent representation, and global self–attention, are chosen to respect the structural constraints inherent in Hamiltonian dynamics.

---

**Algorithm 2** BUILDNP_DAG: Neural–Poisson Composition DAG construction

---

**Require:** Polynomial degree bound $r \geq 1$, DAG depth $K \geq 0$, local phase dimension $d$.
**Ensure:** Nodes $\mathcal{G}^{(\leq K)} = \bigcup_{k=0}^{K} \mathcal{G}^{(k)}$, lookup map idx : Exp $\rightarrow \{1, \ldots, d_r\}$, and structure tensor $C \in \mathbb{R}^{d_r \times d_r \times d_r}$ satisfying $\{m_i, m_j\} = \sum_m C_{ijm} m_m$ on $P_{\leq r}$.

1: Construct the truncated monomial basis $P_{\leq r} = \text{span}\{m_\alpha : q^a p^b \mid |a| + |b| \leq r\}$ and its exponent list Exp $= \{(a, b)\}$ in $\mathbb{N}^{2d}$.
2: Set $\mathcal{G}^{(0)} \leftarrow P_{\leq r}$ and initialize $\mathcal{G}^{(\leq 0)} \leftarrow \mathcal{G}^{(0)}$.
3: Build idx by assigning a unique index to each exponent in Exp and set $d_r \leftarrow |\text{Exp}|$.
4: Initialize $C \leftarrow 0$ in $\mathbb{R}^{d_r \times d_r \times d_r}$.
5: **for** $k = 1$ to $K$ **do**
6:     $\mathcal{C} \leftarrow \emptyset$                                               ▷ candidate set at layer $k$
7:     **for** each $g \in \mathcal{G}^{(k-1)}$ and each $h \in \mathcal{G}^{(\leq k-1)}$ **do**
8:         $u \leftarrow \pi_{\leq r}(gh)$                               ▷ projected product
9:         $v \leftarrow \pi_{\leq r}\{g, h\}$                       ▷ projected Poisson bracket
10:         Add $u$ and $v$ to $\mathcal{C}$
11:     **end for**
12:     $\mathcal{G}^{(k)} \leftarrow$ unique elements of $\mathcal{C}$ under exponent equality
13:     $\mathcal{G}^{(\leq k)} \leftarrow \mathcal{G}^{(\leq k-1)} \cup \mathcal{G}^{(k)}$
14: **end for**
15: **Structure tensor** $C$: for monomials $m_i = q^{a_i} p^{b_i}$ and $m_j = q^{a_j} p^{b_j}$,
16: **for** $i = 1$ to $d_r$ **do**
17:     **for** $j = 1$ to $d_r$ **do**
18:         **for** $\ell = 1$ to $d$ **do**
19:             $s \leftarrow a_i^{(\ell)} b_j^{(\ell)} - a_j^{(\ell)} b_i^{(\ell)}$
20:             **if** $s \neq 0$ **then**
21:                 $a \leftarrow a_i + a_j - e_\ell$, $b \leftarrow b_i + b_j - e_\ell$
22:                 **if** $(a, b) \in$ Exp with index $m = \text{idx}(a, b)$ **then**
23:                     $C_{ijm} \leftarrow C_{ijm} + s$
24:                 **end if**
25:             **end if**
26:         **end for**
27:     **end for**
28: **end for**
29: **return** $\left(\mathcal{G}^{(\leq K)}, \text{idx}, C\right)$

---

**Neural–Poisson Composition DAG construction**. Algorithm 2 describes the construction of the Neural–Poisson composition directed acyclic graph (DAG), which encodes the closure of a truncated polynomial Hamiltonian basis under both pointwise multiplication and the Poisson bracket. The procedure begins by generating the set of all monomials $m_\alpha = q^a p^b$ whose total degree satisfies $|a| + |b| \le r$, forming the truncated polynomial space $P_{\le r}$ together with its exponent list $\mathrm{Exp}$. This list is used to initialise the zeroth layer $\mathcal{G}^{(0)}$ of the DAG, which contains all basis monomials, and to build an index map $\mathrm{idx}$ assigning a unique integer to each exponent tuple. The integer $d_r = |\mathrm{Exp}|$ records the size of the basis.

Subsequent layers are constructed iteratively up to depth $K$. At each layer $k$, all possible projected products $\pi_{\le r}(gh)$ and projected Poisson brackets $\pi_{\le r}\{g, h\}$ are computed between generators $g$ from the previous layer $\mathcal{G}^{(k-1)}$ and generators $h$ from all earlier layers $\mathcal{G}^{(\le k-1)}$. These operations are projected back into $P_{\le r}$ to preserve the degree bound. The set of distinct resulting monomials, identified by their exponent tuples, forms the new layer $\mathcal{G}^{(k)}$, which is then added to the cumulative set $\mathcal{G}^{(\le k)}$.

After the generator set is complete, the algorithm computes the structure tensor $C \in \mathbb{R}^{d_r \times d_r \times d_r}$, which encodes the Poisson bracket structure on $P_{\le r}$. For each ordered pair of basis monomials $(m_i, m_j)$, the integer coefficient $s = a_i^{(\ell)} b_j^{(\ell)} - a_j^{(\ell)} b_i^{(\ell)}$ is evaluated for each coordinate index $\ell$, reflecting the antisymmetry of the canonical Poisson structure. When the resulting exponent tuple $(a, b)$ lies in $\mathrm{Exp}$, the corresponding structure constant $C_{ijm}$ is incremented by $s$. The output of the procedure is the complete generator set $\mathcal{G}^{(\le K)}$, the exponent–to–index map $\mathrm{idx}$, and the structure tensor $C$, which together provide an explicit, finite–dimensional Lie algebra representation of the truncated Hamiltonian vector fields.

---

**Algorithm 3** PARALLELSCAN: Associative prefix composition of Lie coefficients

---

**Require:** Initial state $z_0 \in \mathbb{R}^{B \times N \times 2d}$, horizon $T$, repeat factor $L \ge 1$, step size $\varepsilon$, coefficient network output $\{c_t\}_{t=1}^T$ with $c_t \in \mathbb{R}^{B \times d_r \times 6}$, structure tensor $C$ from Alg. 2.
**Ensure:** Trajectory $(z_0, z_1, \ldots, z_T)$ with $z_t \in \mathbb{R}^{B \times N \times 2d}$.
1: Define the bilinear Poisson contraction on coefficients

$$[c^{(1)}, c^{(2)}]_C \text{ with } \left([c^{(1)}, c^{(2)}]_C\right)_{b,m,\gamma} = \sum_{i,j} c_{b,i,\gamma}^{(1)} c_{b,j,\gamma}^{(2)} C_{ijm}.$$

2: Define the associative combine operator

$$c^{\mathrm{new}} \oplus c^{\mathrm{acc}} = c^{\mathrm{new}} + c^{\mathrm{acc}} + \tfrac{1}{2}\varepsilon [c^{\mathrm{new}}, c^{\mathrm{acc}}]_C.$$

3: Form the length $TL$ sequence by repetition $\tilde{c}_{(t-1)L+\ell} \leftarrow c_t$ for $t = 1, \ldots, T$ and $\ell = 1, \ldots, L$.
4: Compute scanned coefficients by an associative prefix scan

$$s_k = \tilde{c}_1 \oplus \tilde{c}_2 \oplus \cdots \oplus \tilde{c}_k \quad \text{for } k = 1, \ldots, TL$$

using a tree based parallel scan with depth $O(\log TL)$.
5: Extract the effective per step coefficients $c_t^{\mathrm{eff}} \leftarrow s_{tL+L-1}$ for $t = 1, \ldots, T$.
6: Set $z \leftarrow z_0$ and initialize $\mathrm{Traj} \leftarrow [z_0]$.
7: **for** $t = 1$ to $T$ **do**
8: $\quad z \leftarrow$ APPLYLIE$(z, c_t^{\mathrm{eff}}, C, \varepsilon)$ $\qquad\qquad\qquad\qquad$ ▷ $r$-th order symplectic update in code
9: $\quad$ Append $z$ to $\mathrm{Traj}$
10: **end for**
11: **return** $\mathrm{Traj}$

---

**Parallel Scanning.** Algorithm 3 implements the associative parallel–scan procedure used to compose, in logarithmic depth, the sequence of truncated Hamiltonian flows generated by the coefficient network. The method operates entirely in coefficient space by exploiting the bilinearity and antisymmetry of the Poisson bracket, together with the finite–dimensional closure property of the Neural–Poisson algebra. Given the initial state $z_0 \in \mathbb{R}^{B \times N \times 2d}$, the per–step coefficients $\{c_t\}_{t=1}^T$ produced by COEFNET$_\theta$, and the structure tensor $C$ computed from Algorithm 2, the first step defines the bilinear Poisson contraction $[c^{(1)}, c^{(2)}]_C$, which maps a pair of coefficient tables to the coefficient

table of their Poisson bracket. This operation realises the algebraic commutator of the corresponding truncated Hamiltonian vector fields.

An associative binary operator $\oplus$ is then defined on coefficient tables by

$$c^{\text{new}} \oplus c^{\text{acc}} = c^{\text{new}} + c^{\text{acc}} + \frac{1}{2}\varepsilon[c^{\text{new}}, c^{\text{acc}}]_C,$$

which corresponds to the second–order Baker–Campbell–Hausdorff truncation for composing two infinitesimal Hamiltonian flows with step size $\varepsilon$. The sequence $\{c_t\}$ is repeated $L$ times per physical time step to form a length–$TL$ list $\{\tilde{c}_k\}$, allowing for sub–stepping or repeated application of the same coefficients.

The algorithm then applies a tree–based associative prefix scan with the operator $\oplus$ to $\{\tilde{c}_k\}$, producing the partial compositions $s_k = \tilde{c}_1 \oplus \cdots \oplus \tilde{c}_k$ in $O(\log TL)$ parallel depth. From these, the effective coefficients $c_t^{\text{eff}}$ for each physical step are extracted as $c_t^{\text{eff}} = s_{tL+L-1}$.

Finally, starting from $z_0$, the state is advanced sequentially by applying the symplectic Lie integrator APPLYLIE with coefficients $c_t^{\text{eff}}$ and step size $\varepsilon$ for $t = 1, \ldots, T$. The trajectory $\{z_t\}_{t=0}^{T}$ so obtained matches, up to the truncation order of $\oplus$, the result of composing all $T$ Hamiltonian flows in the order prescribed by the Neural–Poisson Lie–group walk, while benefiting from the parallel efficiency of the scan.

---

**Algorithm 4** APPLYLIE_R (explicit, even $r$): $r$-th order symplectic update via Suzuki–Yoshida composition

---

**Require:** Current state $z \in \mathbb{R}^{B \times N \times 2d}$, coefficient table $c \in \mathbb{R}^{B \times d_r \times 6}$, exponent tensor Exp, step size $\varepsilon > 0$, target order $r \in 2\mathbb{N}$ (even).
**Ensure:** Next state $z_+ = \Phi_r(\varepsilon)z$.
1: **Base 2nd–order kernel** *(Störmer–Verlet)*:

$\quad\quad$ KERNELSV$(z, h)$ :

$\quad\quad\quad g_1 \leftarrow \nabla_{q,p}\hat{H}(z; c, \text{Exp})$

$\quad\quad\quad \dot{q}_1 \leftarrow \partial_p\hat{H}(z), \quad \dot{p}_1 \leftarrow -\partial_q\hat{H}(z), \quad p_{1/2} \leftarrow p + \frac{h}{2}\dot{p}_1, \quad q' \leftarrow q + h\dot{q}_1$

$\quad\quad\quad g_2 \leftarrow \nabla_{q,p}\hat{H}\big((q', p_{1/2}); c, \text{Exp}\big), \quad \dot{p}_2 \leftarrow -\partial_q\hat{H}(q', p_{1/2}), \quad p' \leftarrow p_{1/2} + \frac{h}{2}\dot{p}_2$

$\quad\quad\quad$ **return** $z' = (q', p')$

2: **Yoshida/Suzuki coefficient recursion** (order $2 \to 4 \to \cdots \to r$):
$\quad$ • Initialize the coefficient list for order 2: $\mathcal{W}^{(2)} \leftarrow [1]$.
$\quad$ • For $m = 1$ to $r/2 - 1$: set $\alpha \leftarrow 2^{1/(2m+1)}$, $w_1 \leftarrow \dfrac{1}{2 - \alpha}$, $w_0 \leftarrow -\dfrac{\alpha}{2 - \alpha}$, and

$$\mathcal{W}^{(2m+2)} \leftarrow \big[w_1\mathcal{W}^{(2m)}, \ w_0\mathcal{W}^{(2m)}, \ w_1\mathcal{W}^{(2m)}\big]$$

$\quad\quad$ (concatenation with all entries scaled by the prefactors).
3: **Explicit composition:** set $z^{(0)} \leftarrow z$ and let $\{w_j\}_{j=1}^{S} \leftarrow \mathcal{W}^{(r)}$.
4: **for** $j = 1$ to $S$ **do**
5: $\quad z^{(j)} \leftarrow$ KERNELSV$\big(z^{(j-1)}, w_j\varepsilon\big)$
6: **end for**
7: **return** $z_+ \leftarrow z^{(S)}$.

---

Algorithm 4 implements an explicit $r$th–order symplectic Lie–integration scheme for truncated polynomial Hamiltonian systems, where $r$ is even. The method is based on the Suzuki–Yoshida composition of a symmetric second–order kernel, which in this context is given by the Störmer–Verlet splitting of the Hamiltonian vector field.

The input consists of the current state $z = (q, p) \in \mathbb{R}^{B \times N \times 2d}$, the coefficient table $c$ defining the truncated Hamiltonian $\hat{H}(\cdot; c, \text{Exp})$ in the monomial basis indexed by Exp, a time step $\varepsilon > 0$, and the target even order $r$. The procedure begins by defining the base integrator KERNELSV, which applies one step of the symmetric Störmer–Verlet method to $\hat{H}$ with step size $h$. This kernel consists of a

first half–step update of the momenta (kick), a full–step update of the positions (drift), and a final half–step update of the momenta, where the forces are obtained from the gradient $\nabla_{q,p}\hat{H}$ computed by the truncated polynomial gradient routine.

To reach order $r > 2$, the algorithm constructs the Yoshida–Suzuki composition coefficients $\mathcal{W}^{(r)}$ recursively. Starting from $\mathcal{W}^{(2)} = [1]$ for the base kernel, each stage $m$ is extended to order $2m + 2$ by computing $\alpha = 2^{1/(2m+1)}$ and the weights

$$w_1 = \frac{1}{2-\alpha}, \qquad w_0 = -\frac{\alpha}{2-\alpha},$$

and concatenating the sequence $[w_1\mathcal{W}^{(2m)}, w_0\mathcal{W}^{(2m)}, w_1\mathcal{W}^{(2m)}]$. This recursion preserves the symmetry of the composition and ensures that the local error order increases by two at each step.

In the final composition phase, the method applies the base kernel KERNELSV successively with scaled step sizes $w_j\varepsilon$ for each weight $w_j$ in $\mathcal{W}^{(r)}$, propagating the state from $z^{(0)} = z$ through intermediate states $z^{(j)}$ until $z^{(S)}$, where $S = |\mathcal{W}^{(r)}|$. The output $z_+ = z^{(S)}$ is the state advanced by a single $r$th–order symplectic step of size $\varepsilon$. This explicit composition preserves the symplectic structure exactly and achieves the prescribed order $r$ for smooth truncated polynomial Hamiltonians, making it suitable for high–accuracy long–time integration in the Neural–Poisson framework.

## A.6 IMPLEMENTATION DETAILS

This section specifies the choices required to reproduce our results, including the experimental setup, baseline families, evaluation metrics, runtime and memory protocol, and hyperparameters. It then details the datasets and physical models used in our studies. Unless stated otherwise, all methods share the same splits, physical step size, trajectory length, and device configuration. Any deviations are explicitly highlighted in the corresponding paragraphs and in the appendix.

### A.6.1 EXPERIMENTAL SETUP

**Baselines.** We group methods into three families that mirror the table layout in Fig. 2 and Table 1. **(A)** Classical symplectic and high-order integrators using the ground-truth Hamiltonian include Störmer–Verlet (Verlet, 1967), Forest–Ruth 4th order (Forest & Ruth, 1990), and the $s = 2$ Gauss–Legendre collocation scheme (Hairer et al., 2006). Learned Hamiltonian generators include SymODEN (Zhong et al., 2020), SRNN (Chen et al., 2020), and Nonseparable Symplectic Neural Networks (Xiong et al., 2021). **(B)** We include KoVAE, a Koopman-regularized variational autoencoder for dynamical data (Naiman et al., 2024), and SKOLR, a structured Koopman-operator linear RNN forecaster (Zhang et al., 2025b). **(C)** For a line of research in this group, we evaluate EqMotion (Xu et al., 2023), a learned SE(3)-equivariant motion model for atomic trajectories. GeoTDM (Han et al., 2024), a diffusion model over geometric trajectories that enforces SE(3)-equivariance and ET-SEED (Tie et al., 2025), a recent deep generative trajectory model. These methods serve as representative black-box generative baselines that do not incorporate Hamiltonian or Koopman structure.

**Evaluation Metrics.** We assess prediction accuracy and symplectic structure preservation using four complementary metrics. Prediction accuracy is quantified by the Average and Final Displacement Error, given by $\text{ADE} = \frac{1}{N} \sum_{i=1}^{N} \frac{1}{T_i} \sum_{t=1}^{T_i} \|z_{i,t} - \hat{z}_{i,t}\|$ and $\text{FDE} = \frac{1}{N} \sum_{i=1}^{N} \|z_{i,T_i} - \hat{z}_{i,T_i}\|$, respectively, where $z_{i,t}$ are predicted states, $\hat{z}_{i,t}$ are ground-truth states, and $T_i$ is the length of the $i$-th trajectory. Lower values of ADE and FDE indicate superior trajectory prediction accuracy. Structure preservation is evaluated via energy drift ($\Delta E$) and symplectic-form violation ($\Delta \omega$). Energy drift is measured as the average Hamiltonian difference, $\Delta E = \frac{1}{N} \sum_{i=1}^{N} \frac{1}{T_i} \sum_{t=1}^{T_i} |H(z_{i,t}) - H(\hat{z}_{i,t})|$. Symplectic-form violation, defined as $\Delta \omega = \|J^\top \nabla J + \nabla J^\top J\|_F$ quantifies deviations from canonical symplectic structure. Lower values of $\Delta E$ and $\Delta \omega$ indicate better physical consistency.

**Runtime and Memory Measurement.** We report *Inference* (seconds per $T$–step rollout) and, when available, peak GPU memory under a standardized protocol. Specifically, this value shows the wall–clock time to propagate a single initialization $z_0$ through a $T$–step rollout (no backward pass), while the training wall–clock per iteration divided by $T$ (one full rollout + loss + backward + optimizer step, normalized by the physical horizon). All timings are obtained with device–synchronized hardware timers. and the memory is reported as the peak device–resident allocation recorded by the runtime. We average over three runs after a short warm-up, and keep the device type, batch size, trajectory length, and numerical step size identical across methods to ensure fair comparison.

**Hyperparameters.** Unless otherwise noted, the Poisson basis degree and DAG depth are set to $(r, K) = (5, 2)$, and the model step is matched to the data step, $\varepsilon = \Delta t$. Associative composition uses the first-order BCH correction where higher-order variants and sensitivity to $(r, K)$ are reported in ablation study. Random seeds are fixed for data generation and training, and validation performance (ADE) is used for early stopping and minor tuning of method-specific regularization where applicable.

A.6.2   SPIN DYNAMICS

In our first experiment, we consider the classical dynamics of spin systems arranged on periodic lattices in one, two, or three dimensions. Each spin is described by a unit vector $\mathbf{S}_i \in \mathbb{S}^2$, representing a classical magnetic moment at lattice site $i$. A representative atomistic Hamiltonian capturing the principal interactions typically encountered in magnetic simulations and experiments is expressed as follows:

$$H = -\sum_{\langle ij \rangle} J_{ij} \mathbf{S}_i \cdot \mathbf{S}_j + \sum_{\langle ij \rangle} \mathbf{D}_{ij} \cdot (\mathbf{S}_i \times \mathbf{S}_j) - K \sum_i \left( \mathbf{S}_i \cdot \hat{\mathbf{n}} \right)^2 - \sum_i \mathbf{h} \cdot \mathbf{S}_i$$
$$+ \frac{1}{2} \sum_{i \neq j} \mathbf{S}_i^\top \mathbf{T}_{ij} \mathbf{S}_j - \sum_{\langle\langle ij \rangle\rangle} J_{ij}^{(2)} \mathbf{S}_i \cdot \mathbf{S}_j. \tag{44}$$

Each term within this Hamiltonian corresponds to a distinct physical mechanism shaping the static and dynamic properties of spin systems. Below, we briefly describe the nature and physical implications of each interaction.

**(i) Heisenberg exchange interaction.**   The nearest-neighbor Heisenberg exchange interaction, given by the first term $-\sum_{\langle ij \rangle} J_{ij} \mathbf{S}_i \cdot \mathbf{S}_j$, primarily governs the alignment between neighboring spins. For positive exchange coefficients ($J_{ij} > 0$), parallel spin alignment is energetically favored, thus leading to ferromagnetic ordering. Conversely, negative coefficients ($J_{ij} < 0$) promote antiparallel alignment, resulting in antiferromagnetic configurations. This interaction sets the fundamental energy scale for magnetic ordering phenomena and determines short-range correlations and spin-wave excitations observed in spin lattices (Heisenberg, 1928).

**(ii) Dzyaloshinskii–Moriya interaction (DMI).**   The second term, known as the Dzyaloshinskii–Moriya interaction, arises from antisymmetric spin-orbit coupling effects inherent in crystal structures lacking inversion symmetry. Formally represented as $\sum_{\langle ij \rangle} \mathbf{D}_{ij} \cdot (\mathbf{S}_i \times \mathbf{S}_j)$, the DMI promotes non-collinear spin textures exhibiting distinct chirality, such as helices, spirals, and skyrmions. This interaction competes with the symmetric exchange and magnetic anisotropy terms, thus significantly influencing the formation and stabilization of complex spin configurations (Dzyaloshinskii, 1958; Moriya, 1960).

**(iii) Magnetocrystalline anisotropy.**   The magnetocrystalline anisotropy, expressed by the uniaxial term $-K \sum_i (\mathbf{S}_i \cdot \hat{\mathbf{n}})^2$, incorporates crystal-field effects and spin-orbit coupling that energetically distinguish particular crystallographic directions. Depending on the sign of the anisotropy constant $K$, spins preferentially align either along a particular axis (easy-axis anisotropy for $K > 0$) or within a particular plane (easy-plane anisotropy for $K < 0$). This term strongly influences critical magnetic properties such as coercivity, domain-wall structure, and ferromagnetic resonance frequencies (Aharoni, 1996).

**(iv) Zeeman interaction (external field coupling).**   The Zeeman interaction, represented as $-\sum_i \mathbf{h} \cdot \mathbf{S}_i$, captures the response of magnetic moments to external magnetic fields. This term governs magnetization reversal processes, field-driven hysteresis loops, and spin-wave frequency shifts induced by external magnetic fields. By controlling external fields, experimentalists exploit this coupling to study fundamental spin dynamics and to manipulate spin textures and magnetization states (Evans et al., 2014).

**(v) Dipolar (magnetostatic) interaction.**   The long-range dipolar, or magnetostatic, interaction is encapsulated by the term $\frac{1}{2} \sum_{i \neq j} \mathbf{S}_i^\top \mathbf{T}_{ij} \mathbf{S}_j$, where

$$\mathbf{T}_{ij} = \mu_0 \mu_s^2 r_{ij}^{-3} \left( \mathbf{I} - 3 \hat{\mathbf{r}}_{ij} \hat{\mathbf{r}}_{ij}^\top \right).$$

This interaction accounts for the internal demagnetizing fields arising from spatially distributed magnetic moments. It profoundly influences domain formation, domain-wall structures, and shape-dependent magnetic phenomena such as thin-film magnetism, shape anisotropy, and vortex formation (Aharoni, 1996).

**(vi) Next-nearest neighbor exchange** ($J_2$). Finally, the next-nearest neighbor exchange interaction $-\sum_{\langle\langle ij \rangle\rangle} J_{ij}^{(2)} \mathbf{S}_i \cdot \mathbf{S}_j$ introduces competing interactions, thus inducing magnetic frustration in the spin system. The competition between nearest and next-nearest neighbor exchanges ($J_1$-$J_2$ models) can lead to complex ground states including spirals, stripes, or even disordered spin liquids, depending sensitively on the relative magnitude of $J_2/J_1$. Such interactions are critical in understanding frustrated magnetic materials and their associated spin-wave spectra and phase transitions (**??**).

### A.6.3 MOLECULAR DYNAMICS

In this work, we study two prototypical molecular systems: **aspirin** (SMILES: CC(=O)OC1=CC=CC=C1C(=O)O) and **methane** (SMILES: C). For each molecule, we first generate the initial three-dimensional atomic coordinates directly from their respective SMILES strings using RDKit with the ETKDGv3 embedding algorithm. Hydrogen atoms are explicitly added, and the resulting molecular structures undergo a brief geometry optimization step using the Universal Force Field (UFF).

**System construction (OpenMM).** We then construct corresponding molecular systems based on these optimized geometries using OpenMM. The molecular interactions are parameterized via template generators provided by the openmmforcefields package. Unless stated otherwise, we employ the **OpenFF Sage 2.1.0** force field via the SMIRNOFFTemplateGenerator, utilizing the parameter file openff-2.1.0.offxml. Additionally, for comparison and ablation purposes, we optionally adopt the **GAFF-2.11** force field. Nonbonded interactions default to the NoCutoff method, which is suitable for small isolated molecular systems, though the Particle-Mesh Ewald (PME) method is also supported and can be used if required. Constraints are disabled, and the removal of center-of-mass (COM) motion is explicitly enabled at the system-building stage.

**Unit conventions.** Throughout the experiments, all public API interfaces strictly follow standard OpenMM unit conventions. Specifically, atomic positions $q \in \mathbb{R}^{N \times 3}$ are expressed in nanometers (nm); atomic momenta $p \in \mathbb{R}^{N \times 3}$ are given in units of dalton·nm/ps; energies, including kinetic ($K$), potential ($U$), and total Hamiltonian ($H$), are consistently reported in kilojoules per mole (kJ/mol). Atomic velocities are similarly represented in nm/ps.

**Hamiltonian evaluation.** We evaluate the total Hamiltonian energy as $H(q, p) = K(p) + U(q)$, utilizing a lightweight OpenMM Context. The potential energy $U(q)$ is directly obtained from the OpenMM context via the getState() function. Similarly, the kinetic energy $K(p)$ is computed from the context using getKineticEnergy() after atomic velocities are set according to their corresponding momenta and masses ($v = p/m$). Furthermore, each force term (e.g., bond, angle, torsion, nonbonded) in the system is assigned a unique force-group identifier, enabling individual extraction and analysis of potential energy contributions per term.

**Initialization.** Initial atomic coordinates $q_0$ are generated based on the RDKit molecular embeddings, followed by the addition of small, independent Gaussian perturbations to prevent initial structural biases. Specifically, the perturbation magnitude $\sigma_{\text{Å}}$ is set to 0.02 Å for aspirin and 0.01 Å for methane, balancing the need for structural variability without inducing significant distortions:

$$q_0 \leftarrow q_0 + \delta, \quad \delta \sim \mathcal{N}(0, \sigma_{\text{Å}}^2 I) \quad (\text{Å}), \text{ then converted to nm.}$$

Initial atomic momenta $p_0$ are sampled according to the Maxwell–Boltzmann distribution at a chosen temperature of $T = 1000$K, implemented via OpenMM's setVelocitiesToTemperature method. The sampled velocities are subsequently mapped to momenta using atomic masses ($p_0 = mv_0$). Unless specifically noted, we do not remove the system's total center-of-mass momentum. All random seeds used during initialization (*i.e.*, ETKDGv3 embedding seed ($= 7$) and velocity sampling seed ($= 2025$)) are explicitly fixed to ensure reproducibility across experiments.

**Time integration (NVE).** We propagate classical Hamiltonian dynamics under **microcanonical** (NVE) conditions, thereby ensuring the conservation of total energy without introducing external thermostats. Specifically, we employ the standard **Velocity Verlet** integrator, adopting a time step of 1.0fs and simulating each trajectory for a total of 1000 integration steps unless otherwise indicated.

Table 4: **Key simulation settings used for the aspirin and methane molecular dynamics simulations**. Both molecules are parameterized using the OpenFF Sage 2.1.0 force field without cutoffs for nonbonded interactions. Initial atomic coordinates ($q_0$) are perturbed by Gaussian noise with standard deviations ($\sigma_{\text{Å}}$) specified below. Initial atomic momenta ($p_0$) are sampled from a Maxwell–Boltzmann distribution at temperature $T = 1000K$. Center-of-mass momentum removal is not performed. Trajectories are propagated using the Velocity Verlet integrator with a 1.0fs timestep and run for 1000 integration steps. Atomic coordinates from each integration step are exported in XYZ format for visualization.

|  | **Aspirin** | **Methane** |
| --- | --- | --- |
| SMILES | CC(=O)OC1=CC=CC=C1C(=O)O | C |
| Force field | OpenFF 2.1.0 (main) | OpenFF 2.1.0 (main) |
| Nonbonded | NoCutoff | NoCutoff |
| $\sigma_{\text{Å}}$ (coord. jitter) | 0.02 | 0.01 |
| $T$ for $p_0$ (K) | 1000 | 1000 |
| COM momentum removal | No | No |
| Time step (fs) | 1.0 | 1.0 |
| Steps | 1000 | 1000 |
| Trajectory export | XYZ (stride 1) | XYZ (stride 1) |

At each integration step, the atomic forces are computed directly from OpenMM by querying the state with `getState(getForces=True)` at the current molecular coordinates. Subsequently, atomic positions and momenta are updated according to the classical Velocity Verlet scheme, implemented explicitly in Python. We utilize OpenMM's built-in `VerletIntegrator` solely for initializing the simulation context, whereas the explicit propagation is conducted externally to provide precise control and transparent tracking of energy accounting throughout each trajectory.

**Observables and logging.** Throughout the simulation trajectory, we record the instantaneous kinetic energy $K_t$, potential energy $U_t$, and total Hamiltonian $H_t = K_t + U_t$ at every integration step, with energies consistently expressed in kilojoules per mole ($\text{kJ/mol}$). Additionally, we separately log the contributions of individual potential energy terms (e.g., bonds, angles, torsions, nonbonded interactions) enabled by the unique assignment of force-group identifiers. The quality of energy conservation during simulation is characterized by analyzing the energy drift $\Delta H_t = H_t - H_0$, from which summary statistics including mean, maximum, and root-mean-square (RMS) deviations are computed over each trajectory. For external analysis and visualization purposes (such as with OVITO), multi-frame atomic trajectories are also exported in XYZ format (positions in Å).

**Platform configuration.** Unless explicitly noted, all simulations presented here are conducted OpenMM `CUDA` platform configured with `Precision='mixed'` to achieve an optimal balance between computational efficiency and numerical accuracy, thereby enhancing energy conservation while maintaining high throughput. Unless otherwise stated, results reported throughout the main text are generated using this mixed-precision setting on the specified computational platform.

