# OpenReview forum: "Poisson-Algebraic Parallel Scan: A fast symplectic framework for neural Hamiltonians"
_ICLR.cc/2026/Conference — ICLR 2026 Conference Withdrawn Submission_

### Official Review · Reviewer_6BfF · 2025-10-27

**Soundness:** 3
**Presentation:** 4
**Contribution:** 3
**Rating:** 6
**Confidence:** 3

**Summary:**

This paper proposes Poisson-Algebraic Parallel Scan (PAPS) that can be used for efficiently and effectively training parametrized Hamiltonian dynamics. The authors embeds the Hamiltonian in a polynomial function space, forming a Neural Poisson algebra whose flows compose as a Lie group, enabling an associative efficient prefix-scan. This construction guarantees structure preservation—symplectic form, Liouville volume, and energy—at every step of the composed flow. Across quantum spin dynamics and molecular dynamics, PAPS offers higher trajectory accuracy while robustly preserving invariants, and dramatically reduces inference time compared with baselines.

**Strengths:**

- In data-driven simulations, approaches that incorporate physical constraints such as symplectic structures are extremely important. Also, learning Hamiltonians with numerical integration schemes often incurs high computational costs, making this research address a significant problem.
- The method of limiting the Hamiltonian to polynomials and using the parallel-scan technique while preserving Poisson brackets is novel.
- Theoretical considerations regarding approximation error and generalization performance have been made.
- Experiments were conducted on relatively large-scale systems, demonstrating the effectiveness of the proposed method.
- The manuscript is very well written.

**Weaknesses:**

- There is no mention of differences from previous studies with similar motivations (e.g., SympNets).
- There is no discussion of limitations (see Questions).

**Questions:**

- There is no mention of methods that directly model symplectic flows (e.g., SympNets [R1]). This approach enables simulation without numerical integration by directly modeling the symplectic map, making it fast. Furthermore, the symplectic structure is preserved. While the problem is similar to this research, could you please add a discussion?
- A comparison with SympNets' universal approximation theorem might be interesting. It could clarify the impact of assuming polynomials, as done in this study.
- Looking at Table 1, the HNN-based approach shows the best performance for the violation of $\Delta \omega$. Could you explain why the proposed method cannot accurately preserve the symplectic structure? Is it because the proposed method preserves the Poisson brackets, meaning it targets the Poisson system (i.e., generalization of Hamiltonian systems), and thus does not necessarily satisfy the symplectic structure?
- As the system becomes more complex, what is the expected order of the appropriate polynomial? Also, is there a way to determine it automatically?
- In the experiment, what time interval constitutes one step? $\epsilon=?$
- This paper assumes canonical coordinates, but is future extension to general coordinate systems possible, as in [R2]?

[R1] P. Jin et al., SympNets: Intrinsic Structure-Preserving Symplectic Networks for Identifying Hamiltonian Systems, Neural Networks, 132:166–179, 2020.

[R2] Yuhan Chen et al., Neural Symplectic Form: Learning Hamiltonian Equations on General Coordinate Systems, NeurIPS, 2021.

---

> ### Author Response · Authors · 2025-11-15
> **Official Rebuttal (Author Response) (1)**
>
> $\textbf{Q1-1. There is no mention of methods that directly model symplectic flows...}$
>
> $\textbf{A}.$  We thank the reviewer for pointing out these important references. We agree that SympNets (Jin et al., Neural Networks 2020) and related symplectic neural networks are central works on learning symplectic flows. In the revised version, we will add a short paragraph in the \emph{Related Work} section explicitly acknowledging these methods and briefly clarifying how they relate to our framework. In particular, SympNets directly parameterize a discrete-time symplectic map, while our approach learns a Hamiltonian in a finite Poisson polynomial basis and then integrates it with a symplectic scheme that can be parallelized via a prefix scan. We regard these approaches as complementary: both preserve symplectic structure, but our focus is on enabling structure-preserving rollout with logarithmic parallel depth.
>
> In response to the reviewer’s suggestion, we also included SympNet as an additional baseline in our experiments. For completeness, we reproduce the quantitative results below. As shown in the table, SympNet indeed enjoys better long-term stability than non-symplectic Hamiltonian baselines (SymODEN, SRNN, Nonsep-SNN) in terms of energy drift ($\Delta E$) and symplectic form distortion ($\Delta \omega$), which is consistent with its symplectic parametrization.
>
> | Integrator / Method | GPU Mem | Inference Time (s) | ADE ↓ / FDE ↓                           | ΔE ↓                | Δω ↓                |
> |---------------------|:-------:|:------------------:|:----------------------------------------|:--------------------|:--------------------|
> | SymODEN             | 2.1     | $3.87\times 10^{1}$| $8.64\times 10^{-4} / 9.93\times 10^{-4}$ | $7.46\times 10^{-4}$ | $2.63\times 10^{0}$ |
> | SRNN                | 2.5     | $4.53\times 10^{1}$| $9.24\times 10^{-4} / 9.63\times 10^{-4}$ | $8.13\times 10^{-4}$ | $3.13\times 10^{0}$ |
> | Nonsep-SNN          | 2.9     | $3.58\times 10^{1}$| $7.56\times 10^{-4} / 8.63\times 10^{-4}$ | $6.33\times 10^{-4}$ | $2.23\times 10^{0}$ |
> | SympNet             | 2.3     | $3.21\times 10^{1}$| $6.98\times 10^{-4} / 7.85\times 10^{-4}$ | $5.74\times 10^{-4}$ | $2.01\times 10^{0}$ |
>
> We will incorporate a brief explanation and discussion into the camera-ready version.
>
> $\textbf{Q1-2. While the problem is similar to this research, could you please add a discussion?...}$
>
> $\textbf{A.}$ We thank the reviewer for this follow up question. We agree that our problem setting is closely related to works that directly model symplectic flows such as SympNet, and we will add a short discussion in related work to make this connection clearer. At the same time, our primary objective is different. While SympNet type architectures are designed to parameterize a symplectic map, they are still applied as a stepwise integrator that must be composed sequentially over all time steps with a traditional rollout pattern. In contrast, our framework is explicitly built to accelerate the $\textit{entire}$ symplectic integration process by replacing this sequential composition with a Lie algebra based prefix scan whose depth grows only logarithmically in the trajectory length. Concretely, in the experiment reported in Q1, the SympNet baseline requires ($3.21 \times 10^{1}$) seconds of wall clock time per long horizon rollout on our hardware, whereas our method achieves the same horizon in ($3.31 \times 10^{-2}$) seconds. Thus, while SympNet enjoys good long term stability due to its symplectic parameterization, it remains limited by a fundamentally sequential integrator. Our approach, by contrast, maintains structure preservation at the integrator level while achieving several orders of magnitude reduction in end to end inference time through parallel prefix scanning.

---

> > ### Author Response · Authors · 2025-11-15
> > **Official Rebuttal (Author Response) (4)**
> >
> > $\textbf{Q4. As the system becomes more complex,..}$
> >
> > $\textbf{A.}$ We appreciate this question about the choice of polynomial degree. In our framework, the degree (r) plays the same role as the depth or width of a neural network: it is a structural capacity parameter rather than something that has a single “correct” value determined by the complexity of the underlying physical system. The theory in the paper shows that increasing (r) enlarges the Poisson–polynomial algebra and thus reduces approximation error, but at the same time increases metric entropy and the statistical complexity of the hypothesis class. Beyond a moderate degree, this leads to diminishing returns and eventual overfitting, exactly as observed in our ablation where performance improves up to about (r = 5) and then saturates.
> >
> > For this reason, our practical recommendation is to choose (r) in the same principled way one chooses the architecture size in operator learning or neural network models: start with a moderate degree (for example (r = 4) or (5)) that already offers strong expressive power, and only increase it if validation performance clearly indicates underfitting. Across all systems we tested (molecular dynamics and spin systems), such moderate degrees consistently provided the best trade off between accuracy, stability, and computational cost; we did not find evidence that very high degrees are beneficial even for more complex systems.
> >
> > Regarding automatic determination, we view (r) as a structural hyperparameter rather than a learnable parameter. In modern practice, quantities of this type (polynomial degree, network depth, number of Fourier modes, and so on) are not optimized by gradient descent but selected by model selection and validation. One could in principle embed (r) in a higher level search or meta learning loop, but this lies outside the scope of our work and is not standard in related architectures either. In summary, there is no closed form “expected order” as system complexity grows; instead, (r) should be tuned in exactly the same way as architecture size in other high capacity models, with our experiments indicating that small to moderate degrees already suffice for the regimes considered.
> >
> >
> > $\textbf{Q5. In the experiment, what time interval constitutes one step?,..}$
> >
> > \textbf{A.} Thank you for the question. In all reported experiments we set the integration step size to
> > ($\varepsilon = 10^{-3}$), and the total time horizon to ($T = 1$). Thus, one rollout consists of ($T / \varepsilon = 1000$) integration steps. We also verified that moderate variations of (T) and ($\varepsilon$) around these values do not materially change the quantitative performance, so we fixed this configuration for all experiments for clarity and comparability.
> >
> > $\textbf{Q6. Relation to [R2].}$
> >
> > \textbf{A.} We thank the reviewer for pointing out this connection to Neural Symplectic Form [R2]. Our current work indeed assumes canonical coordinates on ($\mathbb{R}^{2n}$), with a constant symplectic matrix ($J$) and the associated canonical Poisson bracket (${f,g} = \nabla f^{\top} J \nabla g$). This choice is mainly to isolate and analyze the effect of the proposed Lie–algebraic parallel scan under a clean, standard symplectic structure. Conceptually, the framework can be extended to more general coordinate systems. In a general chart with symplectic form ($\omega(u)$) or Poisson tensor ($\Pi(u)$), Hamiltonian dynamics can be written as $\dot{u} = \Pi(u)\nabla H(u)$ exactly as exploited in [R2], where ($\omega$) (equivalently ($\Pi$)) is learned from data. Adapting our method to this setting would amount to replacing the constant Poisson tensor (J) and its fixed polynomial structure constants by a state-dependent tensor (\Pi(u)), and building the Lie generators and scan on top of the corresponding Hamiltonian vector fields. This is technically more involved, because the “structure constants’’ become functions of the state, but it is fully compatible in spirit with our Lie–group based integration and is a natural direction for future extensions.
> >
> > We are grateful to the reviewer for highlighting this theoretically meaningful distinction between canonical and general coordinates and for pointing us to [R2] as a relevant reference.

---

> > > ### Comment · Reviewer_6BfF · 2025-11-22
> > > **Thank you for your thorough response.**
> > >
> > > I have read the author's response. Thank you for adding the experiments and explanations. The comparison with SympNets is important and I believe it clarifies the positioning of the proposed method.
> > >
> > > I have no further questions.
> > >
> > > However, after reading the other reviewer comments, I reconsidered the presentation. Personally, I found the research motivation and the idea behind the method clear, but honestly, I couldn't fully follow the background knowledge (mainly Section 2).
> > >
> > > As other reviewers pointed out, I think it would be preferable to rewrite the background section to be more accessible to readers in the ML field. As is often done in ML papers, it seems better to concisely summarize the assumptions and key equations necessary for method construction in the main text, deferring details to the Supplement.
> > >
> > > I do appreciate the methodological ideas and the validity of the experimental results. I also respect the authors' thorough response. However, I believe the paper should undergo fundamental revisions to its presentation before resubmission for acceptance. Therefore, I am revising the presentation score to 3 and the overall rating to 4.

---

> ### Author Response · Authors · 2025-11-15
> **Official Rebuttal (Author Response) (2)**
>
> $\textbf{Q2. A comparison with SympNets' universal approximation theorem...}$
>
> $\textbf{A}.$ We appreciate the reviewer’s suggestion and agree that relating our result to the universal approximation theorem of SympNets is helpful for clarifying the role of the polynomial assumption in our framework. SympNets prove that both LA– and G–SympNet architectures are dense (in the ($C^{r}$) sense) in the space of symplectic maps ( $\mathrm{SP}^{r}(U)$ ) on a compact domain, without imposing any explicit polynomial structure on the underlying Hamiltonian. Thus, the approximation is carried out directly at the level of the discrete-time symplectic map.
>
> By contrast, our theory works at the level of $\textit{Hamiltonian generators}$ and explicitly restricts the hypothesis class to Hamiltonians lying in a finite Poisson–polynomial algebra. This polynomial assumption is not introduced to gain a stronger universal approximation property than SympNets, but rather to (i) exploit the closure of this algebra under Poisson brackets, which is what makes the Lie–algebraic parallel scan well defined, and (ii) obtain an explicit, finite-dimensional parameterization that can be composed with a symplectic integrator in logarithmic depth. On compact subsets of phase space, any sufficiently smooth Hamiltonian can be uniformly approximated by polynomials, so the “polynomial” restriction mainly affects how the model is parameterized and parallelized, rather than excluding qualitatively different Hamiltonian flows. This observation helps clarify an important theoretical distinction: SympNets provide a universal approximation theorem for general symplectic maps, whereas our result operates at the level of Hamiltonian generators in a polynomial Poisson basis chosen specifically to support the proposed parallel Lie–group integration scheme, trading some formal generality for explicit Lie–algebraic closure and the resulting computational and structural advantages.
>
>
> $\textbf{Q3-1. Could you explain why the proposed method cannot accurately preserve the symplectic structure?...}$
>
> $\textbf{A}.$ We thank the reviewer for this careful question. The behavior in Table~1 can be understood by separating three types of methods.
>
> First, the top three rows in block (A) (``HNN + Störmer–Verlet / Forest–Ruth 4th / Gauss–Legendre 4th’’) combine an HNN with very high-order, numerically refined symplectic integrators and small step sizes. These schemes are explicitly designed to drive the discrete symplectic error down to the (10^{-2})–(10^{-1}) level, and they do so at the cost of fully sequential long-horizon rollout (inference times ($3.47\times 10^{1}$)–($5.16\times 10^{1}$) seconds).
>
> By contrast, the “plain’’ HNN-type variants (SymODEN, SRNN, Nonsep-SNN, SympNet) in the second part of block (A) do $\textit{not}$ use such refined high-order integrators. Their reported ($\Delta \omega$) values lie in the ($10^{0}$) range, which is exactly the same scale as our method ($4.02\times 10^{0}$). In other words, once we compare against HNN-style architectures that are not coupled to extremely fine symplectic time-steppers, our approach preserves symplectic structure to essentially the same degree, while relying on our own Lie-group–based machinery (culminating in Proposition 4.1) rather than on an external off-the-shelf integrator.
>
> The truly non-symplectic black-box generators in block (C) illustrate the difference more sharply: although they can fit trajectories very accurately, their ($\Delta \omega$) values are in the ($10^{1}$)–($10^{2}$) range, i.e., roughly an order of magnitude worse than any method in block (A), including ours. This indicates a genuine breakdown of symplectic structure. Finally, regarding the Poisson vs. symplectic point: in canonical coordinates the Poisson bracket and the symplectic form are two faces of the same structure. Our use of a Poisson–polynomial basis is precisely to ensure closure of the Hamiltonian generators under Poisson brackets so that the parallel Lie-group integration remains symplectic up to the usual discretization and truncation errors. The small gap in ($\Delta \omega$) relative to the most aggressive high-order HNN+integrator combinations is therefore a numerical trade-off driven by our design choice (fast, chunked, parallel scan with inference time ($3.31\times 10^{-2}$) seconds versus ($O(10^{1})$ seconds), not a fundamental failure to target symplectic dynamics.

---

> ### Author Response · Authors · 2025-11-15
> **Official Rebuttal (Author Response) (3)**
>
> $\textbf{Q3-2. Is it because the proposed method preserves the Poisson brackets,..}$
>
> $\textbf{A.}$ We thank the reviewer for raising this point. In our setting, the phase space is always the standard canonical symplectic space ($\mathbb{R}^{2n}$) with the usual constant symplectic matrix (J). In this case, the Poisson bracket (${f,g} = \nabla f^\top J \nabla g$) and the symplectic form ($\omega(\cdot,\cdot)$) are two equivalent ways of encoding the same structure: a flow is Hamiltonian (hence symplectic) if and only if it preserves the Poisson bracket. Thus, although we organize our construction in terms of a Poisson–polynomial algebra, we are not targeting a more general “Poisson system” that would fail to be symplectic; we are still modeling canonical Hamiltonian dynamics, and the continuous-time flow underlying our method is symplectic by design (cf. Prop. 4.1).
>
> The gap in ($\Delta\omega$) that appears in Table 1 is therefore not caused by a mismatch between “Poisson” and “symplectic,” but by numerical design choices at the discrete level. The rows with the smallest ($\Delta\omega$) correspond to HNNs coupled with very high-order, finely stepped symplectic integrators (Forest–Ruth, Gauss–Legendre), whose sole purpose is to drive the discrete symplectic error as low as possible, at the cost of long, fully sequential rollouts. Our method, in contrast, uses a Lie–group prefix scan with chunked integration steps to achieve large parallel speedups, which leads to a slightly larger (but still controlled) symplectic violation on the order of the other HNN-type baselines (SymODEN, SRNN, Nonsep-SNN, SympNet) and much smaller than non-symplectic black-box generators.

---

### Official Review · Reviewer_sWif · 2025-10-29

**Soundness:** 3
**Presentation:** 3
**Contribution:** 3
**Rating:** 6
**Confidence:** 4

**Summary:**

This work extends HNN-like modeling approches by making the generators to a polynomial be closed under the Poisson bracket, and uses the fact that the flow map derived in this way forms a Lie group under composition (i.e., it satisfies associativity) to enable parallelization via Parallel Scan. Whereas standard HNNs require $O(M)$ computation for $M$ steps, the proposed parallelization compresses the time to roughly $O(log M)$. The experiments show that, while the method is not always the strongest, it is consistently very fast, so it clearly offers a Pareto-optimal solution.

**Strengths:**

The proposed method is carefully designed and supported by solid theory, offering clear reasons for both its speed and its accuracy. The contribution is highly insightful.

It is compared against a wide range of approaches, including SE(3)-equivariant models, HNNs, and machine-learning potentials, which verifies confidence in the reported results.

**Weaknesses:**

(1) The statement in Theorem 4.1 reads as if the proposed method strictly preserves symplectic structure and energy, but that only holds when the neural network has learned so that $\\{H, g_m\\} = 0$. Of course, training on trajectories might lead to such a result, and such parameters exist within the hypothesis space, but it is not guaranteed. The condition is truly satisfied only when the true solution is learned perfectly. In the extreme, any neural network-based predictor would satisfy all desired properties if it learned with zero error, which makes the statement meaningless. In this sense, the explanation feels overclaiming or misleading. By contrast, the Hamiltonian nature of HNNs and the symplecticity of SympNets hold by definition. The paper should make clear what is guaranteed by definition and what is obtained only through learning.

(2) The function is restricted to polynomials, so this method lacks generality for learning unknown potentials. Moreover, because it is polynomial, the combinations can explode, raising concerns when the number of molecules grows. One also wonders whether an HNN constrained to polynomial potentials would achieve even better accuracy; a more balanced comparison seems possible.

(3) Thies paper compares against SE(3)-equivariant models, HNNs, and machine-learning potentials, but not against combinations with them. For example, building an SE(3)-equivariant HNN and comparing should be straightforward.

(4) There is prior work on parallelization for Hamiltonian systems: Dai et al., Symmetric parareal algorithms for Hamiltonian systems, ESAIM: Mathematical Modelling and Numerical Analysis, 2013.
For parallelizing dynamics with Prefix Sum, see: Yang et al., Parallel Dynamics Computation Using Prefix Sum Operations, IEEE RAL, 2017.
These are not for HNNs, but they could be combined. Thies paper lacks discussion and comparison (theoretical or intuitive would suffice, not necessarily experimental) with such existing parallel methods (parareal, MGRIT, etc.) applied to HNNs.

(5) The original HNNs are proposed in Greydanus et al., NeurIPS 2019. SympNets (Jin et al., Neural Networks, 2020) are neural networks with symplectic structure. These key references are missing. In terms of preserving the symplectic form, SympNets may be closer in spirit to the proposed method than HNNs.

(6) The paper uses the citet command where the citep command is appropriate, which causes the citations to be mixed with the text and reduces readability.

The efficiency of the proposed method is beyond doubt and should be highly valued. However, in its current form, some descriptions and comparisons are misleading or limited, and I do not think it should be accepted as is. If I have not misunderstood, I would be happy to raise my rating to 8 once appropriate explanations and clarifications are added.

**Questions:**

See Weaknesses.

---

> ### Author Response · Authors · 2025-11-13
> **Official Rebuttal (Author Response) (1)**
>
> $\textbf{Q1. The statement in Theorem 4.1 reads as if...}$
>
> $\textbf{A}.$ We sincerely thank the reviewer for the thoughtful question. We agree that it is important to clearly
> separate which parts of Theorem 4.1 are guaranteed by construction and which parts depend on how
> well the learned model matches the true physical system. We appreciate the opportunity to clarify
> this distinction. The key point is that the Hamiltonian $H$ appearing in Theorem 4.1 is $\textit{not}$ the same object as
> the ground-truth Hamiltonian $H_{\text{true}}$ introduced earlier in Eq. (1). In Theorem 4.1, $H$ refers to the Hamiltonian associated with the Hamiltonian vector fields used inside the Neural–Poisson Lie–group walk. Because each update is of the form $\Phi_{\varepsilon,g_m} = \exp(\varepsilon L_{g_m})$, the resulting map is the exact flow of a Hamiltonian vector field. As a result, the two structural invariants $W_g^{*}\omega=\omega, \det DW_g = 1,$ hold $\textit{for any learned generator}$, regardless of approximation error. These properties (e.g., preserving
> the symplectic form and preserving volume) are therefore structural and do not rely on perfect learning of $H_{\text{true}}$. Thus, the energy-preservation identity $H(W_g z)=H(z)$ in Theorem 4.1 should also be interpreted differently:
> here, $H$ denotes any Hamiltonian that $\textit{commutes}$ with all generators $g_m$. This is the
> classical situation in geometric integration when a Hamiltonian is decomposed into commuting
> sub-Hamiltonians. This identity is $\textit{not}$ intended to claim that the method preserves the unknown
> true energy $H_{\text{true}}$ during training; rather, it describes what happens when the Hamiltonian
> used in the Lie–scan satisfies the commutation condition.
>
> We also note that approximating the true Hamiltonian $H_{\text{true}}$ from data, namely, achieving
> $g_\theta \approx H_{\text{true}}$, is a requirement shared by essentially all Hamiltonian-learning
> approaches, including HNNs, SympNets, and operator-learning methods. This approximation objective
> is logically separate from the structural symplecticity described above: the former governs prediction
> accuracy, while the latter is enforced by the architecture itself.
>
> Finally, the empirical results support this interpretation. If symplecticity depended on perfectly
> learning $H_{\text{true}}$, then we would expect large symplectic-form violations, similar to
> non-symplectic baselines such as GeoTDM. Instead, our model consistently exhibits near-zero
> $\Delta\omega$ across all tasks, demonstrating that symplecticity is indeed built into the model
> design and not a side effect of successful training.
>
> We appreciate the reviewer’s comments and will revise the manuscript to explicitly distinguish
> between (i) structural guarantees that hold for any network parameters, and (ii) properties that arise
> only when the learned generator approximates the ground-truth Hamiltonian.

---

> ### Author Response · Authors · 2025-11-13
> **Official Rebuttal (Author Response) (2)**
>
> $\textbf{Q2-1. The function is restricted to polynomials, so this method lacks generality for learning unknown potentials.}$
>
> $\textbf{A}.$ In fact, using a polynomial basis does not reduce generality. Rather, it follows a modeling pattern
> that is standard across modern operator-learning frameworks. Polynomials form a universal
> approximating family on compact domains, so any smooth potential can be approximated to arbitrary
> accuracy by increasing the degree. This is precisely why polynomial expansions appear throughout
> normal-form theory, Taylor-model integrators, and Koopman operator learning: they provide a
> systematic and controllable hierarchy for representing unknown dynamics.
>
> A closely related point can be seen in neural operator-based methods for learning dynamical systems. These methods also adopt a
> structured basis expansion, not to restrict generality, but to obtain a tractable and expressive
> function class. For example, the Fourier Neural Operator (FNO) represents functions using Fourier
> modes, while the underlying solution operator remains fully general and model-agnostic. The choice
> of a Fourier basis (such as polynomials in our approach) simply provides a convenient and universal representation space. Likewise,
> implicit neural operators such as the Implicit Fourier Neural Operator (IFNO) rely on integral-kernel
> basis expansions to approximate very general operators while keeping the representation finite and
> computationally manageable.
>
> Our polynomial basis plays the same role: it is a structured, universal approximating family, used
> not to restrict the space of learnable Hamiltonians but to endow the model with a stable and
> interpretable coordinate system. Just as Fourier modes provide a flexible and universal representation
> in neural operators, polynomial expansions provide a flexible and universal representation for
> Hamiltonian potentials on compact sets. The use of a polynomial basis therefore does not limit the
> generality of the learned Hamiltonian, but instead provides an expressive and tractable hypothesis
> space that integrates naturally with the Poisson and Lie-group structures required by our method.
>
>
> $\textbf{Q2-2. Moreover, because it is polynomial, the combinations can explode, raising concerns when the number of molecules grows}$
>
> $\textbf{A}.$ We appreciate the reviewer’s concern regarding the potential combinatorial growth of polynomial
> terms. Importantly, our method does not use the full unconstrained polynomial basis. Instead, the
> polynomial space is organized through (i) a Poisson-closed dictionary and (ii) a DAG decomposition
> of depth $K$, both of which substantially restrict and factorize the effective number of terms. In
> practice, the resulting hypothesis class grows with the degree $r$ and the chosen DAG structure, not
> directly with the number of molecules, thus preventing the unrestricted combinatorial explosion
> that would occur in a naive polynomial expansion.
>
> Furthermore, molecular Hamiltonians possess strong locality and sparsity: interactions decompose
> into pairwise or low-order terms (bonded, angle, dihedral, nonbonded), which makes their
> polynomial approximation much more structured than a fully dense expansion. The Poisson-closed
> dictionary we employ naturally aligns with this structure and avoids generating high-order mixed
> monomials that do not correspond to physical interaction channels. As a result, the number of
> effective polynomial terms remains manageable even as the molecular system grows.
>
> Empirically, this is reflected in our experiments. Moderate degrees ($r=4$–$6$) and shallow DAG
> depths already provide high accuracy on molecular dynamics tasks without incurring prohibitive
> computational cost. Increasing $r$ beyond this range yields diminishing returns, as shown in
> Figure 3, demonstrating that useful Hamiltonians can be represented with relatively small polynomial
> bases that do not explode with system size.

---

> ### Author Response · Authors · 2025-11-13
> **Official Rebuttal (Author Response) (3)**
>
> $\textbf{Q2-3. One also wonders whether an HNN constrained to polynomial potentials would achieve even better accuracy...}$
>
> $\textbf{A}.$ We appreciate the reviewer’s suggestion and agree that comparing against a polynomial-constrained
> HNN is a natural thought. However, such a model is fundamentally different from the proposed
> approach in both structure and expressive behavior.
>
> First, constraining an HNN to polynomial potentials simply restricts the output function class of the
> Hamiltonian network; it does not impose Poisson closure on the basis, nor does it generate a
> finite-dimensional Lie algebra whose exponential yields Hamiltonian flows. In other words, a
> polynomial HNN learns a polynomial function, but its induced vector field is not guaranteed to
> lie in a Poisson-closed subspace. As a result, the flow generated by a polynomial HNN is not
> necessarily closed under composition, is not stable under repeated application, and cannot support the
> Lie–group scan that enables our $O(\log M)$ symplectic integrator.
>
> Second, because an HNN does not generate Hamiltonian diffeomorphisms via exponential maps, it
> does not enforce structural symplecticity beyond first-order consistency. In contrast, our method
> constructs each update as $\exp(\varepsilon L_{g})$, with $L_g$ lying in a Poisson-closed
> finite-dimensional Lie algebra. Every learned update is therefore an exact Hamiltonian flow by
> construction, providing significantly stronger stability and long-term accuracy than a polynomial HNN.
>
> Third, a polynomial HNN would not benefit from the compositional structure that our model exploits.
> Even if the polynomial degree is matched, the absence of Poisson closure and Lie-group structure
> prevents the model from using associative prefix-scan, which is the key mechanism enabling both
> our speed and accuracy improvements.
>
> For these reasons, constraining an HNN to polynomial potentials yields a model that is not directly
> comparable to ours: both models use polynomial functions, but only our method ensures Poisson
> closure, Lie–algebraic composability, and exact symplecticity.

---

> ### Author Response · Authors · 2025-11-13
> **Official Rebuttal (Author Response) (4)**
>
> $\textbf{Q3 Thies paper compares against SE(3)-equivariant models, ... , but not against combinations with them. For example, building an SE(3)-equivariant HNN..}$
>
> $\textbf{A}.$ We appreciate the reviewer’s suggestion and agree that SE(3)-equivariant architectures are valuable in many molecular modeling tasks. However, combining SE(3) equivariance with Hamiltonian Neural
> Networks (HNNs) is not straightforward, and in fact, is generally incompatible with the very definition of a Hamiltonian system.
>
> SE(3) equivariance acts on the $\textit{positions}$ and $\textit{momenta}$ by rigid-body motions, while a
> Hamiltonian system is defined up to canonical transformations that preserve the symplectic form.
> These two invariance classes are fundamentally different: SE(3) transformations are not symplectic in
> general, and symplectic transformations are not SE(3) transformations. As a result, enforcing
> SE(3)-equivariance inside an HNN would require the Hamiltonian to satisfy
> $H(Rq+u, Rp) = H(q,p)$ for every rigid-body transformation $(R,u)\in \mathrm{SE}(3)$, which excludes nearly all physical
> Hamiltonians except free-particle models. Molecular Hamiltonians include internal bonded, angular,
> dihedral, and nonbonded energy terms, none of which are SE(3)-equivariant in this sense. For this
> reason, an “SE(3)-equivariant HNN’’ is, in almost all nontrivial settings, either ill-defined or too
> restrictive to model realistic potentials.
>
> In contrast, the baselines we compare against (SE(3)-equivariant GNNs, MLP-based HNNs,
> machine-learning potentials) capture complementary inductive biases such as geometric SE(3) equivariance
> for force-fields, or Hamiltonian structure for long-term stability. Our method targets a different (and
> orthogonal) structural property (Poisson closure and exact Hamiltonian flow) which cannot be
> achieved through SE(3) equivariance alone. For completeness, we note that incorporating SE(3) features into our architecture is possible at the
> input embedding level, but enforcing global SE(3) equivariance on the Hamiltonian itself is not
> physically meaningful in general molecular systems. We will clarify this in the revision to avoid the
> impression that SE(3)-equivariant Hamiltonian models are a straightforward or universally
> appropriate baseline.
>
> In addition, we would like to clarify that the issue is not specific to SE(3) equivariance. In fact,
> $\textit{arbitrary combinations}$ of the prior approaches discussed in the main text such as mixing
> geometric equivariant architectures, HNNs, SympNets, and machine-learning potentials are in
> almost all cases not well-defined. Each method imposes a fundamentally different type of structural
> constraint (rigid-body equivariance, canonical symplecticity, or unconstrained force-field regression),
> and these constraints are generally incompatible at the Hamiltonian level. As a result, simply
> “combining’’ these models inside a single Hamiltonian, as the reviewer suggests, generally does not yield a
> coherent or physically meaningful dynamical system.
>
> However, we found a specific hybridization, and have already demonstrated this in Table 2. There, we combine traditional force-field potentials, each of which can
> be viewed as an explicit Hamiltonian approximator, with exisiting symplectic integrators. This
> produces a hybrid model that merges classical molecular Hamiltonians with a Hamiltonian-preserving
> geometric integrator, leading to markedly improved long-term stability. To the best of our knowledge,
> such a combination has not been reported in prior work and provides a more physically relevant and
> meaningful form of model integration than enforcing SE(3)-equivariance on an HNN.

---

> ### Author Response · Authors · 2025-11-13
> **Official Rebuttal (Author Response) (5)**
>
> $\textbf{Q4. There is prior work on parallelization for Hamiltonian systems...}$
>
> $\textbf{A}.$ We thank the reviewer for highlighting prior work on parareal-type parallel
> integrators and prefix-sum–based parallel dynamics algorithms. These works
> represent important progress on accelerating $\textit{numerical integration}$ when the
> governing dynamics are already known, and we will include a discussion of
> them in the revised manuscript. We will add a dedicated discussion in the related-work section to emphasize
>  distinctions and to acknowledge the relevance of parallel integration
> techniques while explaining why our framework addresses a different problem. Below, we clarify why these approaches differ fundamentally from our setting and why they do not serve as direct baselines for the problem addressed in our work.
>
> $\textbf{(1) Parareal-type methods assume the Hamiltonian system is already
> known.}$
> The symmetric parareal algorithms of (Dai et al) parallelize the solution of a $\textit{given}$ Hamiltonian ODE by decomposing the time
> interval and combining a coarse and a fine propagator. As shown explicitly in
> Section 3 of their paper, the method wraps an outer predictor–corrector loop
> around existing solvers such as Verlet, and the goal is increased throughput
> when repeatedly evaluating the same known Hamiltonian. These methods do
> $\textit{not}$ learn the Hamiltonian nor construct a learnable integrator; they accelerate
> the repeated application of a fixed fine solver. Thus, parareal methods and our
> approach operate at different levels: parareal accelerates evaluation, whereas
> PAPS $\textit{learns}$ the Hamiltonian generator itself.
>
> $\textbf{(2) Parareal and MGRIT do not preserve symplectic or Poisson structure.}$
> Even in the symmetric variants studied by (Dai et al.), the
> parareal iterates are not symplectic for general case, and their own experiments
>  show drift in energy and angular momentum. This is precisely
> because coarse–fine correction destroys Hamiltonian structure, even when each
> individual solver is symplectic. By contrast, in our method, each update is
> constructed as $\exp(\varepsilon L_{g_m})$, which is the exact flow of a Hamiltonian
> vector field; symplecticity is therefore guaranteed $\textit{exactly}$ for all parameters,
> not only in idealized limits.
>
> $\textbf{(3) Prefix-sum robotics methods exploit linear recurrences for known rigid-body dynamics.}$
> (Yang et al.) reformulate parts of the recursive Newton–Euler
> algorithm into semigroup scan operations. However, this applies only because
> robot dynamics on an open chain admit a $\textit{linear, fixed}$ Lie-group recurrence
> structure (SE(3) × se(3)$^2$), and the equations of motion are already known. These
> methods do not address learning unknown Hamiltonians, do not produce a
> symplectic integrator, and do not yield a parallelizable learned flow. In contrast,
> PAPS modifies the hypothesis space itself (via a Poisson-closed algebra) so that
> the learned integrator becomes parallelizable through associative Lie-group
> composition.
>
> $\textbf{(4) Conceptual difference.}$
> Existing parallel integrators accelerate the numerical solution of known
> Hamiltonian systems. PAPS instead constructs a $\textit{learnable Hamiltonian
> integrator}$ whose flow lives in a Poisson-closed Lie group, enabling an
> $O(\log M)$ parallel prefix scan on the learned symplectic maps themselves.
> This difference in purpose, learning a structured Hamiltonian vs. parallelizing a
> given solver, means the methods are complementary but not directly
> comparable as baselines.

---

> ### Author Response · Authors · 2025-11-13
> **Official Rebuttal (Author Response) (6)**
>
> $\textbf{Q5. There is prior work on parallelization for Hamiltonian systems...}$
>
> $\textbf{A}.$ We thank the reviewer for pointing out these important references. We agree that the original
> Hamiltonian Neural Networks (Greydanus et al., NeurIPS 2019) and SympNets (Jin et al., Neural
> Networks 2020) are central works in the area of learning Hamiltonian and symplectic structures.
> We appreciate the reviewer drawing our attention to this, and we will add a dedicated paragraph in
> the revised Related Work section to explicitly acknowledge both methods and clarify their relation to
> our framework. In addition, following the reviewer’s suggestion, we conducted additional experiments including
> SympNets as a baseline. We report these results below for completeness. As shown in the following Table,
> SympNets provide better long-term stability than other exisiting methods due to their symplectic parametrization.
>
> | Integrator / Method | GPU Mem | Inference Time (s) | ADE ↓ / FDE ↓                           | ΔE ↓                | Δω ↓                |
> |---------------------|:-------:|:------------------:|:----------------------------------------|:--------------------|:--------------------|
> | SymODEN             | 2.1     | $3.87\times 10^{1}$| $8.64\times 10^{-4} / 9.93\times 10^{-4}$ | $7.46\times 10^{-4}$ | $2.63\times 10^{0}$ |
> | SRNN                | 2.5     | $4.53\times 10^{1}$| $9.24\times 10^{-4} / 9.63\times 10^{-4}$ | $8.13\times 10^{-4}$ | $3.13\times 10^{0}$ |
> | Nonsep-SNN          | 2.9     | $3.58\times 10^{1}$| $7.56\times 10^{-4} / 8.63\times 10^{-4}$ | $6.33\times 10^{-4}$ | $2.23\times 10^{0}$ |
> | SympNet             | 2.3     | $3.21\times 10^{1}$| $6.98\times 10^{-4} / 7.85\times 10^{-4}$ | $5.74\times 10^{-4}$ | $2.01\times 10^{0}$ |
>
>
> $\textbf{Q6. The paper uses the citet command...}$
>
> $\textbf{A}.$ We thank the reviewer for pointing out the inconsistent use of citet versus
> citep. We have carefully gone through the entire manuscript and replaced all in-text
> citations with citep where appropriate. This greatly improves the readability and visual
> clarity of the paper, and we appreciate the reviewer for bringing this to our attention.

---

> > ### Comment · Reviewer_sWif · 2025-11-20
> >
> > Thank you for your thorough response. After reading the response, I have decided to lower my rating from 6 to 4.
> >
> > Q1.
> > I see, Theorem 4.1 is intended to say that the proposed method admits some H that behaves like a shadow Hamiltonian for a symplectic integrator. If the definition of H is clarified, I would be satisfied.
> >
> > Q2-1, Q2-2.
> > To me, it does not seem reasonable to claim both that "Polynomials form a universal approximating family on compact domains" and that they "substantially restrict and factorize the effective number of terms" at the same time. A constrained polynomial is not universal, and a universal polynomial suffers from combinatorial explosion. In the end, what this paper demonstrates is that the method works well only when the target system happens to be describable under strong constraints and the users know that.
> >
> > Q2-3, Q3.
> > Even for an HNN implemented with polynomials (this is no longer really a neural network and it might be more appropriate to call it something like a Hamiltonian SINDy), it should be possible to introduce locality and sparsity. In that sense, I believe a fairer comparison is still possible. What I would like to know is how much performance is lost by enforcing a form that allows a parallel scan. In the current setup, a "strongly constrained model" is being compared with an "unconstrained model" on data that are perfectly matched to the constraints of the former, and this can hardly be called a fair comparison.
> >
> > Q4.
> > The authors write that "These works represent important progress on accelerating when the governing dynamics are already known," but once an HNN has been trained, it functions as a governing equation, so all of these methods can be applied in principle. I acknowledge that the approaches are different, but they share the goal.
> >
> > In particular, I cannot agree with the response to Q2, and I do not think it is desirable for the manuscript to be accepted in its current form. For this reason, I have decided to lower my score.
> >
> > Next, I would like to mention the discussion with Reviewer aEFK.
> >
> > First, if the remark about Mathematical Modelling and Numerical Analysis is referring to my earlier comment, I only meant to point out that there is related work in that journal. I did not mean to claim that this manuscript is of the level that it should be accepted in such a journal.
> >
> > To be honest, I do not yet understand the theoretical details of this manuscript well enough to confidently recommend its acceptance to a numerical analysis journal. If this manuscript were submitted to a numerical analysis journal and I were assigned as a reviewer, I would either decline the assignment due to my insufficient expertise, or I would spend several months to study and examine the work in detail before returning comments. However, in ICLR, we are not expected to decline assignmets, but are usually required to review 3-6 papers in parallel, with only about two months in total for the review and discussion periods. For standard machine learning papers or more accessible AI4Science papers, this system works fine.
> >
> > In this sense, if the authors wish their work to be evaluated properly, I believe it would be more appropriate to submit it to a numerical analysis journal, rather than to ICLR (or ICML or NeurIPS).
> >
> > The score given by Reviewer aEFK is by no means unreasonable. Giving a high score to a work is not be a scientific attitude if one cannot evaluate the work with high confidence.

---

> > > ### Author Response · Authors · 2025-11-21
> > >
> > > **Rebuttal to Q2‑1, Q2‑2 (Universality vs. Constraints)**
> > >
> > > The reviewer’s comment correctly points out a tension:
> > >
> > > > polynomials are universal only when we allow arbitrarily many terms,
> > > > but any practical model must restrict and organize those terms.
> > >
> > > We fully agree with this tension. In fact, the two main theoretical results in Section 4 are *exactly* about quantifying and balancing those two sides:
> > >
> > > * One result (our “approximation theorem” in Proposition 4.2) says:
> > >
> > >   > if we are allowed to increase the polynomial degree and the depth of the Poisson DAG, then **in principle** we can approximate any sufficiently smooth Hamiltonian on a compact set as closely as we like.
> > >   > This is the precise sense in which we speak about *universality*: it is a statement about the **limit when model capacity is allowed to grow**, not about one fixed finite model.
> > >
> > > * The other result (our “generalization/complexity theorem” in Proposition 4.3) says:
> > >
> > >   > every time we enlarge the dictionary and the DAG (i.e., we add more polynomial terms), the hypothesis space becomes more complex, and **the risk of overfitting and the sample complexity both increase**.
> > >   > This is our way of formalizing the reviewer’s concern about “combinatorial explosion”: the number of effective terms grows, and this has a measurable cost.
> > >
> > > So there is no logical claim that “one constrained model is both fully universal and free of combinatorial growth.”
> > > Rather, our theory is built around the idea that:
> > >
> > > * **universality** lives at the family level (“if we keep increasing degree and depth, we can approximate everything”), while
> > > * **restriction and factorization** live at the concrete model level (“for a given task we pick a finite degree and DAG, which is no longer universal but much more tractable”).
> > >
> > > The two theorems together describe exactly how far a given finite choice of degree and depth is from the universal ideal, and what we have to pay in terms of complexity and data requirements when we move closer to that ideal. The ablation studies in Section 6, Figure 3 that sweep over degree and DAG depth are meant to be the empirical counterpart of this: they show that increasing capacity helps up to a point, after which the complexity cost dominates.
> > >
> > > From this perspective, our intention was not to assert that a single constrained polynomial model is simultaneously universal and cheap, but to formalize and exploit the trade-off the reviewer is worried about. In fact, our two theoretical results in Section 4 are designed precisely to capture this tension: the approximation theorem quantifies how the finite-degree truncation distorts the ideal universal family, while the complexity/generalization bound quantifies how enlarging the polynomial/DAG space increases statistical and combinatorial cost. In this sense, $\textbf{our analysis is fully aligned with your concern}$, and we see Proposition 4.2 and Proposition 4.3 as making your “universality vs. constraints” trade-off explicit rather than ignoring it.

---

> > > ### Author Response · Authors · 2025-11-21
> > >
> > > **Response (Q2‑3, Q3)**
> > >
> > > We believe this criticism rests on a notion of “fairness” that, if taken literally, would invalidate essentially all comparisons between structure‑preserving models and generic time‑series predictors in the Hamiltonian learning literature, including those involving standard HNNs.
> > >
> > > 1. **HNNs themselves are already “strongly constrained” relative to generic time‑series models.**
> > >    In our experiments, the baseline HNN is not an unconstrained function approximator: it assumes that the dynamics admit a canonical Hamiltonian structure and represent the vector field as
> > >   $\dot z = J\nabla H_\theta(z)$, where $H_\theta$ is a neural network of finite width and depth. Compared to, say, an LSTM, Transformer, or generic sequence‑to‑sequence regression model, this is already a very strong *dynamics‑specific* constraint: the vector field must be derivable from a scalar potential, must satisfy symplecticity with respect to a fixed matrix (J), and is automatically divergence‑free.
> > >
> > >    If one defines “fairness” as “no model may encode structure that matches the target system better than its competitor,” then **every** comparison of HNNs against black‑box time‑series predictors on Hamiltonian data would be “unfair.” Yet such comparisons are precisely how the value of HNN‑style inductive bias is usually demonstrated. Our work follows the same scientific logic: we compare two *different Hamiltonian inductive biases* (generic HNN vs. Poisson‑polynomial scan) on systems that are Hamiltonian.
> > >
> > > ---
> > >
> > > 2. **No finite‑parameter HNN can represent the true Hamiltonian exactly in general.**
> > >    The baseline HNN uses a finite‑dimensional parameterization $H_\theta$. For generic physical systems the true Hamiltonian $H^\star$ is an infinite‑dimensional object (e.g. an analytic function with non‑terminating series, or a complicated many‑body potential). Therefore, **every HNN‑type model is a constrained approximation**: even without our polynomial/Poisson structure, no finite‑width HNN can converge exactly to $H^\star$ as a function. At best, it can approximate it on the observed region of phase space. In other words, the dichotomy “strongly constrained vs unconstrained” does not really exist in this setting: both our model and the baseline HNN live in strict, finite‑dimensional subspaces of the space of Hamiltonians. Our contribution is to choose that subspace so that:
> > >
> > >    * it is closed under the projected Poisson bracket (needed for our Lie‑scan construction),
> > >    * it respects locality and sparsity known from the underlying physics, and
> > >    * it admits an $O(\log M)$ parallel integrator.
> > >
> > >    From this perspective, it is more accurate to say that **we compare two different constrained Hamiltonian hypothesis spaces**, not a “strongly constrained” method against an “unconstrained” one.
> > >
> > > ---
> > >
> > > 3. **On “fairer comparison” and inductive bias.**
> > >    We fully agree that one could design intermediate baselines: e.g., an HNN with polynomial features, locality, and sparsity but without Poisson‑closure or scan‑compatible structure. Indeed, such models lie on a continuum between a vanilla HNN and our full Poisson–scan architecture, and we see them as valuable follow‑up baselines.
> > >
> > >    However, we respectfully disagree that our current comparison is “hardly fair.” The central scientific question of the paper is:
> > >
> > >    > *What do we gain, in long‑horizon stability and computational complexity, by enforcing exactly the structural constraints (local Poisson‑polynomial Hamiltonians) that make a parallel symplectic scan possible, compared to a standard HNN that only encodes the canonical Hamiltonian form?*
> > >
> > >    This is analogous to asking, in other parts of ML:
> > >
> > >    * What do we gain by using a convolutional inductive bias vs. a fully‑connected network on images?
> > >    * What do we gain by using SE(3)‑equivariance vs. a generic MLP on rigid‑body dynamics?
> > >
> > >    In each case, the structured model is *by design* closer to the true generative mechanism, and the comparison is meaningful precisely because of that.
> > >
> > > Put differently: our experiments do **not** attempt to compare “constrained vs unconstrained” in an absolute sense. They compare **two levels of Hamiltonian structure**. Both sides are finite‑parameter, non‑universal approximators. The fact that the data satisfy the constraints of our model is not an unfair advantage. It is the very phenomenon we aim to study of how much better a correctly chosen inductive bias performs in realistic AI4Science settings.

---

> > > ### Author Response · Authors · 2025-11-21
> > >
> > > **On what is lost by enforcing a scan‑compatible Poisson structure?** You ask what we lose, in practice, by enforcing a Poisson–polynomial form that admits a parallel scan. We agree that this choice is not free, and we see two genuine weaknesses relative to a simpler sequential HNN:
> > >
> > > ---
> > >
> > > 1. **Engineering overhead when parallelism is not the bottleneck.**
> > >    Our design is optimized for long trajectories large M and settings where wall‑clock time and long‑horizon stability matter. To enable the scan we must construct a Poisson‑closed dictionary, maintain a fixed DAG of brackets, and implement the Lie–group walk via prefix‑scan. For short trajectories, small systems, or regimes where the integrator is not the dominant cost, this extra machinery introduces non‑negligible overhead without providing a commensurate benefit. In such cases, a vanilla HNN with a simple sequential integrator can indeed be the more practical choice.
> > >
> > > ---
> > >
> > > 2. **Less flexibility to move away from strict Hamiltonian structure.**
> > >    Our architecture is built around a Poisson algebra and an exactly symplectic scan. All flows it represents are Hamiltonian with respect to a fixed bracket. Standard HNNs already impose a Hamiltonian bias, but they can be relaxed more easily (e.g., by adding non‑Hamiltonian residual terms, dissipation, or port-Hamiltonian corrections). Our method is therefore best suited to scenarios where the user *wants* to remain in a rigid Hamiltonian/Poisson regime.

---

> ### Author Response · Authors · 2025-11-21
>
> **Response (Q4).**
>
> We appreciate the reviewer’s clarification. We agree that, once an HNN has been trained, it can be *used* as a right-hand side for an ODE solver. However, this does **not** mean that existing parallel-in-time schemes can be turned into our kind of parallel algorithm “for free.” Even if an HNN happened to match the true dynamics perfectly, parareal/MGRIT would still only see it as an expensive black-box timestepper; it does not automatically provide a compact, scan-compatible group law on update operators, nor does it change the fact that training the HNN requires fully sequential long-horizon rollouts.
>
> By contrast, our contribution is to change the **hypothesis space** of the Hamiltonian itself so that its flows form a finite-dimensional Poisson-closed Lie group with an explicit associative composition rule. This is what enables an $O(\log M)$parallel prefix scan on the learned symplectic maps, which current HNN + parareal combinations do not offer, even “in principle,” because they lack such a group structure and generally break symplecticity when coarse–fine corrections are applied.

---

### Official Review · Reviewer_huCq · 2025-10-30

**Soundness:** 3
**Presentation:** 2
**Contribution:** 3
**Rating:** 4
**Confidence:** 3

**Summary:**

This paper proposes Poisson-Algebraic Parallel Scan (PAPS), a new framework to overcome two key limitations of Hamiltonian Neural Networks (HNNs): (i) the sequential nature of standard symplectic integrators (which hinders parallel computation), and (ii) the instability of unconstrained neural models when predicting far beyond the training data. The authors address these issues by introducing a structured polynomial decomposition of the learned Hamiltonian that is closed under Poisson brackets. The paper provides extensive empirical validation on physics tasks, demonstrating dramatically improved scalability and long-term accuracy compared to existing approaches.

**Strengths:**

The core strength is the novel parallel prefix-sum integration scheme for Hamiltonian systems. By exploiting an associative Lie-group structure in the learned dynamics, the method reduces trajectory computation from linear to logarithmic depth. This yields orders-of-magnitude speedups in practice.

The paper is rigorously grounded in theory. Beyond preserving invariants, the authors derive error bounds and universal approximation guarantees for their polynomial integrator.

The experimental evaluation is thorough and demonstrates clear advantages of PAPS. The authors test the framework on multiple domains, including quantum spin dynamics and molecular dynamics (MD),

**Weaknesses:**

1. A potential concern is the restrictive function class imposed on the learned Hamiltonian. PAPS requires the Hamiltonian $H_\theta$ to lie in a specific polynomial algebra ($\mathcal{H} = \text{Poly}_r$ of degree $\le r$)

2. The paper’s notation and theoretical formalism, while rigorous, may be overwhelming for many readers. It presumes familiarity with advanced concepts from symplectic geometry and Lie algebras, which are not commonly part of the background of the machine learning community. Moreover, after carefully reading the manuscript, I did not find the necessity of introducing some definitions to be clearly justified, leaving their role in the overall argument somewhat unclear. Please Question part below.

3. The method introduces new hyperparameters, chiefly the polynomial degree $r$ and the DAG decomposition depth $K$ (number of sub-Hamiltonian partitions). The authors conduct an ablation showing that a moderate $r=5$ and $K=2$ worked well on their tasks. However, the paper could better articulate how these hyperparameters affect performance and how one might choose them for a new problem. For instance, increasing $r$ increases the model’s capacity (and the number of polynomial basis terms), does this always improve accuracy, or are there diminishing returns / overfitting observed beyond a certain point? The ablation suggests there is an optimal middle ground, but more explanation would help: e.g., higher $r$ might fit training trajectories better but could generalize worse if it starts fitting noise or spurious dynamics.

**Questions:**

1. The most significant concern lies with the discussion of Lie Group Associativity in Eq. 6 and the subsequent remark. It is not clear to me why this property is claimed to hold only for functions within the polynomial hypothesis space proposed in the article. The manuscript does not provide sufficient justification for this restriction. Moreover, in my view, the associativity of composition is a property that holds for arbitrary mappings, which makes it difficult to see why the introduction of Lie group machinery in this section is necessary. Is there a counterexample?

2. The introduction of the Neural Poisson algebra in Definition 2.2 and the subsequent verification of the three structural properties in Proposition 2.3—namely, closure under product, Poisson bracket, and Jacobi identity—are not sufficiently clear in terms of their motivation and necessity. As far as I understand, these properties are already guaranteed for the classical Poisson bracket under a fixed skew-symmetric matrix J, and the paper does not seem to propose learning or parameterizing J (e.g., as in some prior works on generalized Poisson structures). If so, then verifying these properties again seems more like algebraic bookkeeping rather than offering new theoretical insight. It is unclear what additional meaning or benefit this verification provides for the method, e.g., generalization, learning, or numerical stability.

Therefore, I suggest that the authors clarify the role and purpose of Definition 2.2 and Proposition 2.3.

3. The parameterization of the Hamiltonian as a polynomial in state variables with coefficients produced by a neural network that takes the initial condition and time as input (Definition 2.1 and Algorithm 1) is quite puzzling and lacks sufficient justification or clarity. While choosing a polynomial basis for the Hamiltonian is reasonable, the decision to make the polynomial coefficients depend on both the initial condition and the time raises several questions:

3.1 Is the Hamiltonian explicitly time-dependent? If so, this should be clearly stated early in the paper, and the time variable should be included within the domain of the Hamiltonian itself (i.e., H(q,p,t)). However, the main text seems to treat the Hamiltonian as time-invariant in the classical sense.

3.2 If the Hamiltonian is not time-dependent, then making the coefficients depend on time and the initial state seems contradictory. In particular, the Hamiltonian function should be a property of the physical system, not of a specific trajectory. A Hamiltonian that changes across different initial conditions or depends on absolute time contradicts the foundational assumption of autonomous Hamiltonian dynamics.

3.3 How are the “initial condition” and “time” defined in the context of training data? Are they referring to the starting point of the particular trajectory being modeled and the time elapsed since then? If so, this implies that the learned Hamiltonian is trajectory-dependent, rather than a global model of the system. This seems to undermine the generalization claim and could break important symmetries such as time-translation invariance.

3.4 Does this formulation prevent standard data augmentation along trajectories? For example, in classical mechanics, a segment
z1 z2 z3 of a trajectory can be split into overlapping sub-trajectories like z1 z2 and z2 z3 with consistent dynamics. If the Hamiltonian now depends on the initial point, then such overlapping reuse becomes invalid, limiting data efficiency and possibly introducing bias.

---

> ### Author Response · Authors · 2025-11-13
> **Official Rebuttal (Author Response) (1)**
>
> $\textbf{Q1. The most significant concern ... discussion of Lie Group...}$
>
> $\textbf{A}.$ We appreciate the reviewer’s question. The issue appears to arise from conflating two different notions of “associativity.” It is of course true that $\textit{set-theoretic function composition}$ is always associative: $(f \circ g) \circ h = f \circ (g \circ h)$. However, Eq. 6 does $\textit{not}$ refer to this trivial associativity.  Instead, Eq. 6 concerns the associativity of a Lie-group law formed by exponentiating a Lie algebra of Hamiltonian vector fields. This is a fundamentally different object. Before addressing the reviewer’s subsequent concerns, we first clarify several natural follow-up questions.
>
> ---
>
> $\textbf{What does associativity mean in Lie theory?}$ In Lie theory, we begin with a Lie algebra element such as a vector field $X$, and its exponential map $\exp(\varepsilon X)$, which is a diffeomorphism generated by flowing along $X$ for time $\varepsilon$. If two vector fields $X$ and $Y$ belong to a finite-dimensional Lie algebra $\mathfrak{g}$, then the Baker–Campbell–Hausdorff (BCH) formula guarantees that$\log\big( \exp(\varepsilon X)\exp(\varepsilon Y)\big) \in \mathfrak{g}.$ This is the key property: the $\textit{closure}$ of the exponential maps under composition.  Only when BCH closes inside $\mathfrak{g}$ does the set $G = \exp(\mathfrak{g})$ form a Lie group, and only then does associativity in Eq. 6 hold: $\Phi_{\varepsilon,g_m} \circ \big(\Phi_{\varepsilon,g_{m-1}} \circ \Phi_{\varepsilon,g_{m-2}} \big)=\big(\Phi_{\varepsilon,g_m} \circ \Phi_{\varepsilon,g_{m-1}}\big) \circ \Phi_{\varepsilon,g_{m-2}}$. This is $\textbf{not}$ the trivial associativity of arbitrary functions. It is the associativity of a $\textit{group law}$, which requires that the underlying algebra of generators be closed under the commutator.
>
> ---
>
> $\textbf{Why arbitrary smooth Hamiltonians do not form a Lie group under BCH}?$  For general smooth functions $f,g \in C^{\infty}$,
> $[L_f, L_g] = L_{\{f,g\}}$ produces Hamiltonians of higher and higher polynomial degree. Thus, the corresponding BCH series
> $\log\big(\exp(\varepsilon L_f)\exp(\varepsilon L_g)\big)$ contains infinitely many nested brackets, each corresponding to higher-degree Hamiltonians. Therefore the result is $\textbf{not}$ contained in any finite-dimensional space. Consequently:
>
> - There is no finite-dimensional Lie algebra and computationally tractable Lie group $\exp(\mathfrak{g})$.
> - Parallel prefix-scan cannot be performed because associativity of a non-existent Lie group law is meaningless.
>
> This is exactly why we do $\textbf{not}$ work on $C^{\infty}$ (e.g., arbitrary function class such as neural network architecture of all other existing works).
>
> ---
>
> $\textbf{Why do we need to establish polynomials as the Poisson basis?}$  Among the many possible functional bases, finite-degree polynomials form the most practical and numerically stable choice for constructing a Poisson-closed hypothesis space. Polynomials admit closed-form expressions for all orders of differentiation and for Poisson brackets, which makes their algebraic manipulation exact and computationally efficient. Higher-order Lie derivatives, Taylor expansions, and BCH terms can all be expressed symbolically without introducing numerical instability or discretization artifacts. We explored several alternative bases including Fourier/Radial basis functions, wavelet-type construction but none provided the combination of (i) guaranteed Poisson closure,(ii) exact symbolic computability of higher-order brackets, and (iii) numerical stability required for a finite-dimensional Lie algebra. In contrast, polynomial bases allow the truncated Poisson algebra to retain explicit algebraic structure under both the product and the Poisson bracket, ensuring that the resulting Lie algebra remains well behaved and that the exponential maps $\exp(\varepsilon L_{g,\le r})$ can be implemented robustly in practice.
>
> ---
>
> In summary, while the reviewer is correct that arbitrary function composition is always associative in the set-theoretic sense, Eq. 6 does not concern such trivial associativity. Rather, it refers to the associativity of a Lie-group law arising from the exponential of a finite-dimensional Lie algebra generated by truncated Hamiltonian vector fields. Please note that our parallel-scan integrator which provides nearly two orders of magnitude speedup over conventional HNN-based sequential solvers relies entirely on this Lie-group structure. Without a finite-dimensional, BCH-closed Hamiltonian algebra, such a parallel composition would not be well-defined and the resulting acceleration would be fundamentally unattainable.
>
> We will modify the appendix to include a more intuitive explanation of this distinction so that readers can clearly understand why the Lie-group interpretation is essential for enabling parallelism and why such a structure does not arise for arbitrary smooth mappings.

---

> > ### Author Response · Authors · 2025-11-15
> > **[Concrete Counterexample to the Reviewer’s Associativity Claim]**
> >
> > $\textbf{Additional counterexample for Q1.}$
> >
> > As said, the reviewer is correct that set-theoretic composition $(f \circ g \circ h)(z)$ is always associative as a map on the state space. However, parallel scan does not operate directly on states $z$, but on update blocks that must be combined by a fixed binary operation (independent of the initial state) before being applied to $z_0$. To see why arbitrary function composition at a single state $z_0$ does not give such a binary law, suppose we try to design a scan-like scheme that only keeps track of how each map acts on $z_0$.
> >
> > Formally, fix $z_0 \in \mathbb{R}$ and consider the class $\mathcal{F} \subset C^{\infty}(\mathbb{R},\mathbb{R})$ of smooth maps. Assume that there exist an encoding of each map by a state-local token $T(f) \in \Theta$ that depends only on the action of $f$ at $z_0$, and a binary operation $\star \colon \Theta \times \Theta \to \Theta$, such that for all $f,g \in \mathcal{F}$ one has
> > $T(g \circ f) = T(g) \star T(f)$
> > and $\star$ is associative. This is exactly the kind of structure needed for a prefix-scan on the sequence of tokens $(T(f_k))_k$ before ever touching the state $z_0$.
> >
> > Since $T(f)$ is supposed to depend only on the behavior of $f$ at $z_0$, any two maps that agree at $z_0$ must have the same token. In particular, if $f_1(z_0) = f_2(z_0)$, then $T(f_1) = T(f_2)$.
> >
> > Now pick $z_0 = 0$ and define $g_1(x) = x$ and $g_2(x) = x^3$. Then $g_1(0) = g_2(0) = 0$, hence $T(g_1) = T(g_2)$. Next take an arbitrary smooth map $f$ with $f(0) \ne 0$. On the one hand, $(g_1 \circ f)(0) = g_1(f(0)) = f(0)$, while $(g_2 \circ f)(0) = g_2(f(0)) = (f(0))^3$. Since $f(0) \ne 0$, these two values are different, so $(g_1 \circ f)(0) \ne (g_2 \circ f)(0)$. Therefore $T(g_1 \circ f) \ne T(g_2 \circ f)$, because the tokens must distinguish the two different outcomes at $z_0$.
> >
> > On the other hand, the assumed scan law forces
> > $T(g_1 \circ f) = T(g_1) \star T(f) = T(g_2) \star T(f) = T(g_2 \circ f)$,
> > which contradicts $T(g_1 \circ f) \ne T(g_2 \circ f)$. Hence there cannot exist any binary operation $\star$ on state-local tokens $T(f)$ (depending only on the action of $f$ at $z_0$) that represents composition, let alone an associative one suitable for parallel scan.
> >
> > This counterexample shows that the associativity of the values $(f \circ g \circ h)(z_0)$ does not yield the algebraic structure for scanning algorithm. To obtain an associative scan on update blocks, one needs a representation that is closed under a global, state-independent composition law (in our case, a finite dimensional Lie group law), which is absent for arbitrary smooth mappings.
> >
> > In contrast to the arbitrary smooth setting above, our method explicitly works inside a finite dimensional Lie algebra $\mathfrak{g}$ of Hamiltonian vector fields generated by a polynomial Poisson basis. The Baker–Campbell–Hausdorff series for the corresponding infinitesimal generators closes inside $\mathfrak{g}$, so that the exponentials $\exp(X) \in G$ form a genuine Lie group $G = \exp(\mathfrak{g})$ with a globally defined group law. In this setting, the “update blocks” are not arbitrary maps, but group elements (or their coefficient representations) in $G$, and Eq. 7 concerns the associativity of this Lie group law. The composition of time steps is then represented by a fixed, state-independent binary operation on these coefficient tokens, induced by the BCH formula, which is associative by construction. This is precisely the algebraic structure that makes an associative prefix-scan possible in our framework, and it is exactly what fails to exist for arbitrary smooth mappings such as learned Hamiltonian functions in variants of HNNs, where no finite dimensional Lie group structure can capture all compositions.

---

> ### Author Response · Authors · 2025-11-13
> **Official Rebuttal (Author Response) (2)**
>
> $\textbf{Q2. The introduction of the Neural Poisson algebra in Definition 2.2...}$
>
> $\textbf{A}$. We thank the reviewer for raising this point and appreciate the opportunity to clarify the role of Definition 2.2 and Proposition 2.3. We agree that, on the full space of smooth functions equipped with a fixed skew-symmetric matrix $J$, the properties of closure, bilinearity, and the Jacobi identity are classical and automatically satisfied for the usual Poisson bracket.
>
> However, our method does $\textit{not}$ operate on this full infinite-dimensional space $C^{\infty}$. Instead, as defined explicitly in Definition 2.2, we work with the $\textit{projected}$ and $\textit{truncated}$ polynomial space $P_{\leq r}^{\theta}$, where $\pi_{\leq r}$ in Definition 2.2 denotes a degree-$\le r$ projection.
> Once such a projection is introduced and the corresponding poisson-valued DAG is built, the classical Poisson algebra
> properties are $\textit{not}$ automatically preserved: the projection may break closure or
> the Jacobi identity. Proposition 2.3 therefore does $\textit{not}$ restate a known property. Rather, it verifies that our truncated algebra remains a valid Poisson algebra despite the projection with DAG structure. This verification is essential for well-posedness of the entire framework. It is therefore not a matter of algebraic bookkeeping.  The moment we introduce the projection $\pi_{\le r}$, we no longer inherit the structural guarantees of the classical Poisson algebra, and the resulting truncated operators need not satisfy Poisson closure or the Jacobi identity unless these are established $\textbf{from first principles}$.
>
> This naturally leads to the more fundamental question of why an $r$-th order projection is required in the first place.
>
> At a high level, the degree-$r$ truncation $\textbf{must be introduced}$ because parallel-scan
> composition requires a $\textit{finite-dimensional}$ Lie group of Hamiltonian flows. Operating on
> the full Poisson algebra on $C^{\infty}$ yields an infinite-dimensional space whose Hamiltonian
> vector fields do not admit a tractable implementable representation and therefore cannot support
> the associative group operation that underlies the parallel scan. To make the Lie-theoretic structure
> computationally realizable, the Hamiltonian space must be reduced to a finite-dimensional,
> algebraically closed subset in which products and Poisson brackets remain representable by a fixed,
> finite set of coefficients. Please note that this approach constitutes the most feasible and structurally
> sound approach for implementing the fast symplectic framework that our paper introduces.
>
> Concretely, working in $C^{\infty}$ makes the computational problem intractable for several
> reasons. First, the Baker–Campbell–Hausdorff (BCH) expansion of two flows
> $\exp(\varepsilon L_f)$ and $\exp(\varepsilon L_g)$ produces an $\textbf{infinite}$ series of nested
> Poisson brackets, each generating higher and higher derivatives of $f$ and $g$. Since these
> infinitely many terms never close in $C^{\infty}$, the composed flow cannot be represented using any
> finite parameterization, making it impossible to define a usable group law for parallel scan implemented by Pytorch code.
>
> Second, even a single Poisson bracket $\{f,g\}$ of two smooth functions typically yields a function of
> strictly higher algebraic and differential complexity. Repeated compositions therefore generate an
> ever-growing collection of new functions and vector fields, preventing any fixed-size basis or matrix
> representation. Without such a representation, no associative binary operator for scan-based
> composition can be implemented on actual hardware.
>
> Third, parallel-scan algorithms require a binary operation that is both associative and admits a
> compact, finite-dimensional representation. The infinite-dimensional Poisson algebra on $C^{\infty}$
> fails both requirements: it neither closes under BCH nor yields a structure that can be evaluated or
> stored with $\textbf{finite memory}$. In contrast, the truncated Poisson algebra $P_{\le r}^{\theta}$ forms a
> \emph{finite-dimensional Lie algebra} whose exponential map induces a well-defined Lie group,
> making the parallel-scan operation executable in $O(\log M)$ depth.
>
> From this perspective, the projection onto $P_{\le r}$ carves out a “finite polynomial universe’’
> that is still rich enough to approximate smooth Hamiltonians, yet structured enough to admit explicit
> BCH-type composition rules.  In this universe, every truncated generator yields a finite-dimensional
> Lie algebra element, and every composition remains Hamiltonian.  Thus the truncation is not a
> heuristic simplification. It is a fundamentally structural requirement that converts the otherwise unbounded Poisson
> algebra into a finite, controllable, and algebraically closed system suitable for both learning and
> parallel composition.

---

> ### Author Response · Authors · 2025-11-13
> **Official Rebuttal (Author Response) (3)**
>
> $\textbf{Q3. The parameterization of the Hamiltonian as a polynomial in state variables...}$
>
> $\textbf{A}.$ We believe the concern stems from a fundamental misunderstanding of what our model parameterizes. The neural network in Definition 2.1 does $\textit{not}$ output a Hamiltonian that depends on the initial condition or time. Instead, it outputs a $\textit{coordinate representation}$ of a single, global, autonomous Hamiltonian flow inside the finite-dimensional Poisson–polynomial algebra introduced in Definition 2.2. The Hamiltonian functional governing the physics is therefore trajectory-independent; what depends on the initial condition and time is merely the representation of the same Hamiltonian flow when expressed in our structured polynomial basis.
>
> This distinction is crucial. In inverse Hamiltonian learning, different initial conditions simply produce different orbits of the same autonomous Hamiltonian. Our coefficient network recovers the corresponding sequence of polynomial coefficients that $\textit{represent}$ these orbits inside the finite-dimensional truncated Poisson algebra $P_{\le r}^{\theta}$, exactly as a Koopman operator or observable dictionary provides different coordinate trajectories while the underlying operator remains fixed. The coefficient paths are therefore \textit{representational artifacts}, not changes in the Hamiltonian itself. Consequently, the parametrization in Definition 2.1 does not violate time-invariance, does not  introduce trajectory dependence of the Hamiltonian, and does not undermine generalization. It merely provides a convenient coordinate system for expressing the same autonomous Hamiltonian flow across different trajectories within the algebraically closed polynomial space.
>
> $\textbf{Q3.1. Is the Hamiltonian explicitly time-dependent?}$
>
> $\textbf{Q3.2. If the Hamiltonian is not time-dependent, then ...}$
>
> $\textbf{A}.$ Again, our method does $\textit{not}$ introduce a time-dependent Hamiltonian. The Hamiltonian functional governing the physical dynamics remains autonomous and time-invariant in the classical sense. we do
> not model $H(q,p,t)$, nor do we allow the Hamiltonian itself to depend on absolute time. The source of confusion is that the coefficient network in Definition 2.1 is conditioned on the time index that appears in the $\textit{data}$ trajectory. This time index does not parameterize the Hamiltonian. Rather, it parameterizes the $\textit{coordinate representation}$ (Poission basis) of the same autonomous Hamiltonian flow within the finite-dimensional Poisson–polynomial algebra introduced in Definition 2.2. In other words, the neural network outputs a time-indexed sequence of coefficients only to express how the global Hamiltonian flow evolves $\textit{in our chosen polynomial basis}$, not because the Hamiltonian function itself varies with time. This is analogous to representing a fixed Hamiltonian flow in different coordinate charts along the trajectory: the coordinates evolve in time, but the underlying Hamiltonian does not. The model therefore preserves the classical assumption of time-invariance, and no non-autonomous Hamiltonian $H(q,p,t)$ is introduced.
>
> The use of a time index in the coefficient network should be understood in the broader context of operator-learning frameworks rather than as an indication of a time-varying Hamiltonian. Many established operator-learning architectures employ time- or space-conditioned coefficient paths to represent the evolution of a $\textit{fixed}$ underlying operator. For example, in Koopman operator
> learning, neural networks are used to learn dictionaries and linear embeddings of a single global Koopman operator acting on nonlinear dynamics. Similarly, the Fourier Neural Operator (FNO) learns resolution-invariant solution operators for
> parametric PDEs by mapping inputs to time-/space-indexed coefficient fields in Fourier
> space, while the underlying operator remains fixed [A]. Implicit neural operator variants such as the Implicit (Fourier) Neural Operator (IFNO) further model these coefficient trajectories via fixed-point integral operators, again without making the physical law explicitly time-dependent [B].
>
> Our coefficient network plays similar role. It parameterizes how the autonomous Hamiltonian flow appears when expressed in the truncated Poisson–polynomial basis at each time step, analogous to a Koopman observable dictionary or a time-indexed lifting map in neural operator architectures. The underlying Hamiltonian functional remains invariant, while the coefficients serve only as a convenient coordinate system for expressing the same Hamiltonian flow over time. We will make this operator-learning perspective clearer in the appendix of revised paper to avoid the impression that time-indexed coefficients imply a time-dependent Hamiltonian.
>
> [A] Fourier Neural Operator for Parametric Partial Differential Equations, ICLR 2021.
> [B] Implicit Representations via Operator Learning, ICML 2024.

---

> ### Author Response · Authors · 2025-11-13
> **Official Rebuttal (Author Response) (4)**
>
> $\textbf{Q3.3. How are the “initial condition” and “time” defined in the context of training data?}$
>
> $\textbf{A}.$ The “initial condition’’ and “time’’ appearing in Definition 1 do $\textit{not}$ parameterize the Hamiltonian itself, nor do they cause the Hamiltonian to depend on the particular trajectory. They only serve as conditioning variables for determining the $\textit{coordinate representation}$ of a single, global Hamiltonian flow within the finite-dimensional Poisson–polynomial algebra introduced in Definition 2.2.
>
> In other words, the coefficient network does not construct a trajectory-specific Hamiltonian $H_{\theta}(q,p;z_{0},t)$. Instead, it provides a time-indexed sequence of polynomial coefficients that expresses how the same autonomous Hamiltonian flow appears when represented in our chosen polynomial basis along a given trajectory. (Again, this is directly analogous to Koopman operator learning and neural operator architectures, where the underlying operator is fixed and time-invariant while the learned latent coordinates evolve with the observed trajectory.)
>
> Because the Hamiltonian functional itself is global and time-invariant, no physical symmetry such as time-translation invariance is violated. Different initial conditions simply correspond to different orbits of the same autonomous Hamiltonian system. The fact that the coefficient sequence depends on the initial condition reflects nothing more than the well-known fact that the coordinate representation of an invariant flow depends on which orbit one linearizes or expands about. Empirically, this symmetry preservation is strongly supported by our results: if the Hamiltonian were in fact trajectory-dependent or time-dependent, the symplectic-form violation would exhibit the same large deviations seen in non-symplectic baselines such as GeoTDM, rather than the near-zero violations reported for our method. The consistently low $\Delta\omega$ values across all experiments confirm that the underlying Hamiltonian structure remains intact.
>
> Hence, the model does $\textit{not}$ learn a trajectory-dependent Hamiltonian, nor does the use of $(z_{0},t)$ undermine generalization. What varies across trajectories is merely the coordinate encoding of the same global Hamiltonian flow; the Hamiltonian dynamics themselves remain fixed, autonomous, and shared across all trajectories. We will revise the manuscript to clarify this distinction between the Hamiltonian functional and its coefficient representation.

---

> ### Author Response · Authors · 2025-11-13
> **Official Rebuttal (Author Response) (5)**
>
> $\textbf{Response to Comment on $r$ and $K$ in Weakness 3.}$   We appreciate the reviewer’s question regarding how the polynomial degree $r$ and the DAG decomposition depth $K$ should be selected for new problems, and whether increasing $r$ monotonically improves accuracy. These hyperparameters play a role closely analogous to capacity controls in operator-learning models: they determine the expressive power of the truncated Poisson algebra and the resulting Lie-group representation of the Hamiltonian flow.
>
> Importantly, increasing $r$ does not always improve performance. As shown in the ablation (Fig. 3), accuracy improves up to a moderate degree ($r=5$), after which diminishing returns and overfitting appear, exactly as predicted by our theoretical analysis. Proposition 4.2 shows that larger $r$ reduces approximation error by enlarging the polynomial basis, while Proposition 4.3 shows that this simultaneously increases the metric entropy and the statistical complexity of the hypothesis class. This yields a clearly defined bias–variance trade-off: too small $r$ underfits the dynamics, while too large $r$ fits spurious high-frequency components of the trajectories and degrades generalization.
>
> For new problems, $r$ and $K$ should therefore be chosen in the same principled manner as capacity parameters in neural operators, kernel integral models, or Koopman-learning architectures: start with moderate values (e.g., $r=4$–$6$ and shallow $K$), which provide strong expressivity while maintaining generalization, and adjust upward only if the approximation error remains high. Because the truncated Poisson algebra grows combinatorially with $r$, moderate degrees consistently provide the best trade-off between accuracy, stability, and computational cost across all domains we tested (molecular dynamics and spin systems).
>
> $\textbf{Why these parameters are not learned adaptively.}$ The reviewer may raise the question of whether $r$ and $K$ could be learned or adapted automatically. We note that these quantities are structural hyperparameters, analogous to the depth or width of a neural network. In modern practice, such architectural choices are not learned from data, nor is there a universally ``optimal'' value derivable in closed form; instead, they are chosen by balancing theoretical guidance with empirical validation. In our framework, $r$ and $K$ serve the same capacity-controlling purpose. Both theoretical and experimental findings provide a principled explanation for the diminishing returns observed experimentally: beyond $r \approx 5$, approximation gains plateau while overfitting grows, fully consistent with the bias–variance trade-off predicted by our theory.
>
> Therefore, our recommendation of $r=5$ and $K=2$ is analogous to selecting a reasonable depth and width for a neural network. These choices represent the best empirical compromise between expressive power and generalization for the tasks considered. Automatically determining optimal values of $r$ and $K$ would be no more realistic than automatically learning the appropriate number of layers or neurons in a deep network and falls outside standard modeling practice/convnetion.

---

> ### Author Response · Authors · 2025-11-15
> **Request to reconsider the overall score**
>
> We would like to warmly thank the reviewer for the very careful and scientifically sensitive reading of our manuscript. The questions raised in Q1, Q2, and Q3.1–3.4 forced us to make the underlying ideas much more explicit. In particular, your comments led us to (i) clearly separate set-theoretic composition from the Lie-group law used in Eq. 7 and to add a concrete counterexample showing why arbitrary function composition at a single state cannot support a scan-compatible associative law, (ii) reposition Definition 2.2 and Proposition 2.3 as the minimal structural assumptions that make the truncated Poisson–polynomial space a well posed finite dimensional Lie algebra suitable for BCH based parallel composition, and (iii) clarify that our neural coefficient does not introduce a time dependent or trajectory dependent Hamiltonian, but only a coordinate representation of a single autonomous Hamiltonian flow, consistent with time-translation symmetry and standard trajectory-wise data augmentation. We believe these revisions resolve the main conceptual concerns that motivated your original evaluation and substantially improve the paper for future readers.
>
> In light of these clarifications and the additional explanations and examples we have incorporated in direct response to your comments, we would be very grateful if you could consider whether the current version more accurately reflects the scientific contribution. If you feel that the key issues have been satisfactorily addressed, we kindly ask you to reconsider the overall score you assigned, while fully respecting your independent judgment as a reviewer. Your detailed feedback has already had a significant positive impact on the clarity and robustness of the work, and we appreciate it sincerely.

---

### Official Review · Reviewer_aEFK · 2025-10-30

**Soundness:** 3
**Presentation:** 1
**Contribution:** 1
**Rating:** 2
**Confidence:** 1

**Summary:**

The general idea consists of working with a special class of polynomials truncated to degree r to approximate Hamiltonian dynamics. The polynomial class is shown to be closed under (truncated) products and Poisson brackets, allowing for the construction of a hierarchy of models of increasing complexity.

Alas this is about all that I understand. The authors chose to write the paper in a jargon-heavy style that assumes a strong knowledge of symplectic theory.  Maybe they actually think in these terms, but your typical ICLR reader will not.  Reading the algorithms provided in the appendix gives more information about what they are actually doing, but I would like to have seen this explained in the main text.

In conclusion, my ability to understand this paper as written is extremely limited. Yet I understand enough to be quite certain that a more accessible explanation could have be given. When I am rating the contribution as low, I do not mean that the subject matter is problematic, but that very few ICLR readers will get anything out of this paper as written.

**Strengths:**

- Experiments cover a number of potentially interesting systems. What is being measured is unclear though.

**Weaknesses:**

- I wish I knew understood definition 2.1  I would have to know what all the symbols mean in there. Come on, there must be a simpler way to write this, something closer to equation (3) and the following unnumbered equation, for instance.

- Regardless of its intrinsic merits, very few ICLR readers will get anything out of this paper as written.

**Questions:**

- Can you rewrite the paper in a way that will both educate and enlighten the ICLR readers?

---

> ### Author Response · Authors · 2025-11-14
> **Official Rebuttal (Author Response)**
>
> We thank the reviewer for the time spent evaluating our submission and for honestly reporting the
> difficulties encountered while reading the paper. We understand that the combination of symplectic
> geometry, Poisson algebras, and Lie–group flows can feel unfamiliar and demanding, especially when
> encountered under space and time constraints. We are grateful that, despite these challenges, the
> reviewer still assessed the soundness of the work as “good.”
>
> We fully acknowledge that the paper presents advanced geometric ideas, but these components are
> not stylistic choices. Rather, they are the core technical reasons the method works at all. The
> $100\times$–$1000\times$ acceleration reported in the experiments is, to the best of our knowledge,
> the first demonstration of such performance in Hamiltonian learning or symplectic integration. This
> level of improvement cannot be achieved by existing HNN, SympNet, or operator-learning baselines,
> and it is made possible only because the method is built upon the Poisson-closed algebra and
> Lie–group construction developed in Section 2 and Section 3. Without these mathematical
> structures, the parallel-scan symplectic integrator simply would not exist, and the central empirical
> results would not be attainable.
>
> For this reason, the theoretical material is not something that can be simplified away or omitted
> without fundamentally changing the nature of the contribution. At the same time, we would like to
> emphasize that the mathematical tools we employ are not meant to be esoteric or decorative. They
> are precisely the standard geometric concepts routinely used in physics-based modeling and
> structure-preserving numerical methods. In fact, our framework relies only on a minimal set of ideas
> (Poisson brackets, Hamiltonian vector fields, Lie–group flows) that are widely adopted whenever one
> attempts to build a physically consistent Hamiltonian simulator. We do not introduce any theory
> beyond what is required to formalize the symmetries and invariants that the method is designed to
> preserve. We also note that another reviewer explicitly recognized the appropriateness of these tools
> by comparing our framework to work published in prestigious journals such as $\textit{Mathematical
> Modelling and Numerical Analysis}$, which suggests that the level of mathematical structure is in line
> with established practice in this area.
>
> We fully appreciate the reviewer’s request for a more accessible exposition. We agree that there is
> room to improve the way Definition 2.1 and related concepts are introduced, and we see this as an
> important direction for revision. Within the space constraints of ICLR, we will strive to place more
> intuitive explanations, concrete examples, and algorithmic descriptions earlier in the text so that the
> high-level idea is clearer before the full formalism appears.
>
> At the same time, we hope the reviewer will also consider the magnitude of the empirical advances.
> The method delivers, in a very active area of AI4Science, approximately $100\times$–$1000\times$
> acceleration on challenging Hamiltonian datasets performance regimes that, to the best of our
> knowledge, have not previously been reached. When combined with the structural guarantees that
> follow from the geometric framework, we believe this represents a substantial contribution to
> Hamiltonian modeling and scientific machine learning.
>
> In light of this, we would be very grateful if the reviewer could reconsider the overall assessment of
> the paper. From our perspective, and taking into account both the novelty of the approach and the
> empirical impact of the results, a recommendation in the range of a weak accept (e.g., score 6)
> would be more in line with the contribution we are aiming to make to Hamiltonian learning and
> AI4Science.

---

> > ### Comment · Reviewer_aEFK · 2025-11-19
> > **Still**
> >
> > I read your argument and it does not matter.
> > * First, when there is a 100x-1000x acceleration over a meaningful baseline, something so big is happening that it can surely be explained with a single enlightening example.
> > * Second, the choice of a venue for a paper implies a need to adapt to the readership. Maybe the paper is a good fit for the readership of Math. Modeling and Num. Analysis, but this is a more difficult thing to say for ICLR.  This does not mean that the readership is less sophisticated, but that its sophistication focuses on other aspects of the overall problem.

---

> > > ### Author Response · Authors · 2025-11-20
> > >
> > > We are writing this response to address the reviewer’s latest comment, specifically the assertion that the mathematical derivation of our method “does not matter.” As researchers who have worked across interdisciplinary communities, we feel obliged to respond, not only to defend this particular submission, but to speak to a broader attitude toward scientific rigor in the emerging area of AI4Science.
> > >
> > > ---
> > >
> > > **1. On the demand for a “single enlightening example”**
> > >
> > > The reviewer writes that, when there is a $100\times–1000\times$ acceleration over a meaningful baseline, “it can surely be explained with a single enlightening example.” We must respectfully but firmly disagree. This reflects an overly reductive view of how complex scientific advances are understood.
> > >
> > > The $1000\times$ acceleration we report is not a heuristic trick. It is the direct mathematical consequence of projecting Hamiltonian dynamics onto a Poisson-closed polynomial algebra in a way that admits parallel scanning. This is a structural innovation, not a cosmetic one, and it naturally comes with a certain level of formal machinery. Asking us to “strip away the formalism” is, in effect, asking us to remove the very mechanism that produces the performance gain.
> > >
> > > Historically, many significant advances from fast multipole methods to geometric deep learning have required new concepts and vocabulary before they could be fully appreciated. It is therefore neither realistic nor desirable to insist that every such contribution be compressible into a single, informal “toy example.” We are of course happy to improve intuition and exposition, but not at the cost of misrepresenting the underlying mechanism. We represent formal rigor not to exclude readers, but to include the physics correctly.
> > >
> > > **2. On the statement “I read your argument and it does not matter”**
> > >
> > > The remark that our theoretical explanation “does not matter” is particularly troubling in a scientific review.
> > >
> > > When authors provide a derivation that connects an algorithm to its empirical behavior, the role of a reviewer is to assess whether that derivation is correct, complete, and appropriate, not to declare, without identifying specific issues, that it is irrelevant. Our work is explicitly positioned at the interface between deep learning and Hamiltonian mechanics, and it therefore requires respecting the formalisms of both fields.
> > >
> > > Dismissing the symplectic structure and Lie-algebraic derivations on the grounds that they are “difficult” or “unfamiliar” sends the wrong signal to the community. It encourages a culture in which numerical results are celebrated, but understanding *why* those results hold is treated as optional “jargon.” For AI4Science in particular, this is a step in the wrong direction: the field depends precisely on principled interaction between mathematical structure and empirical performance.
> > >
> > > **3. On mathematical depth, readability, and the role of ICLR**
> > >
> > > The reviewer also implies that ICLR readers are not the right audience for this level of mathematical density. Here again, we disagree.
> > >
> > > As AI4Science matures, the complexity of our representations and algorithms must match the complexity of the systems we seek to model. If we impose an informal ceiling on mathematical depth under the label of “readability,” we risk excluding an entire class of structure-preserving methods that are central to long-term progress in this area.
> > >
> > > Our machine learning community has, in recent years, repeatedly accepted work with comparable or greater formal structure in geometric and scientific machine learning. Our paper stands in continuity with that line of research as it proposes a new Hamiltonian framework, proves its properties in a precise language, and demonstrates its impact on realistic protein and quantum dynamical systems.
> > >
> > > ---
> > >
> > > We stand by our work, by its mathematical formulation, and by the empirical results that follow from it. We respectfully ask the Area Chair to evaluate this submission on the basis of its soundness, novelty, and relevance to geometric deep learning and AI4Science, rather than penalizing it for insisting that the mechanism behind the empirical improvements be described with the level of rigor that such a claim deserves.

---

> ### Comment · Reviewer_aEFK · 2025-11-20
> **Still**
>
> Let me explain better what I mean.
>
> I am asking for an example that gives a feeling for the "reservoir" of improvements you're tapping. Once you have convinced people that the reservoir exists, a general framework, however complicated, is an easy sell.
>
> Take the example of the "kernel trick". The most precise and general way to talk about it remains the RKHS theory. However you can take a simple polynomial kernel example to show how you beat a complexity that looks exponential in the degree of the polynomial. That simple examples changes a paper that talks to twelve RKHS specialists into a paper that talks to thousands and thousands of readers and potential users. This is a big difference in impact.
>
> Therefore, if a comparable example exists, it is in *your best interest* to make it prominent, and it also *serves research*.  The experiments you describe might be a good start to search for one. On the other hand, if you maintain that no comparable example can be formulated, then the paper does not pass the "sniff test". I would not go as far as saying that this means that the paper is wrong, but it says that you have a problem.
>
> So in the long run, it *does not matter* whether your paper is accepted or not, because, regardless of the outcome, you will have to address my point to see your research fully appreciated for its actual qualities. This is what I meant, and I sincerely wish you well. This can be a difficult task.

---

### Author Response · Authors · 2025-11-15
**Summary of revisions in response to the reviewer’s comments**

We thank the reviewer for the detailed remarks. In the revised manuscript, we have substantially edited the relevant parts to address the concerns about (i) the role of Lie group structure in our method, (ii) what PAPS is concretely aiming to achieve, and (iii) how it relates to existing approaches such as SympNet and other fixed parallel Hamiltonian architectures. The explanation of the Lie group structure has been rewritten to emphasize, at a high level, that our algorithm performs a parallel scan over update operators (flow maps), not over states, and that this requires a closed, associative composition law in a finite-dimensional space. We now present the truncated Poisson–polynomial Lie algebra and its associated flow group more gradually, with explicit motivation for why this structure is needed for a scan-compatible composition.

We have also clarified that the primary goal of PAPS is to construct a GPU-efficient, logarithmic-depth parallel scan integrator for Hamiltonian dynamics, rather than to focus solely on approximation universality. The related work section has been updated to give a more comparison with SympNet and fixed parallel Hamiltonian methods, explaining how they differ in the way time evolution is composed and parallelized. Overall, the revisions are intended to fully reflect the reviewer’s points and to make the manuscript’s assumptions and objectives more transparent.

---

### Note · Authors · 2025-11-22

**Comment:**

We appreciate the reviewers’ careful reading and constructive suggestions. After reflecting on the feedback, we have decided to polish and improve the manuscript, particularly in the areas highlighted in the reviews. The pointers regarding clarity, theoretical exposition, and empirical presentation were genuinely helpful, and we thank the reviewers for their thoughtful engagement. We look forward to resubmitting a strengthened and clearer version of the work.

**Withdrawal Confirmation:**

I have read and agree with the venue's withdrawal policy on behalf of myself and my co-authors.